# Systemwide energy return on investment in a sustainable transition towards net zero power systems

Hasret Sahin [1] ✉, A. A. Solomon [1] ✉, Arman Aghahosseini [1] & Christian Breyer [1]

The Glasgow Climate Pact articulated the vital importance of renewables in reducing emissions on the way to net-zero pledges. During the power sector transition, foreseeing conditions affecting the plausibility of pathway options is crucial for specifying an optimal system development strategy. This study examines the net energy performance of nine decarbonisation global energy transition scenarios until 2050 by applying a newly developed systemwide energy return on investment (EROI) model. All scenarios result in an EROI value above the upper limit of the net energy cliff, expected to be around 10. EROI trends heavily depend on transition paths. Once achieving higher renewable energy shares begin requiring significant enabling technologies, EROI continually declines as the shares increase. Shortening the transition period leads to a sharper declining of EROI, which stabilises after achieving 100% renewables. The vulnerability arising from natural gas and oil depletions may have worst impact on EROI of fossil fuels dominated systems.

Enormous efforts have been poured into altering modern power systems to reduce dependence on fossil fuels. The Russian war in Ukraine has clearly shown that despite all efforts, fossil fuel dependency is still a sore point of present power systems considering the immense turbulence in commodity markets due to the gas crisis and despite the depotentiation of fossil fuel consumption[1]. This has further steered the overlong debate regarding the extent to which the dependency on fossil fuels can be reduced through energy efficiency and the rapid deployment of renewable energy (RE) sources[2]. The Glasgow Climate Pact[3] elucidated that the efforts to fulfil nationally determined contributions (NDCs) targets remain ineffective and the chance of keeping the global warming below 2.0°C is less than 50%. Prior NDCs targets were updated and a few countries including China pledged to accelerate their energy transition (ET) to achieve net-zero $CO_2$ or net-zero greenhouse gas (GHG) emissions by 2050[3,4]. However, the long-term transition towards net-zero $CO_2$ emissions is a path full of obstacles affected by biophysical, economic, political, and technological limitations[5], as reducing the current emissions to almost zero requires a radical decarbonisation for all sectors. The eminent impact of

diverging transition paths[6] requires various tools to narrow the path selection.

Energy return on investment (EROI) has been widely used as a metric indicator in energy studies. Fundamentally, it is the ratio of the energy output of a system or a technology to the energy invested in building and operating that system or technology. The pioneering EROI studies present diverse concepts due to methodological inconsistencies[7,8], with some deriving varieties of EROI concepts[9–12], implementing different boundary conditions[13–15], comparing fossil fuel and RE technologies with or without enabling technologies[16–19]. However, recent studies[20–23] diverted this attention towards EROI analysis for ETs[24,25] to foresee the feasibility of 100% RE systems[26]. Some of these studies questioned the plausibility of 100% RE in terms of net energy production[20,22]. A recent study that implemented an advanced systemwide EROI approach has challenged these studies' conclusions[27] and suggested that fundamental gaps of existing EROI estimation techniques, such as the inability to capture the impact of optimal interoperability of multi-processes, and the typical methodological gaps of EROI[28], may lead to such conclusions. Thus, applying an

[1]School of Energy Systems, LUT University, Yliopistonkatu 34, 53850 Lappeenranta, Finland. ✉e-mail: hasret.sahin@lut.fi; solomon.asfaw@lut.fi

improved systemwide EROI tool to a global transition scenario can eliminate these salient issues while simultaneously contributing to enhancing the ET path selection.

Hereby, this study expands the newly developed Excel-based LUT-EROI model (LUT stands for Lappeenranta-Lahti University of Technology) to study global systemwide EROI using the nine global transition scenarios presented in Aghahosseini et al.[6], to evaluate the sustainability risk of these transition scenarios from the perspective of physical EROI. The overall modelling framework improved on the existing shortcomings of corresponding EROI studies and enhances the representativeness of the estimated physical EROI values by implementing a holistic approach for estimating primary energy quality at electricity level[29], creating a broader cumulative energy demand (CED) database for technologies based on life cycle assessment (LCA) databases[28], and integrating EROI estimation with energy system model output[27]. Notably, the potential impacts of the electricity generation mix on EROI and significant factors contributing to these impacts are deeply analysed. Finally, this study also presents how EROI links to systemwide levelised cost of electricity (LCOE) and $CO_2$ emissions.

## Results

### An overview of nine global energy transition scenarios

The analysis is based on the scenarios aiming to reach a net-zero $CO_2$ power system. In terms of modelling methodology, the scenarios are divided into two groups: (i) optimisation of the Best Policy Scenario (BPS) and alternative faster ET scenarios based on BPS, and (ii) replication of reputational scenarios generated by the International Energy Agency (IEA), and Teske and the German Aerospace Centre (DLR).

Nine global ET scenarios form the ground of this net energy study. Out of nine transition scenarios, three (BPS-plus scenarios) aim at achieving zero $CO_2$ emissions before 2050, while four scenarios (two BPS and two Teske/DLR scenarios) target the same by 2050 and the remaining two IEA scenarios achieve the goal after 2050. Thus, the scenarios selection was motivated to get a diverse representation of climate change targets together with an associated change in the energy mix and system cost. Also, these representative scenarios provide a good technical benchmarking due to the inclusion of certain technologies in the energy mix, e.g., wave power, fossil carbon capture and storage (CCS), fuel cells, etc., which are not considered as prominent technologies in BPS scenarios, and due to variations in the energy mix because of different scenario definitions, financial assumptions, $CO_2$ targets, technology diversity and transition speed. For all scenarios (see Table 1), similar techno-economic assumptions are run in the same environment. Conducting ways to reach the same purpose of these scenarios are differentiated from the point of technology selection, system composition over years, and finally phase-out of fossil-fuelled and nuclear power plants. Substantial increments in RE technology capacity is aimed in BPS scenarios with no addition of nuclear power and fossil fuel power plants, except gas-based technology due to fuel switch possibilities to biomethane and e-methane. Comparatively, the tendency to shift from coal-based power plants to nuclear power and gas power plants has been observed in IEA scenarios. Considering the contribution of nascent technology, such as fuel cell CHP, wave power converter, and fossil CCS, can be partially seen in Teske/DLR and IEA scenarios. The comprehensive information about the building process of the scenarios can be found in Aghahosseini et al.[6] and the SI.

Five scenarios of BPS[6,30,31] are differentiated according to the acceleration of the ET. Contrary to BPS No Wind Force (NWF), running on cost-optimisation without any constraint, BPS Wind Force (WF) scenario is built upon repowering wind power after reaching technical life duration, so the lower limit for each next period is equal to the total functional capacity of the previous period. Derived from the BPS scenario, BPS-plus scenarios (BPS-plus2030, BPS-plus2035, and BPS-

plus2040) are prepared to foresee variations in technology selection under different acceleration rates towards a net-zero $CO_2$ power system. As simulation scenarios, IEA's Stated Policy Scenario (STEPS)[32,33] is a business-as-usual scenario, following the current status of the energy system and assuming that all current government policies and regulations will be going forward as planned. The second scenario of IEA is the Sustainable Development Scenario (SDS)[32,33], which targets a significant reduction in oil and coal consumption. However, it compensates for these losses by increasing shares of nuclear power, fossil-fuelled power plants with CCS units, and RE sources. Teske/DLR-2.0°C[34], declared as an ambitious scenario to reduce greenhouse gas emissions (GHG), considers the delays that might happen due to political and societal processes during the ET, since this scenario is energy efficiency and RE driven. Alternative to this scenario, Teske/DLR-1.5°C[34] is framed as a technical benchmark scenario having a lower $CO_2$ emission budget for the whole system and not considering any political and societal barriers. However, the transition toward a net-zero $CO_2$ emission power system is quicker, so the deployment of RE and the integration of new technologies are faster than in the 2°C scenario.

All scenarios (described in Table 1) are run from 2015 to 2050 with 5-year time intervals for the nine major regions. The global-level power sector model presents an aggregated version of all nine major regions.

In BPS and BPS-plus scenarios, rapid decarbonisation is mainly carried out by balancing solar photovoltaics (PV) and wind power technologies (see Fig. 1), which require additional storage capacity and management of the curtailment. The PV systems growth in the scenarios shows a linear trend due to its low costs and stays in the range of 19.7-26.3 TW in 2050 compared to 0.6 TW in 2020. More explicitly, 58-79% of total electricity is met by the generation from PV systems, and the remaining part is first fulfilled by the generation from wind power, and then by other technologies. In terms of changes in wind power, repowering shows a substantial impact on electricity generation. The repowering partly limits the PV systems' electricity generation shares. The BPS-NWF and BPS-WF scenarios variation of PV systems electricity generation shares in 2050 is estimated to be 20%, which is shifted to the wind share solely for this reason. As expected, the total electricity generation from fossil-fuelled power plants is swiftly diminished, and biomethane and e-methane production is promoted instead of fossil gas use, moving to net-zero $CO_2$ emissions.

The main feature of IEA scenarios is aiming for a low-carbon pathway of power systems by replacing coal by nuclear power and gas with/without integrating CCS technology. While the nuclear electricity generation share is slightly increased from 9.4% in 2020 to 10.9% in 2050 for the SDS, it drops to 8% in the STEPS, illustrating the difference of the two scenarios. On the other hand, the electricity generation share of gas-based technologies is estimated at 11.4% in 2020. While this value diminished to 6.7% in 2050 by promoting renewables and diversification in technologies in the SDS, oppositely, it is increased to 17.6% in the STEPS. The integration of CCS to gas-based power plants in the SDS only covers a 1.4% electricity generation share in 2050. Finally, the sum of total electricity generation shares by PV and wind power plants are estimated as 53% and 33.2% for the SDS and the STEPS scenarios respectively. They are relatively low compared to BPS and BPS-plus scenarios.

In Teske/DLR scenarios, even though solar PV and wind power are still primary technologies at the core withholding proximate to a 72% share of total electricity generation, the prioritisation of technology types differs. The contribution of geothermal and CSP ST power plants reaches 6.5% and 15.7%, respectively shares of total electricity generation in 2050 in turn, is significantly higher compared to other scenarios. Wave power technology holds a just over 2% electricity generation share at the end of 2050, which is the highest value among all scenarios. The phase-out of nuclear power and fossil-fuelled power plants by 2050, as expected, results in shifting their big portion in the

**Table 1 | Energy transition scenarios processed with the LUT Energy System Transition Model (LUT-ESTM) for the power sector[6]**

| ET scenarios | | Main goals | Key Scenario Assumptions |
|---|---|---|---|
| LUT | BPS-NWF | Achieving a zero $CO_2$ emissions target by 2050 while minimising total annual system costs. No repowering of once-installed wind power. | Deployment of RE based on their techno-economic energy potentials. Ensuring phase-out of the installed fossil-fuelled power plants after completing their lifetime, and no addition of new capacity by 2050. Continuing to use the available nuclear sources until completion of their lifetime, and no addition of new capacity by 2050. e-Methane production by power-to-gas processes, and bio-methane. Integration of utility and prosumer-scale battery and PV systems. Introduction of storage systems (pumped hydro energy storage (PHES), adiabatic compressed air energy storage (A-CAES), thermal energy storage (TES), gas and biogas storage, and hydrogen storage). No integration of the CCS units into fossil fuel and gas power plants. |
| | BPS-WF | Achieving a zero $CO_2$ emissions target by 2050 while minimising total annual system costs. Repowering of once-installed wind power. | |
| | BPS-plus2030 | Forcing the completion of the ET by 2030, while minimising total annual system costs. | Deployment of RE based on their techno-economic energy potentials. Repowering of once-installed wind power. Ensuring phase-out of the installed fossil-fuelled power plants after completing their lifetime, and no addition of new capacity by 2050. Continuing to use the available nuclear sources until completion of their lifetime, and no addition of new capacity by 2050. e-Methane production by power-to-gas processes, and bio-methane. Integration of utility and prosumer-scale battery and PV systems. Introduction of storage systems (PHES, A-CAES, TES, gas and biogas storage, and hydrogen storage). No integration of the CCS units into fossil fuel and gas power plants. |
| | BPS-plus2035 | Completes the ET by 2035 while minimising total annual system costs. | |
| | BPS-plus2040 | Completes the ET by 2040 while minimising total annual system costs. | |
| IEA | SDS | Achieve the Sustainable Development Goals related to clean energy and the commitments stated in the Paris Agreement. Reducing 90% of the power sector $CO_2$ emissions by 2050 compared to 2020 levels (estimated). | Deployment of RE based on their techno-economic energy potentials and energy diversity. Rapid deployment of RE technologies. Annually 12% increase in installed solar capacity. Enhancing solar and wind electricity generation by 30% by 2030. Remarkable reduction in consumption of coal and oil as a fuel. Lower utilisation of natural gas methane compared to the STEPS scenario. A new addition to the total installed nuclear power capacity by 2050 (25% more installed capacity compared to the STEPS scenario in 2040). Integration of only utility-scale batteries to PV systems. Introduction of storage systems (PHES and TES). A major contribution of gas and coal power plants with CCS units. |
| | STEPS | Presentation of the current and planned energy policies declared by the governments. Cutting 15% of $CO_2$ emissions in the power sector by 2050 (estimated). | Deployment of RE based on their techno-economic energy potentials and energy diversity. Stable deployment growth of solar and wind electricity generation despite the decrease in cost and supportive policy mechanisms. Slow decrease in installed capacity of fossil-fuelled power plants. Gradually replacing coal and oil utilisation with natural gas. A new addition to the total installed nuclear power capacity by 2050. Integration of only utility-scale batteries to PV systems. Introduction of storage systems (PHES and TES). Inconsiderable contribution of gas and coal power plants with CCS units. |
| Teske/DLR | 1.5 °C | Targeting zero $CO_2$ emissions in 2050 not taking into account political, economic, and societal barriers. | Deployment of RE based on their techno-economic energy potentials and energy diversity. 65% of wind and solar electricity generation due to the addition of CSP integration with TES. No addition of new capacity in fossil-fuelled power plants by 2050 due to the carbon budget limitation. No addition of new capacity, continuing to use the available nuclear power sources until completion of their lifetime. Hydrogen usage Integration of only utility-scale batteries to PV systems. Introduction of storage systems (PHES, TES, and hydrogen storage). |
| | 2.0 °C | Targeting zero $CO_2$ emissions in 2050 while permitting delays triggered by political, economic, and societal barriers. | |

Note that the modelling results of IEA and Teske/DLR scenarios are re-simulated in LUT-ESTM. The main results (the electricity generation, installed capacity, share of renewables, energy storage technologies, role of prosumers, electricity demand, and $CO_2$ price) come from these studies. However, the load profiles, renewable resource profiles and technical and financial assumptions are mostly similar to LUT-ESTM due to the limit of availability. The presented LUT scenarios aim to achieve the decarbonisation of the power sector by 2050 or before, so it is not designed to model shared socioeconomic pathways and climate-related scenarios considering socioeconomic context.

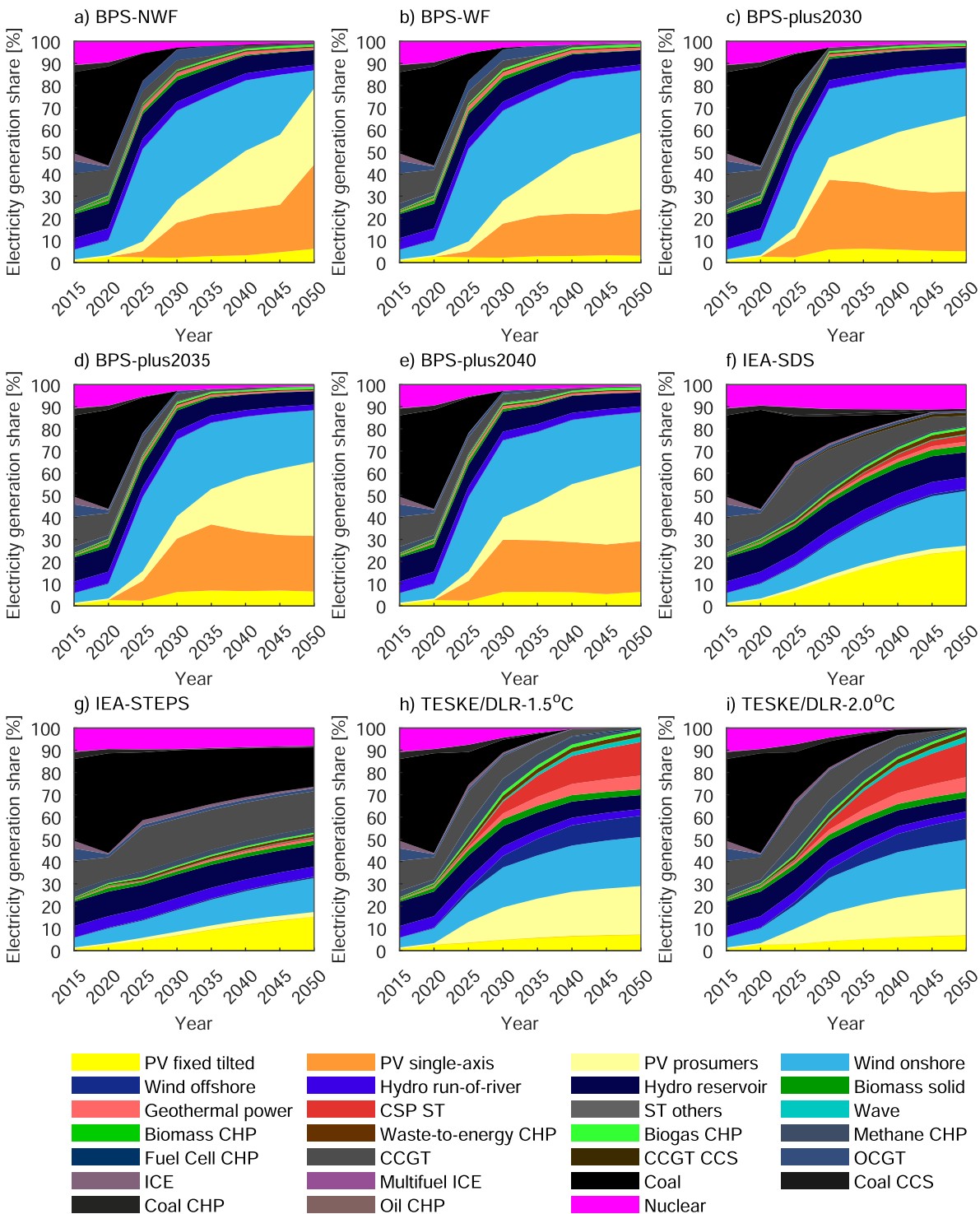

**Fig. 1 | Energy mix of all scenarios.** The panels **a**–**e** show all Best Policy Scenarios (BPS), which are modification of the standard LUT-BPS ET scenario. BPS No Wind Force (NWF) scenario is based on the techno-economic feasibility of global power systems grouped into nine major regions, and repowering of wind power plants is ensured in the BPS Wind Force (WF) scenario. In addition to these main scenarios that target 100% RE by 2050, three BPS-plus scenarios are derived by targeting 100% RE by earlier years, specifically 2030, 2035 and 2040. Additionally, the simulation scenarios of International Energy Agency's (IEA) Sustainable Development Scenario (SDS) and Stated Policy Scenario (STEPS), and Teske and the German Aerospace Centre (TESKE/DLR), given in the panels **f**–**i**, are replicated in the same environment as optimisation scenarios. The abbreviations are °C Celsius, CCGT combined cycle gas turbines, CCS carbon capture and sequestration, CHP combined heat and power, CSP concentrating solar thermal power, ICE internal combustion engine, OCGT open cycle gas turbines, PP power plant, ST steam turbines, and PV photovoltaic.

total electricity generation to solar PV, wind power, geothermal and CSP ST, but a small portion of it diversified among CHP technologies, mainly waste-to-energy, biogas, and fuel cell.

The path to a net-zero $CO_2$ emission power system requires adequate financing sources, the lack of which may delay the ET. Although this is the front facet, these rising costs can be offset by net savings from decommissioning fossil fuel-based systems, and the faster the transition proceeds, the greater the overall net savings are obtained, as reported by Way et al.[35], which overlaps the results of Aghahosseini et al.[6]. The modelling results reveal that the LCOE for all scenarios

starts at 70.9 €/MWh in 2015 and gradually decreases as the RE deployment increases. The estimated LCOE remains in the range for 45.2-49.7 €/MWh for the BPS scenarios, 53.9-54.1 €/MWh for the Teske/DLR scenarios, and 59.2-69.5 €/MWh for the IEA scenarios[6]. LCOE of all scenarios peaks in the period 2020-2030 as the shifting from fossil fuel systems to renewable systems accelerates and $CO_2$ cost for emissions of the fossil-fuelled power plants is preluded after 2020, which brings additional burden on overall costs. The proportion of $CO_2$ costs varies between 10% and 20% depending on the scenario and the type of fuel sources prioritised. After 2030, the ET effect becomes more evident in the LCOE trends. The IEA-STEPS displays a stable trend until 2040 and then shows a slight decrease after this year. The largest decrease is seen in BPS scenarios moving towards 100% RE systems. The fuel costs and $CO_2$ costs are replaced by the costs of storage technologies and even the integration of larger quantities of storage does not exceed the sum of these costs, which remains between 10% and 30% of the system LCOE. The IEA-SDS and Teske/DLR scenarios follow the same bearish trend but at a slower pace. The underlying reasons are the deployment of gas and nuclear power technologies in the IEA-SDS scenario and the integration of nascent technologies in Teske/DLR scenarios, with the efficiency of the nascent technologies and the amount of electricity generated having an impact on this pace. These scenarios have a limited inclusion of storage technologies. As total net savings, the cumulative pathway cost of the IEA-STEPS scenario is proximate to 90,000 billion euros (b€) by 2050, with the IEA-SDS and Teske/DLR scenario slightly exceeding this threshold. On the other hand, the BPS scenarios remain in the range of 60,000-70,000 b€ for the same year and have a negligible difference among them. Even rough estimates show that 20,000 b€ can be saved in systems dominated by solar and wind technologies. Thus, the energy mix and the choice of technology types are one of the key factors determining the cost of sustainable power systems.

## Perspectives of systemwide EROI

EROI analysis traditionally accounts for the energy flowing within several processes of product manufacturing and fuel production as well as the use of those outputs for electricity generation and supply. Systemwide EROI deals with even more complex processes, as it involves the consideration of actual technological use and its interoperability for the reliable supply of electricity, the criteria, and rules of which may continually change with time during the ET. Additional issues correspond to setting appropriate boundary conditions for all involved technologies and estimating all primary energy at the same energy quality. This study overcomes related issues by applying a newly constructed LUT-EROI model, which estimates all types of CED data following LCA rules as detailed in Methods and the Supplementary Information while also maintaining the same primary energy quality as recommended in Solomon et al.[29]. The model applies a mathematical algorithm that allows capturing the change in operation rules of each technology implemented in the LUT Energy System Transition Model (LUT-ESTM)[30,31]. LUT-ESTM is ranked as a leading cost-optimisation energy system model[36] and one of the two most used tools for highly RE system analyses[26]. Dissimilar to Integrated Assessment Models (IAMs), LUT-ESTM is designed for analysing short- and medium-term goals to achieve 100% net-zero $CO_2$ power systems, thus, the software tool architecture is structured around this scope rather than targeting the long-term climate goals, as these can be set as constraints. It is a multi-scale energy modelling tool that allows flexible implementation at national, regional, and global scales[31]. As with all cost-based energy modelling tools, techno-economic parameters are the underpinnings of the structure. Using reliable and widely accepted references and iterating micro- and macro-scale energy systems in different studies is one way to overcome this disadvantage[26,37]. LUT-ESTM uses updated and internationally recognised references clarified in Aghahosseini et al.[6] and Bogdanov et al.[30].

The LUT-EROI not only improves on the previous key short-comings of EROI estimation but also achieves sound comparability with different system types. This study relies on nine global ET scenarios reported in Aghahosseini et al.[6]. The nine transition scenarios are modelled by dividing the world into nine major regions. These scenarios are five LUT-BPS, two scenarios published by the IEA, and two scenarios published by Teske/DLR. Even though the choice of these scenarios were primarily motivated by the availability of the required detailed data for the EROI calculation, they also provide a representation of the variety of discussed transition paths.

The detail study of the corresponding systemwide EROI is presented below and further information on each scenario is provided in Methods. The systemwide EROI was estimated at the point of electricity generation and consumption, with estimates at the point of final energy consumption leading to lower EROI. Due to similarities in trends in this result, we present the detail using point of final consumption (F) and provide the point of generation (G) estimation, not including losses from electricity transmission and distribution, in Supplementary Information Note 4.

## Impacts of the energy transition on global EROI

Figure 2 illustrates the global EROI results for nine main scenarios. During the 30-year ET period, global EROI values were shown to remain above 16, maintaining a value above 10, the upper limit for the net energy cliff[22,38]. The fluctuations in the EROI trends at specific periods are largely driven by dramatic changes in system composition and the prioritisation order of technology types, in other words, differences in ET pathways.

From 2015 to 2020, Fig. 2 provides a summary of the historical situation of global power systems. The EROIs for all scenarios start from 18.8 and increase to above 20. The growing trends are observed because of the slow integration of renewables into modern power systems, which reduce fossil fuels use and thus improve EROI, while further enabling technologies are not yet required.

After 2020, three major EROI trends emerge depending on the concrete ET pathway (Fig. 2a, b). The five LUT-BPS scenarios form the first group of trends in which EROI continues its increasing trend until 2025, and then shifts its trend to a continual decline up to 2050. The continued EROI increase by around 18% through 2025 is due to the replacement of fossil-fuelled power plants with renewables. At this point, variable renewable energy (VRE) has gone up fivefold (Fig. 2c) and the gross electricity generation from solar and wind technologies go beyond 50%. The decline after 2025 (Fig. 2a) is associated with the expansion both of solar PV and wind power capacities and of the enabling technologies, mainly batteries and gas storage towards the end of the ET period. The upsurge of renewables and emerging technologies' installed capacities brings an additional burden to total energy invested (EI), leading to a decline in EROI. However, the rate of decline changes considerably with scenarios. Three scenarios, namely BPS-NWF, BPS-WF, and BPS-plus2040 show a gradual decline with small differences between their trends. The differences can be credited to changes in the energy mix, which can be seen from the differences of BPS-NWF and BPS-WF, and the difference in storage build-up, especially as observed in the last period. On the other hand, the BPS-plus2030 and BPS-plus2035 scenarios show a sharp decline between the 2025–30 and 2030–35 periods, respectively. As the transition in these BPS-plus scenarios is ambitiously carried out in the specified periods, EROIs of these scenarios decline more slowly in the later periods, showing a sign of EROI stabilisation after achieving 100% RE. The underlying reason for the sharp decline is the very fast-growing capacity need for solar PV systems and their complementary technologies. Notably, the upsurge of water electrolysis and methanation capacities, the key technologies for gas storage, accelerates this declining trend substantially in the respective periods.

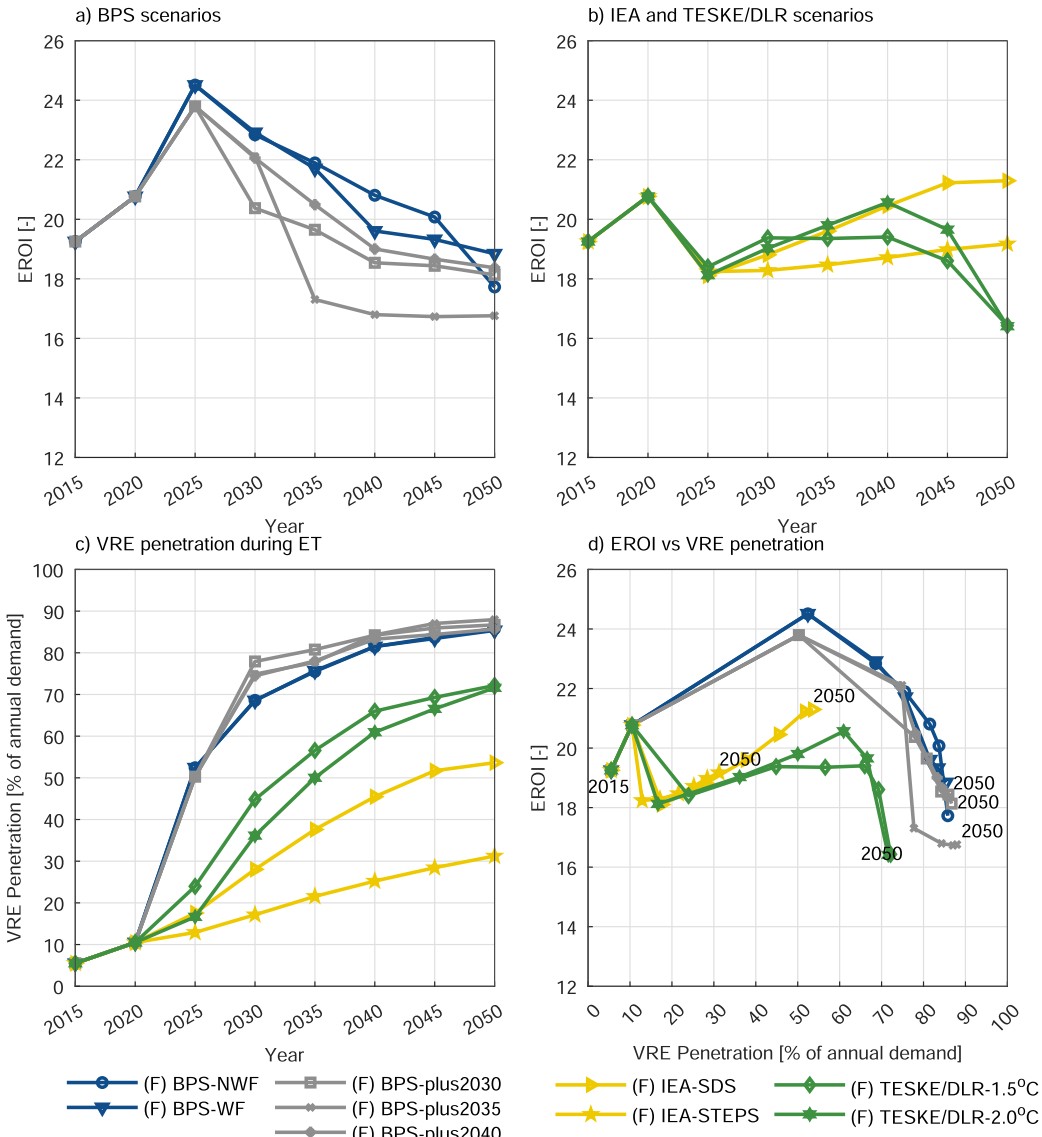

**Fig. 2 | Global systemwide EROI trends. a** BPS scenarios, **b** IEA and Teske/DLR scenarios during the energy transition (ET) period. Changes in variable renewable energy (VRE) penetration and EROI versus (vs) VRE penetration are shown in **c** and **d**. VRE penetration values only include solar photovoltaic and wind power technologies. (F) represents the final consumption.

The two IEA scenarios form the second group of EROI trends (Fig. 2b). Here, EROI drops from nearly 20 to approximately 18 in the 2020-25 period but rebounds slightly in the following periods to finalise at an EROI value above 19 by 2050. The rate of VRE share increase in these scenarios remains very low as compared to the BPS scenarios due to the applied modelling constraints, and as a result it could not achieve a comparable increase in EROI (Fig. 2c, d). The rebounding effects of both IEA scenarios were due to the phase-out of coal-fired power plants and reduction in gas consumption, which was replaced by RE and nuclear power, with the latter ranging with the EROI of coal-based and gas-based electricity. Teske/DLR scenarios form the third group of EROI trends, which show similar characteristics to the IEA scenarios until 2040 when they start their declining trends. The trend up to 2040 is due to the change in fossil fuel use, which was replaced by RE. EROI decrease beyond 2040 in Teske/DLR scenarios are mainly affected by even the slightest capacity expansion of geothermal and concentrated solar thermal power with steam turbine (CSP ST) power plants. Especially for geothermal, its CED value remains very high compared to other technologies due to the low electric efficiency and the involvement of drilling and injection

processes referred to as high energy-intensive processes. Conversely, the interval between EROIs of Teske/DLR scenarios seems to appear due to minor differences. In Teske/DLR 1.5°C, limiting the $CO_2$ emissions budget without considering any delays in political and societal processes forced the system to produce more electricity from renewable sources faster. Such a dramatic change is not seen in the Teske/DLR 2.0°C scenario, where electricity generation from coal is shifted to gas before finally being replaced by RE around the end of the ET. The nascent technologies, such as wave power converter and fuel cell combined heat and power (CHP), hold insignificant impacts on EROI for the time being. Evidently, storage technologies that are a part of power generators (e.g., TES, hydropower with reservoir etc.) support the increase in system EROIs, conversely complementary storage technologies (such as batteries, hydrogen, and methane storages) that are integrated into electricity generators lead to a substantial drop in overall EROIs as correspond to the findings of Diesendorf and Wiedmann[21]. Note that global EROIs present the average results for all regions due to the usage of aggregated data during ET modelling. The results belonging to regions are provided in Supplementary Information Note 4.

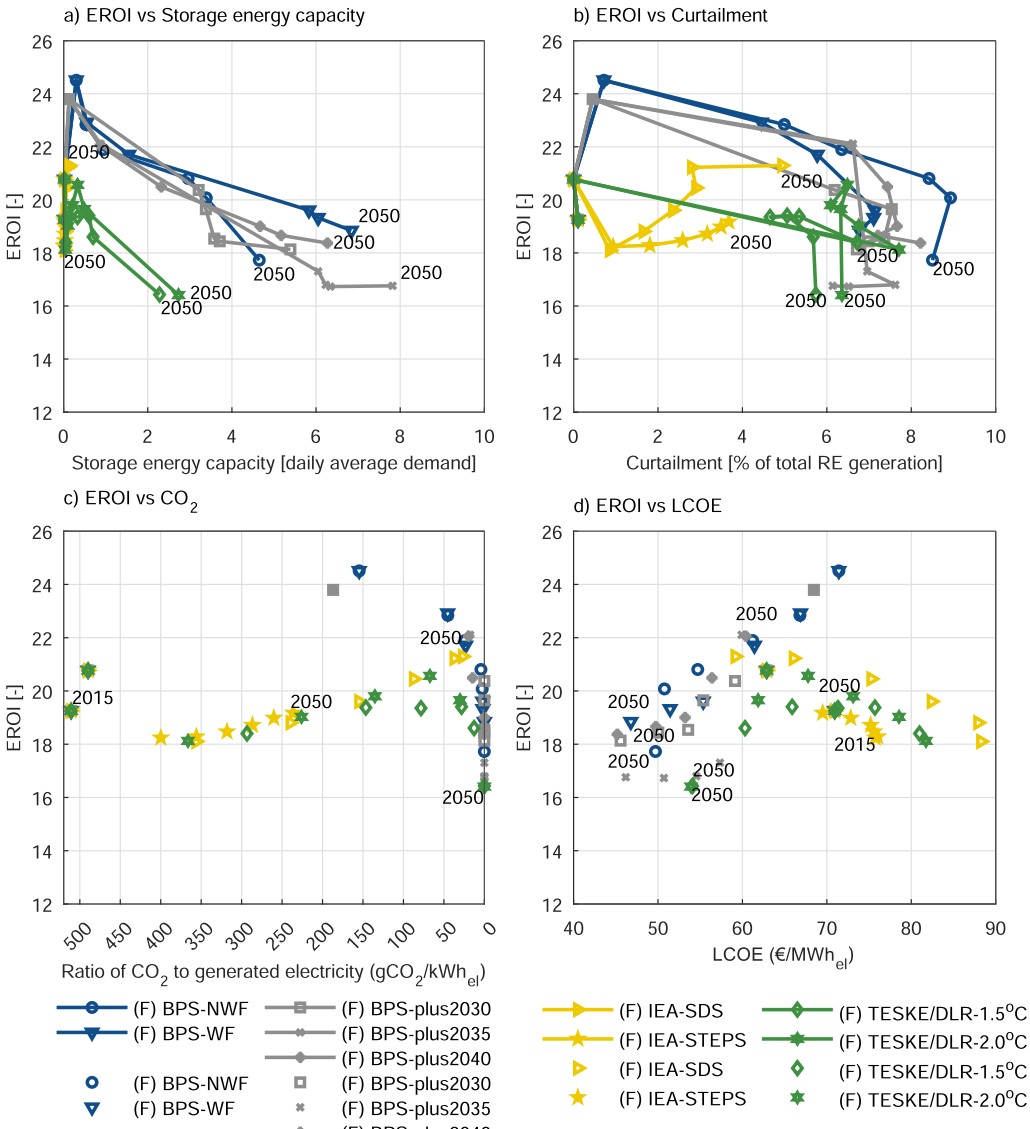

**Fig. 3 | The relative trends of EROIs.** EROI trends relationships with **a** storage energy capacity, **b** curtailment, **c** the ratio of $CO_2$ to generated electricity and **d** levelised cost of electricity (LCOE) versus (vs) year. Note that the represented variable renewable energy penetration values only include solar photovoltaic and wind power technologies. (F) represents the final consumption. BPS scenarios are shown in blue and grey lines and dots, IEA scenarios in yellow and Teske/DLR scenarios in green lines and dots.

Revisiting the foregoing discussion of EROI trends by combining Fig. 2a–d reveals that, despite scenario-dependent differences, all trends can be linked to the corresponding VRE penetration. The next section explains the reasons of the above trends and the dependence of EROI on various factors in more detail (Fig. 3). Further analysis of the EROI relation with system LCOE (Fig. 3d) shows that low-cost solutions correlate to low EROI. Replacing fossil and nuclear power with RE reduces the system's LCOE by lowering fuel and $CO_2$ costs. The EI for the upstream processes of the fuel chain initially account for a significant percentage of the total system EI. As the transition progresses, this share is replaced by the EI of the RE technologies and their enabling systems, and the addition of enabling technologies leads to a further decrease in EROI values. The level of reduction in the LCOE and EROI is related to the key decisive factors, which are system composition, the type of technology, the costs associated with the technologies, the cost of capital, the capacity utilisation, and the required energy investments. The observed trend may change depending on these decisive factors, and the interpretation here is presented by a relative trend analysis of two different indicators. This conclusion may change in future depending on the system types (predominantly fossil-nuclear powered or highly RE with enabling technologies), as the investigation of system level ET studies increases, and thus, the complex nature of this relationship can be better elucidated.

On the other hand, low EROI was found to be one of the cleanest options (Fig. 3c, d), while systems with high EROI are usually the ones with high $CO_2$ emissions as observed from IEA scenarios, which are not achieving net-zero $CO_2$ power systems by 2050. Thus, a multicriteria system designing will provide a better means of defining the proper path. The presence of high uncertainty due to data aggregation[39], other limitations and sensitivity covered in Solomon et al.[27] are the additional reasons justifying the need of multicriteria future system designing.

Low-carbon pathways generated by IAMs differ from cost-optimisation energy modelling in terms of methodology, and conceptualisation of the scenarios, and could yield different EROI outcomes. Together with the divergence of our EROI methodology, developed by Solomon et al.[27,29], a careful comparison is necessary. The typical results of our study are consistent with the findings of King

et al.[19] that applied IAMs to investigate EROI and indicated a decrease in net energy for society. The technology level EROIs for solar and wind technologies are increasing over time while the reverse situation is observed for fossil fuelled power plants[19]. Gas and nuclear power plants mostly maintain their low EROI values compared to renewables or show minor change over years. This is also founded in the study by Sers[24] according to which the EROI of fossil fuels shows a precipitous drop by 2050 due to the depletion of reserves. Additionally, the renewable EROI has been set at 40 and increases as the cumulative greenhouse gas emissions decrease[24]. However, in this study, the EROI of renewable technologies varies depending on operating patterns, energy learning rate, and the storage technologies reduce the systemwide EROI, in contrast to the study of Sers[24], where storage EROI is estimated separately from renewable EROI. Another study presented by Slameršak et al.[25] notes that the early years of the transition could result in an increase in GHG emissions due to more fossil fuel usage. This partially overlaps with our findings in an indirect way. By shortening the ET period, the systemwide EROI will be drastically reduced due to the deployment of renewable power plants and complementary technologies. Our study does not apply energy flow techniques as in the case of PROI, but the data suggests that construction phase energy consumption may present a challenge for those scenarios. We leave such matters to further studies. Note that IAMs studies discussed here aim to model the entire energy system, rather than just focusing on the power sector, and follow a broader energy modelling framework, including socio-economic relationships, indirect GHG estimates and energy efficiency implications, that differs from the study of Aghahosseini et al.[6] as regards to the contextualisation of scenarios and modelling approach. Further information on the key differences in IAMs implemented in EROI studies is explicitly discussed in a recent study by Delannoy et al.[40].

Note that the five BPS scenarios created with a slightly higher demand projection that reaches 48.38 PWh in 2050 as compared to the IEA and Teske/DLR scenarios, which remain in the range of 45-46.5 PWh. However, no clear evidence was found that demand influenced the EROI trends such as BPS-WF scenarios show a slower decline compared to Teske/DLR scenarios. Further in-depth examination of the data also reveals that the system composition and the technology choice have more impact on the trend.

Furthermore, the sustainability risk of the systemwide EROI is reanalysed by the annual energy investment flows (AEF) for each scenario (Fig. 4). The profound analysis signifies that none of the scenarios has exceeded 16% of the final energy consumption of the respective year (Supplementary Fig. 20) while the upper limit for the systemwide energy investment flow is estimated as 7% (Supplementary Fig. 19). In both cases, the energy need for the upstream supply chain of fuel production (excluding energy content of fuels) approaches zero during the transition as this is gradually substituted by the energy necessity of RE investments in the defossilisation scenarios. However, this shift occurs more slowly in the IEA scenarios because of the lower share of RE and continued utilisation of fossil and nuclear power plants (Fig. 4c). Also, the shortening of the ET triggers the need for more energy for RE investment in the BPS-plus scenarios (Fig. 4b), as observed in the annual energy investment flows (Supplementary Fig. 20). Considering the CED in the year of the investment creates greater fluctuations in the AEF during the ET (Fig. 4a–d). Despite a high expansion of RE, the estimated annual energy return on investment values do not fall below 5. Further details are presented in the Supplementary Information Note 6.

## Effects of PV−storage−curtailment paradigm on the global EROI trends
A closer examination of the factors contributing to the observed EROI trend shows VRE penetration, storage capacity, and total loss (representing curtailment plus storage loss) plays a decisive role in defining

the trend together with other limitation coming with diversity for some scenarios. The trend of the BPS scenarios (Fig. 2d) shows that EROI initially increases together with VRE penetration until VRE penetration passes 50%. EROI then starts declining, with the decline getting even sharper as VRE penetration approaches 80%. The initial increasing range is a range where VRE penetration requires lower storage capacity (Fig. 3a) and total loss. Thus, the VRE increase can compensate the added burden of the EI. However, as VRE penetration increases further, the increase in added burden of the enabling technologies forces EROI to the declining trend, which gets worse as the use of seasonal storage increases at high VRE penetration (Fig. 2d, e). The overall trend reported above is similar to the one in Solomon et al.[27], where detail evidence of the relation of EROI trend with the VRE-penetration-curtailment-storage nexus[41,42] is presented.

However, significant system differences produce curve variations depending on the system composition. This global case is highly diverse, which depends on the mix of wind power and solar PV plus other renewables. A separate analysis shows that spatial aggregation generates a modified VRE-storage-curtailment relationship due to smoothing effects and lead to an enhancement of the EROI, especially at higher VRE penetration[39]. This may partly explain the clear difference between the scenarios with higher and lower VRE penetration (see Figs. 2c, d, 3a, b, and 1c–g). Note that this trend represents a global average and thus obscure some local issues.

IEA and Teske/DLR scenarios finish the transition at a lower VRE penetration (approximately 70% for Teske/DLR, 54% for IEA-SDS, and 31% for IEA-STEPS scenarios), which correspond to EROI values of above 23 in the BPS scenarios (Fig. 2c, d) as compared to the lower EROI values in these scenarios. This explains that EROI in IEA scenarios rebounded depending on VRE penetration, but are still not at the achievable potential (seen in BPS scenarios) because of the inclusion of nuclear power, which achieves a lower EROI performance compared to wind power and solar PV. The trend of Teske/DLR scenarios may have also occurred due to factors other than the burden coming from high geothermal CED and its low thermal efficiency as this may also be seen from a comparatively higher total loss (Fig. 3a and Supplementary Fig. 9c) and storage capacity (Fig. 3a and Supplementary Fig. 9a). This suggests that Teske/DLR scenarios may improve the EROI values with further optimisation of the system design and operation even with the same resource mix.

## Sensitivity of the global EROIs to the embodied energy requirements for natural gas and oil sources
The sensitivity of non-renewable EROIs to the fuel-embodied energy requirements is a discussion that emerges in EROI studies due to its ability to change their EROI trends[17,43–45]. The core of this sensitivity analysis relates to the natural gas and oil consumption during the ET. Thus, the sensitivity scenarios (Fig. 5) are based on published fuel EROI trend curves in Delannoy et al.[44] (DF) and Sgouridis et al.[43] (SF). The corresponding deviation in EI trends create a significant difference, particularly in EROIs of IEA scenarios since natural gas is prioritised as a replacement for coal (Fig. 5c). Meanwhile, a declining EROI trend of the IEA-STEPS scenario is worsened by almost 6 points, whereas this reduction is around 3 points for the IEA-SDS DF scenarios as compared to the default SF scenarios. This clearly indicates that IEA scenarios are extremely sensitive to smallest changes in EROIs of natural gas and oil.

BPS and BPS-plus scenarios are not affected by these changes because their system design consists mostly of RE technologies (Fig. 5a, b). In comparison, Teske/DLR scenarios (Fig. 5d) still demonstrate small changes. According to the estimations, the decrease in EROIs for both scenarios remain around 1 point during the transition period.

Further study is required to identify the sensitiveness of EROIs to the changes in coal and nuclear fuel CEDs during the transition. At the

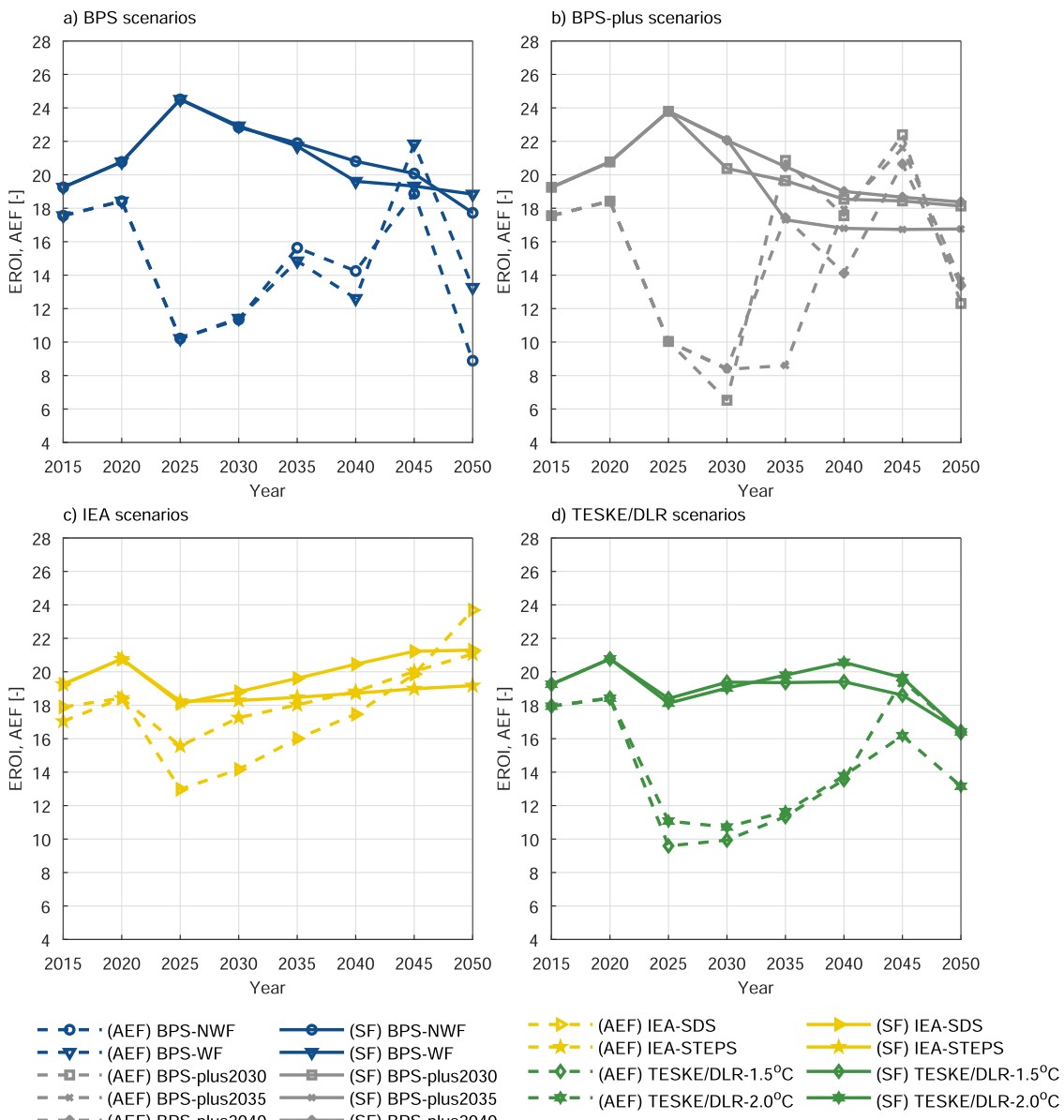

**Fig. 4 | Systemwide EROI and annual energy investment flow estimates (AEF) for the global power system. a** BPS scenarios, **b** BPS-plus scenarios, **c** IEA scenarios, and **d** Teske/DLR scenarios. The cut-off point for (F) EROI is the net electricity delivered to end-users (including transmission and distribution losses). (AEF) denotes the annual energy investment flow (dash lines), while (SF) (for Sgouridis et al.[43] final net electricity delivering to end-users) represents the systemwide energy investment flow (solid lines).

same time, the sensitivities to several other factors can be inferred from Solomon et al.[27].

## Significant variations in EROIs due to regional differences

Global averages mask the regional heterogeneity of EROI trends. The EROI trends for all scenarios vary with location due to the uneven distribution of natural resources. In BPS and BPS-plus scenarios, most of the regions' generation portfolio is dominated by solar and wind resources. As described in the preceding section, rapid solar PV and wind power capacity expansion, along with their enabling technologies, leads to declining EROI at higher VRE penetrations, a situation commonly observed after 2030 dominantly in some regions as given in Supplementary Information Note 4, Supplementary Fig. 10-18 (e.g., Eurasia, Northeast Asia (NE-Asia), sub-Saharan Africa (SSA), and North America (N-AM) regions).

Targeting an earlier achievement of 100% RE as in BPS-plus scenarios usually ends with a precipitous drop in EROIs, except for

Europe, Southeast Asia (SE-Asia), and South America (S-AM) regions where the observed decrease in the trend is relatively less pronounced compared to other regions, particularly for the BPS-plus2035 scenario. This is because of the low requirement for battery and gas storage capacities caused by resource diversity of these regions, as compared to other regions. Specifically, as is the case the S-AM region, possessing great hydropower sources and utilisation of all its techno-economic potential combined with a solar-wind dominated power system is escalating the EROIs above 26. Conversely, the regional results of IEA scenarios provide a different insight compared to the BPS-plus scenarios. In general, the SDS scenario generates higher EROI values due to the restriction of fossil fuel use; however, this situation is reversed for Eurasia, NE-Asia, and SAARC regions. Even though coal consumption for electricity generation is substantially reduced, filling this gap via solar PV systems accompanied by natural gas and nuclear power led to higher EI, consequently leading to a decrease in EROIs of the SDS scenario. Peculiarly, in such diversified systems, any slight increase in

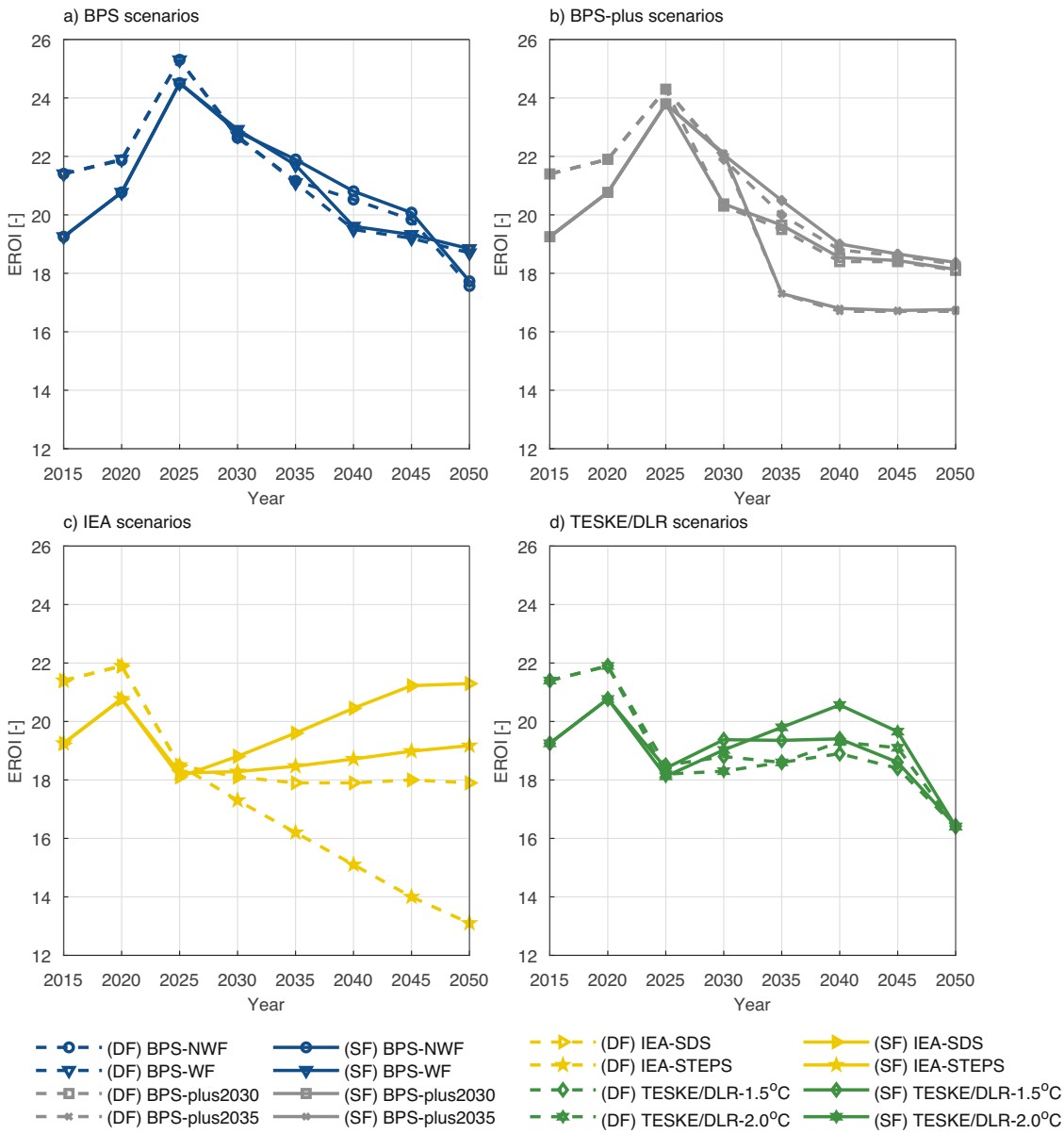

**Fig. 5 | Sensitivity analysis of global (F) EROI trends in case of any changes in natural gas and oil fuel CED values. a** BPS scenarios, **b** BPS-plus scenarios, **c** IEA scenarios, and **d** Teske/DLR scenarios. Note that the presented sensitivity analysis cut-off point is selected as net electricity to end-users (F), and variations in natural gas and oil fuel CED values are presented based on Delannoy et al.[44] (DF) as dash lines and Sgouridis et al.[43] (SF), presented as solid lines. (SF) is chosen as the default scenario in the analysis.

the capacity of technologies holding higher CED values might pull the EROI into lower values and/or change the trend completely. Similar cases occurred in Eurasia and NE-Asia regions. For both regions, small additions to the EI percentage of geothermal technology contributed to altering the trends since they have comparatively higher EI values than other renewables. The findings on CCS technology are surprising. The integration of fossil fuel power plants with CCS units produces slightly higher CED results as expected. Their impacts on the system from the perspective of EROI remain negligible compared to fossil-fuelled power plants for IEA scenarios, which is compatible to the findings of Sgouridis et al.[18]. Exceptionally, this situation is different for the NE-Asia region. The partial shifting of coal and gas-based technologies generation to ones integrated with CCS resulted in an increment in EI as expected that causes a drop in EROI. It should not be forgotten that CED values of nascent technologies might decrease due to advances in material types and consumption rates.

The EROI trend of Teske/DLR scenarios typically moves towards lower EROI values. Only one region, MENA, shows an opposite trend where EROI values step up from almost 12 to above 17. This situation is related to EI balances between fossil-fuelled and RE technologies. Higher EI occurring because of the introduction of CSP ST power plants and the enlargement of geothermal power plant capacities somehow counter-balanced by the sudden reduction in gas and oil consumption. More crucially, the capacity increment in battery and gas storage is restricted. When this restriction is lifted due to the enlargement of PV and storage needs, the trend path is turning downward, especially for the Teske/DLR-2.0°C scenario. This coincides with the global-level EROI results.

## Discussion
Deciding on how to select an optimal power system composition is as vital as predicting under what conditions the transition to net-zero $CO_2$

power systems will occur. This study contextualises this question from the perspective of EROI for ET scenarios. The findings on EROI clarify that the replacement of fossil fuel technologies with RE results in lower EROI values supporting the arguments of Lovins and Price[46]. As solar PV and wind technologies dominate the energy mix, the numerical increase in battery and storage capacities triggers a further decrease in EROIs. The interactions of PV, storage, and curtailment elucidate that at a high VRE penetration level, seasonal storage services preponderate over diurnal storage, and even a small increment in VRE ends up in requiring more storage capacity than usual. Conversely, the effect of nascent technologies (e.g., wave power converters, fuel CHP) mostly stays insignificant due to their capacity amounts. Additionally, the results demonstrated that for all scenarios, none of the global EROIs goes below 10, which is a value considered to be the upper limit for the net energy cliff. This signifies that: (i) all scenarios are feasible from a techno-economic point of view, (ii) the diversification in RE technologies helps to smooth sudden EROI drops, (iii) restricting ET time leads to sharp declines in EROIs temporarily, but as time proceeds, EROIs are stabilised, and (iv) the storage dependency has an adverse impact on the EROIs trends. These important findings support the claims of technological optimism[47] that the transition to 100% RE systems does not result in a significant disruption from a physical EROI perspective. The impacts of high electrification and energy efficiency should be further investigated to anticipate the changes in systemwide EROI trends. Note that taking all sustainability aspects into account, a steep drop in the systemwide EROI might cause irreparable economic consequences[48–50].

Executing the radical changes on time is crucial for stabilising invested energy towards net-zero $CO_2$ power systems. Undoubtedly, any alterations in the system design will bring some profits and losses. The crucial point here is how to counterbalance them in the long term. Nevertheless, moving towards a low $CO_2$ power system without achieving net-zero $CO_2$ emissions can also be accomplished by the prioritisation of nuclear power and gas-based technologies as is the case in IEA scenarios. The sensitivity analysis shows that this situation creates a temporary illusion for the EROI results. As the system becomes more dependent on natural gas and nuclear power; concurrently, it is inevitable that the vulnerability that arises from limited fossil fuel and nuclear resources will be intensified in the future, and consequently, cause more dramatic falls in EROIs.

## Methods
### CED database for LUT-EROI
The arguments[16,28,51] related to the delimitation of CED analysis have been gathered around the uncertainties arising from the system boundary identification, incompatibility of the system model selection, disregarding interpretation of interactions in multi-process systems, standardisation of energy quality, and geographic variations. Expectedly, these disputable issues originating from the life cycle impact assessment (LCA) concept[52] cannot be fully eliminated, but can be minimised by the selection of the proper methodology, providing reliable data, and presenting transparency in the calculations. In light of these discussions, the following eight fundamental steps were established to constitute a reliable and transparent database for the LUT-EROI: (i) in-depth analysis of existing life cycle inventory (LCI) studies to identify the selection of proper system boundaries for LUT-ESTM technologies; (ii) comparing dissociated inventories based on each material type and amounts to foresee if they fit LUT-ESTM technologies; (iii) selecting the most appropriate studies eligible to represent LUT-ESTM technologies in terms of their commercialisation level and applicability on the market; (iv) collecting and extracting datasets from an acknowledged and reliable database, hereby, the ecoinvent database V.3.7.1;[53] (v) testing the compatibility of the selected datasets with LUT-ESTM technologies; (vi) calculating the CED value for each technology; (vii) updating the found CED values due to

global market trends and future improvements on a technology level; and finally (viii) comparing estimated CED values to the literature findings to approximate real cases.

The complexity and nesting levels of energy modelling tools differ depending on its purpose and the required level of accuracy in representing the real-case, so it is vital to adjust the selected inventories according to the model necessities and the operating conditions. LUT-ESTM is a multi-nodal approach energy modelling tool and operates based on the full hourly resolution for a year[30]. The system components for the power sector consist of various types of technologies, thus, the disentanglement of the system in a technology-level approach is inevitable. This can be achieved by understanding interactions between technologies, and then by selecting the right allocation model. For LUT-ESTM, allocation, cut-off by classification[54] is the most appropriate sub-system model. It turns multi-product activities into single ones without including recycling of materials and waste products because it requires specific ratios of the material types and precise information about the recycling process. At this point, cut-off point levels of system boundaries are decided based on the process's types, exchanges between flows and commodities, and the available resources (resources extraction and their processing), and recycling of the materials and wastes are not considered in this analysis. A detailed explanation of the criteria for the identification of system boundaries can be found in Supplementary Information Note 2.

Still, the uncertainty of estimating primary energy at the same energy quality remains a fundamental problem of CED and EROI studies. This issue was solved by applying energy conversion factors to estimate primary energy at the level of electricity quality as recommended in Solomon et al.[29]. The ecoinvent database provides CED values by separating them into eight sub-categories (renewables: biomass, geothermal, solar, potential-barrage water, kinetic-wind, and non-renewables: fossil, nuclear, and primary forest) according to the energy resource type as MJ-equivalents[54]. The same category logic presented by ecoinvent is applied for the determination of conversion factors, and the MJ-equivalent unit is converted to MJ-electricity on a technology level that is represented in Eq. (1). $MJ_{el}$ for each category ($c$) estimated by multiplying the primary energy ($MJ_{pe-eq}$) and the related electricity conversion factor ($f$). The conversion factors are explicitly given in Supplementary Information Note 2.

$$MJ_{el,c} = MJ_{pe-eq,c} \cdot f_c \qquad (1)$$

Beyond these, having been designed on the hourly load operating principle of LUT-ESTM requires dividing the estimated CED value as construction and decommissioning phases, and the operational phase. For this very reason, each inventory of technology is disaggregated according to the purpose of use and then calculated in two separate CED values. The first CED value belongs to the construction and decommissioning phases and is described in terms of capacity, $E_{capacity}$ (as $MJ_{el}$ per kW), and covers the structure materials, and other consumables needed for construction and decommissioning, such as chemical substances, electricity, and fuel consumption for transportation. The second CED value is for the operational phase of a technology, which addresses consumables, not main fuels, used for operating purposes in terms of electricity generation, i.e., $E_{operation}$ (as $MJ_{el}$ per kWh).

### Reflecting the technology evolution effect on CED values
Rapid advancements in RE technologies lead to declining costs over time and expedite the ET at a faster rate[30,55]. As the technology matures, necessary embodied energy for the technology will be affected[56] by the improvements in material types and innovations in processes. Thus, ongoing technology improvement on CED values holds a significant position in dynamic EROI analysis. Particularly, for PV and battery the interpretation of CED values by using average

energy learning rates (ELR) ensures to reflect their future technological improvements. For PV systems, Görig and Breyer[57] define an ELR based on CED and cumulative capacity instead of using the traditional financial learning curve concept. The ELR of PV systems is estimated at approximately 14%, which is taken as an initial value in this study and gradually decreased to estimate future CED values of the PV system by using Eq. 1 in Supplementary Information Note 3. As a complementary technology to PV, the battery composition, its chemistry and integration to the other systems[58] are changing over time due to recent technology improvements. Hsieh et al.[59] go beyond the conventional projections about the battery by developing a two-stage learning curve model for nickel-manganese-cobalt (NMC) battery linked to mineral and material costs. As a base case scenario, 7% of battery pack price will come from active material synthesis costs in 2015 that are foreseen to increase to 21% in 2030. With the support of leading-edge technology, the material synthesis ELR rate is expected to drop to 3.5%[59]. Setting an upper limit ELR for batteries from 5% in 2015 and moving towards the minimum ELR rate of 4% in 2050 might a justifiable approach considering the recent improvements in battery technology. Applying the same method for large-scale power systems, where integration of them to the existing power systems is taking a slow pace, would result in misled CED values because of the time dependency of ELR. Thus, future CED values for certain technologies, especially CSP ST plants[32,60–62] and nuclear power plants[63–66], are decided based on planned capacities and future market shares of technology types.

On the other hand, as a part of LUT-ESTM, Power-to-X (PtX) processes on the verge of commercialisation are a key to solving the variability of renewables-based systems[31]. The quality and quantity of synthetic methane gas, used for seasonal balancing, depend on especially the technology type selection of electrolyser, direct air capture (DAC) and methanation units. Projecting CED values for these nascent technologies is highly dependent on their maturity level and the targets set by the manufacturers and developers. Their present CED values are high, but CED values are expected to decrease over time in parallel to the development of advanced versions (see Supplementary Information Note 3).

## Net energy requirements for extraction, processing, and transportation of fuels

The approaches to the utilisation of fossil and nuclear fuels have been presented in many studies by interlinking fuel CED to the EROI analysis. As a part of these studies, Sgouridis et al.[43] present an approach by defining a constant exponential decay rate for fossil fuels and integrating it into a derived equation following Dale et al.[10]. The derived equation uses a time-based factor that is calibrated according to the historical depletion rate of resources, and it enables to foresee the estimation values for fossil fuels in terms of a CED perspective. Using this logic, Sgouridis et al.[43] derived an equation to calculate EROI for the 2015-2050 time period with 5-years intervals, which was used as a default to find fuel CED values, where these values address the energy requirement needed for the extraction, processing and transportation phases. Estimated EROI values are converted to fuel CED values in terms of MJ_el per kWh.

The vulnerable point of the equation derived by Sgouridis et al.[43] is using the same constant decay rate for oil, gas, and coal based on their historical consumption trends. At this point, an alternative estimation was necessary, specifically for natural gas and oil, to detect any deviations in EROI results. In a recent study, Delannoy et al.[44], envisage the significant changes in oil and natural gas EROIs from a global perspective in a defined period (1950-2050) by using dynamic decline functions. Following the same logic, the EROI values of Delannoy et al.[44] are converted to fuel CED values by following the same conversion calculation.

The available data accuracy from the extraction of nuclear resources to refining and enrichment processes has remained unclear; hence, the nuclear fuel datasets and processes in the ecoinvent database were used to resolve this difficulty. The approach was built around the technology types and capacity percentage changes on a global scale. The nuclear fuel CED is assumed to be the same for all years because the ecoinvent database is static and does not provide any projections for the future. This assumption should be uncritical in the BPS and Teske/DLR scenarios, but may be questionable in the IEA-SDS as a nuclear fuels-specific energy cliff may arise[67].

Notably, it is essential to emphasise that the found fuel CED values address the net energy consumption from the extraction of the selected source to transportation to power plants. It does not include any energy content of the fuel. The explanation, equations, and founded values are presented in the Supplementary Information Note 3.

## LUT-EROI model

Approximating more sustainable power systems, a ratio, energy return on investment (EROI), is defined as a partial analysis of net energy analysis. It is a ratio of final energy delivered by a process or a chain of processes to end-users ($E_{out}$) to total energy required to deliver that energy ($E_{inv}$). At this point, the traditional approach, depending on solely units of capacity, can be simplified to form an annualised calculation as seen in Eq. (2). For a process, $E_{out}$ is found by multiplying the annual supply, $E_{an}$, with the lifetime, LT, thus the simple definition of EROI can be described in terms of annual energy output and lifetime.

$$\text{EROI} = \frac{E_{out}}{E_{inv}} = \frac{E_{an} \cdot \text{LT}}{E_{inv}} = \frac{E_{an}}{(E_{inv}/\text{LT})} \tag{2}$$

Energy flow in a supply chain starts from extraction from sources, converted through various types of processes, resulting in products (fuels, electricity, and heat) consumed by end-users. EROI analysis for such a long process, disparities in features of technologies (efficiency, process types, lifetime) and the product utilisation purpose are indisputable main issues. Ensuring energy quality consistency[29] and continuity through all energy supply chain processes is the priority of a systemwide EROI analysis. Thus, the traditional approaches usually do not fit highly complex energy systems. Building upon Eq. (2), systemwide annual energy return on investment notated as $\text{EROI}_{sys,year}$, is given in Eq. (3). Note that $\text{ES}_{year}$ is annual net electricity supply, $t$ addresses the technology type (both existing and new) included in an energy system model whereas $n$ refers to the number of capacity units, as kW or kWh. Besides, Capacity$_t$ refers in terms of kW for electricity generators and kWh for storage devices and $N_t$ represents the maximum capacity of the technology.

$$\text{EROI}_{sys,year} = \frac{\text{ES}_{year}}{\sum\limits_{t} \frac{\left(\sum\limits_{n=1}^{N_t} E_{inv_{t,n}} \cdot \text{Capacity}_t\right)}{\text{LT}_t}} \tag{3}$$

The above calculation should not be confused with the power return on invested (PROI) metric, which enforces an energy flow technique and yet get confused with EROI that implements no energy flow[14,15]. The annual approach implemented in this study is a direct mathematical equivalent to a lifetime EROI calculation. A holistic approach for a systemwide EROI extends the boundary conditions to the end-user, which includes calculating all associated losses and investments through the energy supply chain[68]. Equation (3) presents this issue in terms of this holistic approach. After extracting annual self-consumption, curtailment, storage losses, and associated losses of related processes, yearly net electricity supply, $\text{ES}_{year}$, is supplied to the transmission network. Note that subtracting the transmission losses from $\text{ES}_{year}$ to reach the distribution level gives the net electricity

delivery to end-users, which is notated in EROI estimation as (F) to distinguish it from the generation level estimation (G). On the other hand, energy invested ($E_{inv}$) through the entire energy supply chain is estimated by using the found CED values on a technology level. For a technology, invested energy per unit of kW capacity, $E_{inv}$, is estimated by using Eq. (4).

$$E_{inv,t} = E_{capacity,t} + E_{operation,t} \cdot FLH_t \cdot LT_t \qquad (4)$$

For technology $t$, the converted CED value per unit of capacity is presented by $E_{capacity}$, whereas the operational CED value per unit of kWh is notated as $E_{operation}$. In dispatched power systems, the operating conditions of each technology must comply with the hourly demand requirement. Particularly, if the system goes toward 100% RE, the full load hour (FLH) of each technology becomes more crucial. Thereof, $E_{operation}$ was reinterpreted to fit this situation by multiplying it with the full load hour and lifetime duration of the related technology. To bring uniformity, the same methodology is applied for global and regional level analysis.

On the other hand, the estimate of the AEF is derived based on equation (5) and Eq. (6). Equation (4) is modified to provide annual energy investments for structural and operational purposes. $E_{inv_{t,year}}$ is the annual energy investment for the selected year of a technology ($t$). $E_{capacity_{t,year}}$ is the energy requirement for Capacity$_{t,year}$, which is the annual average value of newly added capacity for any 5-year time step. Capacity$_t$ is the total operational capacity in Eq. (5). This calculation step is repeated for each technology for the respective year and the AEF$_{year}$ for the selected scenario is calculated based on Eq. (6).

$$E_{inv_{t,year}} = E_{capacity_{t,year}} \cdot Capacity_{t,year} + E_{operation,t} \cdot FLH_{t,year} \cdot Capacity_t \qquad (5)$$

$$AEF_{year} = \frac{ES_{year}}{\sum_t E_{inv_{t,year}}} \qquad (6)$$

The AEF estimation differs from PROI in terms of methodology and conceptualisation, and does not capture the time transfer of energy flow as exactly implemented in the PROI. The AEF estimation does not amortise the upfront energy investment over the plant lifetime after installation, but allocates it to the period of investment commissioning.

Note that the final CED database (see Supplementary Data) underwent a calibration procedure by application of the LUT-EROI model at global, regional, and country levels. The procedure of selection of inventories and datasets, defined conversion factors, and CED values at a technology level, as well as a methodology workflow, are given in the Supplementary Information Note 2. Importantly, LUT-EROI is an Excel-based model using the results of LUT-ESTM, in this case outputs of Aghahosseini et al.[6], along with the final CED database. This model has been updated after analysing the implications for country[27] and regional level[39] analyses.

Finally, transparency and reproducibility should be significant features for modelling tools that are built on massive amounts of data and justified assumptions. The LUT-EROI model offers a standardised new methodology in the background for energy models and enables the reproducibility of EROI analyses by using the information provided in the SI. The LUT-EROI model can be merged with another energy system models and studies by altering the technology types and/or revising the CED values and operational characteristics. The technical approach of the model including the assumptions is provided in the Supplementary Information Note 1, Note 2, and Note 3.

## Limitations and uncertainties arisen from the nature of the models

The limitations of this study are born from the uncertainties originating from the methodological perspective, energy modelling, the conceptualisation of the CED analysis and the LUT-EROI model. Uncertainties coming from modelling significantly rely on the selection of aggregated data (both technical parameters and costs), the operational anomaly of the LUT-ESTM, the deviations in seasonal variations, and resource distribution on the regional level.

This study compares the CED values and identification of system boundaries at the technology level to the literature findings before estimating primary energy quality on the electricity level where both showed only small variations. From the EROI perspective, a sensitivity analysis for possible changes in embodied energy requirements for natural gas and oil sources is provided.

Other limitations related to the EROI analysis include using only one database for the CED analysis that uses a linear extrapolation method to estimate the CED values for technology, and unforeseen changes in technological advances, for instance, alterations in CED values under different energy learning rate percentages for specific technologies. The impact of such uncertainties can be deduced from the results analysing the CED values[29]. However, the foresight is supported by the output of published scientific findings/studies. In addition to this limitations, recycling of materials and waste is not considered in this study due to lack of sufficient specific data. The same situation occurs with the CED estimations related to the expansion of the transmission and distribution networks, and their CED values are at negligible amounts based on the limited LCIs given in the ecoinvent database. Thus, the EIs associated with these networks are not involved in the EROI estimates.

The relationship between EROI and LCOE is compared by a relative trend analysis of the corresponding modelled LCOE results and physical EROI estimations. The two indicators change during the ET in response to a change in the energy mix and technology selection. The reported relationship between EROI and LCOE may not follow similar conditions as the present system, where fossil fuel consumption takes a notable share of the required energy investment and LCOE. Systemwide ET studies with higher fossil fuel versus RE shares may reveal a more complex relationship of these two key energy system metrics, such as a time delay in the response or structural differences due to cost-price aspects or the role of supporting technologies among others. Future data will enrich the nature of this relationship. Importantly, this complex relationship should only be examined in ET studies at the system level, where multiple interactions between the technologies occur.

Intersectional impacts of socio-economic-political variations of the available financial mechanisms, direct and indirect effects of climate change, and the extreme events and other impacts on power systems are exempted from the scope of this study. Finally, the directions for further research should comprise a profound analysis of the sector coupling impact on EROI trends.

## Data availability
The data of this study are available from the authors upon reasonable request. The CED database that support the findings of this study are available in figshare with https://doi.org/10.6084/m9.figshare.24602349.v1[69].

## Code availability
All codes used for this paper are available upon reasonable request.

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

## Acknowledgements
A sincere thanks to Gabriel Lopez for proofreading this article. All authors gratefully acknowledge funding from Academy of Finland, for investigating biophysical limits of the energy transition (317681).

## Author contributions
H.S. was responsible for conceptualisation, data curation, formal analysis, methodology, original draft writing, revision, and editing. A.A.S. was responsible for conceptualisation, data curation, formal analysis, methodology, fund acquisition, project management, original design writing, revision, and editing. A.A. contributed to the conceptualisation of the scenarios, modelling, revision, and editing. C.B. contributed to the conceptualisation, formal analysis, methodology, fund acquisition, supervision and coordination, revision, and editing of the writing.

## Competing interests
All authors declare no competing interests.
