## [Peer Review File · Nature Communications]

Systemwide energy return on investment in a sustainable transition towards net zero power systemsEditorial Note: Parts of this Peer Review File have been redacted as indicated to remove third-party material where no permission to publish could be obtained.

REVIEWER COMMENTS

Reviewer #1 (Remarks to the Author):

This paper offers potentially an important contribution to the literature on the energy transition. All the results summarised in the abstract are important and significant for the transition from fossil fuels to renewable energy.

But, without reading the SI (and very few readers will), most readers of the manuscript will be unaware of several key assumptions of this research and will not understand important aspects of the problem. For example, in its attempt to be concise, the paper fails to explain the dynamic EROI problem that the LUT-EROI model apparently addresses (among others): namely, the more rapid the energy transition, the smaller the EROI of the system, because new energy technologies are being built and operated before existing energy technologies have generated the energy needed to build themselves. This research has a long history, e.g. the excellent mathematical analysis by John H. Price in Lovins & Price (1975) merits citation. This issue should be explained in any treatment of dynamic EROI. The abstract has the important statement of result “Shortening the transition period leads to a sharper declining of EROI, which stabilises after achieving 100% renewables”, but strangely this is missing from the results (Fig. 1) and the discussion of the results. Fig.1 needs an additional graph: EROI as a function of transition rate.

Most of the scenarios considered in this manuscript (e.g. IEA) fall short of the requirements of climate science. For the rapid transitions demanded by climate science, values of EROI could fall to much less than 10 until the transition has been completed. Only then could positive net energy be obtained. It's the price we may have to pay for past delays to rapid, effective action.

Another important point that merits inclusion in the results and discussion is the dependence of the EROI results on the choice of storage. As pointed out by Diesendorf & Wiedmann (2020), types of storage that are part of a generator (e.g. hydro with a single dam; CSP with thermal storage) may increase EROI of the whole system, while storage that is not part of a generator (e.g. battery; pumped hydro; compressed air) will decrease system EROI.

Although the manuscript cites the meta-analysis of Hall et al. (2014), it omits to recognise that their results for static EROIs of renewables must be discarded because they are averages over several decades and therefore grossly underestimate today's EROIs of solar PV and wind.

As the authors of the manuscript are no doubt aware, there has been considerable debate in the literature as to whether the output energy E_{out} should be defined in terms of electricity generated or primary energy equivalent (see papers by Raugei and colleagues). Perhaps I have read the manuscript, too quickly, but the choice of E_{out} and the reasons for the choice are unclear. Is that discussion buried in “the standardisation of energy quality”?

If the authors feel that such necessary discussions would make their manuscript too long for Nature

Communications, then they could submit it to Energy or Energies or, with a discussion of the policy implications, to Energy Policy.

References

Diesendorf M, Wiedmann T (2020). Implications of trends in energy return on energy invested (EROI) for transitioning to renewable electricity. *Ecological Economics* 176:106726.

<https://doi.org/10.1016/j.ecolecon.2020.106726>.

Hall, C.A.S., Lambert, J.G., Balogh, S.B., (2014). EROI of different fuels and the implications for society. *Energy Policy* 64, 141–152. <https://doi.org/10.1016/j.enpol.2013.05.049>.

Lovins AB, Price JH (1975). *Non-Nuclear Futures: The case for an ethical energy strategy*. Ballinger.

Reviewer #2 (Remarks to the Author):

Development of energy return on investment in a sustainable transition towards a net-zero CO2 power system

General comments

Overall, I found the article well-presented and well written. The analysis resulting from the assessment of a new system-wide EROI in energy transition scenarios is very interesting and timely. I specifically like the discussion of the sensitivity of global EROIs to the embodied energy of oil and gas, which are important features of this work. However, the manuscript has several flaws that make me recommend that it be accepted with major revisions. In particular, I would like to point out that three important references are not yet available, even as preprints, and that there is no comparison between the EROI results (overall and by technology) and the existing literature in the manuscript.

Specific comments

Abstract

p.1, l.1 I do not think that the net energy analysis should aim at "selecting the right energy mix", but rather at assessing the potentiality and practicality of the transition scenarios. One can also object that there is no "good" or "bad" energy mix (in reference to an ethical standard) independently of the use made of the energy.

p.1, l.3 It might be useful for the reader to add the climate target to which the nine global energy transition scenarios relate, as otherwise essential information is missing.

p.1, l.4 (here and elsewhere in the manuscript) the net energy cliff is a useful concept, I fear however that the upper and lower bounds are questionable. See Brandt, A.R. How Does Energy Resource Depletion Affect Prosperity? Mathematics of a Minimum Energy Return on Investment (EROI). *Biophys Econ Resour Qual* 2, 2 (2017). <https://doi.org/10.1007/s41247-017-0019-y> or Fizaine, F., & Court, V. (2016). Energy expenditure, economic growth, and the minimum EROI of society. In *Energy Policy* (Vol. 95, pp. 172–186). Elsevier BV. <https://doi.org/10.1016/j.enpol.2016.04.039>.

p.1, l.5-7 I find the sentence "The EROI trends, etc." not a major scientific milestone, and I think it might depreciate the impact of the next sentence "Once achieving etc.". The last sentence of the abstract is perhaps the most interesting element of the work done. I suggest that you reword the last three sentences and better emphasize the last one, and its implications for energy transition research.

Introduction

p.1, para 1, l.6 Dependence on fossil fuels can mainly be reduced by consuming less, let's not forget.

p.1, para 1, l.22 Generating consensus is, in my opinion, a difficult wish to achieve. I also find this addition difficult to grasp and feel that the sentence reads more clearly without it.

P.1, para 2 The literature review provided is efficient, but lacks three major studies whose results should later be compared to the findings of the present work in the Results section: Slameršak, A., Kallis, G., & O'Neill, D. W. (2022). Energy requirements and carbon emissions for a low-carbon energy transition. In *Nature Communications* (Vol. 13, Issue 1). Springer Science and Business Media LLC. <https://doi.org/10.1038/s41467-022-33976-5> ; King, L. C., & van den Bergh, J. C. J. M. (2018).

Implications of net energy-return-on-investment for a low-carbon energy transition. In *Nature Energy* (Vol. 3, Issue 4, pp. 334–340). Springer Science and Business Media LLC. <https://doi.org/10.1038/s41560-018-0116-1> ; Sers, M. R. (2022). Ecological macroeconomic assessment of meeting a carbon budget without negative emissions. In *Global Sustainability* (Vol. 5). Cambridge University Press (CUP).

<https://doi.org/10.1017/sus.2022.2>

p.1, para 2, l.14 Correct me if I'm wrong but I find it very surprising that reference 25, "Solomon, A. A., Sahin, H. & Breyer, C. The pitfall in designing future energy system without considering energy return on investment in planning. (2023)" is not yet available, even as a preprint. This lack prevents us from knowing how this work "has challenged these [other] studies conclusions and suggested the fundamental weaknesses of existing EROI estimation techniques".

p.1, para 2, l.19 The system-wide decline of EROI as the transition accelerates is not a point on which, to my knowledge, the NEA authors disagree, and as such I would not invoke the existence of "foregoing confusion."

p.1, para 3, l.1 This is the first occurrence of the LUT-EROI model. It seems to come a bit late, so I suggest that the authors explain in the previous paragraph how this model compares to the existing literature, and explicitly detail how refs 6 and 25 interact, if at all. Please also clarify whether the LUT-EROI model is an AMI, and if so, add Pehl et al. study in the literature review as they tried to incorporate EROI analysis into integrated assessment models for the power sector. (<https://www.nature.com/articles/s41560-017-0032-9>).

p.1, para 3, l.4 See the previous comment about the existence of a supposed "weakness of corresponding EROI studies." I think a rephrasing is necessary.

p.1, para 3, l.8 As ref. 25, I am unable to access the reference 27 "Solomon, A. A., Breyer, C. & Manjong, N. B. The necessity to standardize primary energy quality in achieving a meaningful quantification of

related indicators.”

Results

Perspective of systemwide EROI

p.1, para 1 l.17 Please highlight, succinctly, the advantages AND disadvantages of the LUT model - with particular reference to the dedicated section at the end of the document.

p.1, para 1, l.21 I would like the authors to detail how the selected scenarios "provide an excellent representation of the variety of transition pathways discussed" and analyze how they relate to the SSPs (or at least to the climate goals). A reference to the dedicated section at the end of the paper would also be an appropriate addition.

Impacts of the energy transition on global EROI

p.2, para 1, l.4 As previously mentioned, setting upper and lower bounds to the net energy cliff is questionable.

p.2, para 3, l.1 It is difficult for the reader to understand how the results differ from one scenario to another without first presenting the energy mix, energy demand and socio-economic dynamics by scenario in more detail (although these can be retrieved from the SI and the corresponding literature). A detailed reference to the corresponding section in the Data and Methods section should suffice.

p.2, para 3 What about the deployment of carbon capture and storage (CCS) when these technologies result in a significant net decrease in energy? See Fajardy, M., & Mac Dowell, N. (2018). The energy return on investment of BECCS: is BECCS a threat to energy security? *Energy & Environmental Science* (Vol. 11, Issue 6, pp. 1581–1594). Royal Society of Chemistry (RSC). <https://doi.org/10.1039/c7ee03610h>

p.2 para 3 + 5 These paragraphs lack, in my opinion, an effective purpose. I advise the authors to take back in particular the end of paragraph 3 and the whole of paragraph 5 which do not seem to me to be superfluous, but less important with regard to the rest.

p. 2, para 5 I also cannot find the reference 33 “Sahin, H., Solomon, A. A., Aghahosseini, A. & Breyer, C. The impact of spatial representation in energy transition modelling on system-wide energy return on investment. (2023).”

p.3, fig. 1 The figures are really interesting but I am afraid that they are not of sufficient quality in dpi. Figures (f) and (g) are also a bit blurry and difficult to interpret.

Effects of PV - storage - curtailment paradigm on the global EROI trends

p.4, para 3, l.9 I am surprised by the authors assertion that “nuclear power [...] achieves a lower EROI performance compared to wind and solar PV” as this goes against the emerging consensus. I would like the authors to explain why, and carry a brief analysis on how their EROI numbers compare to the existing literature, taking for instance Murphy, D.J.; Raugei, M.; Carbajales-Dale, M.; Rubio Estrada, B. *Energy Return on Investment of Major Energy Carriers: Review and Harmonization. Sustainability* 2022, 14, 7098. <https://doi.org/10.3390/su14127098> as a reference.

p.4, para 3, l.17 The last sentence is quite obvious, you can delete it.

Significant variations in EROIs due to regional differences

p.4 I think the authors' findings have implications for regional inequality in the transition. An interesting

additional analysis could explore (or at least mention) this.

Sensitivity of the global EROIs to the embodied energy requirements for natural gas and oil sources

p. 4 This section provides really interesting analysis which, I believe, should be further highlighted per region.

Discussions

p.5, para 1, l.18 Beware that all scenarios are only feasible from a technical point of view, and that the drop in the systemwide EROI leads to consequent adverse economic consequences. See Jacques, P., Delannoy, L., Andrieu, B., Yilmaz, D., Jeanmart, H., & Godin, A. (2022). Assessing the Economic Consequences of an Energy Transition Through a Biophysical Stock-Flow Consistent Model. In SSRN Electronic Journal. Elsevier BV. <https://doi.org/10.2139/ssrn.4174917> in this regard. You may also rely on Rye, C. D., & Jackson, T. (2018). A review of EROEI-dynamics energy-transition models. In Energy Policy (Vol. 122, pp. 260–272). Elsevier BV. <https://doi.org/10.1016/j.enpol.2018.06.041> (which you use in the SI) and Jackson, A., & Jackson, T. (2021). Modelling energy transition risk: The impact of declining energy return on investment (EROI). In Ecological Economics (Vol. 185, p. 107023). Elsevier BV. <https://doi.org/10.1016/j.ecolecon.2021.107023>

p.5 I would really like the authors to compare their findings with the work of Slamersak et al. and King and van den Bergh (2018), who estimated a 10-34%/24–31% reduction in net energy per capita for the IPCC 1.5°C SR/IEA low-carbon transition pathway, respectively.

Data and Methods

An overview of nine global energy transition scenarios

p.5, para 1, l.11 I feel like the socio-economic conditions of all scenarios need to be critically compared, as the information provided here are not sufficient to estimate how they compare between each other and relate to SSPs.

CED database for LUT-EROI

p.6, para 2, l.5 Not having access to the ref. 27 is critical here as I am unable to check how the uncertainty of estimating primary energy at the same energy quality has been solved (i.e., applying conversion factors).

Supplementary Information

I have trouble figuring out the share of several factors (increasing recycling capabilities, technological improvements, etc.) in the evolution of the EROI of the technologies. Could the authors elaborate and indicate how these results are challenged?

Reviewer #3 (Remarks to the Author):

Review of NCOMMS-23-01762

TITLE: Development of energy return on investment in a sustainable transition towards a net-zero CO2 power system

GENERAL COMMENTS

What are the noteworthy results?

--- The EROI over time of the electricity sector for a set of 9 future scenarios. This type of result is fine.

Will the work be of significance to the field and related fields? How does it compare to the established literature? If the work is not original, please provide relevant references.

--- This paper cites mostly sufficient studies, and I've mentioned a few more that provide insights into some of my feedback. The authors should also discuss how their study relates to a recent Nature Communications paper by Aljoša Slameršak et al (2022) that also calculates economy-wide EROI over time for some scenarios: "Energy requirements and carbon emissions for a low-carbon energy transition," <https://www.nature.com/articles/s41467-022-33976-5>

MAJOR COMMENTS

- 1) MAIN COMMENT: It is not clear to me that this methodology represents the full dynamics of EROI in relation to when installation occurs, causing the majority of CED at the beginning of the life of most renewable technologies, versus when electricity is generated relatively uniformly over the lifetime of the technology. I take your equation (2) as representing the calculations you are plotting in Figure 1. However, my interpretation of the method is that it is calculating a value every 5 years that is not a number we should want to know because it provides a confusing interpretation about the state of the energy system for any given year. The method seems to be calculating "For a given year, e.g., 2030, the weighted average lifetime EROI of the technologies that are installed in 2030 is EROI = X." which is specifically different from calculating "For a given year, e.g., 2030, the electricity system has an EROI = X calculated by dividing annual electricity output by annual energy inputs."

- a. Your Equation (2):
$$EROI_{sys,year} = \frac{SG_{year}}{\sum_t \frac{(\sum_{n=1}^{N_t} E_{inv,t,n} \cdot Capacity_t)}{LT_t}}$$

- b. Equation (3) defines E_{inv} as the entire lifetime energy investment, since it adds $E_{capacity}$ to $E_{operation} \cdot FLH \cdot LT$.
- c. Thus, the denominator of Equation (2) is = (total lifetime energy of existing generation fleet)/LT = annualized energy investment regardless of when the energy investment occurs.

- d. Thus, since the denominator of Equation (2) divides E_{inv} by LT, it is dividing the annual generation (SG_{year}) by a quantity of energy investment that does not relate to a quantity of invested energy (to date) that provides an insight that is useful.
- e. There are at least 2 problems with the approach of Equation (2):
 - i. Construction energy investment: Equation (2) is calculating the energy investment of construction by the lifetime of the plant, but this is not an accurate description of what is actually happening. 100% of the construction occurs at the beginning and it is not spread over the life of the project, so this makes the EROI calculation seem higher than what comes from an accurate (I think) definition of the input dynamics. This is not a realistic representation of real-world energy flows. IN SHORT: This biases EROI upward for all years (I suppose until decommissioning).
 - ii. Decommissioning energy investment: Equation (2) is dividing generation in a current year by energy investment for decommissioning that has not yet occurred. This is not a realistic representation of real-world energy flows. IN SHORT: This biases EROI downward until decommissioning.
- f. I will point out: Many papers calculate EROI of a system as = (annual energy output)/(annual energy invested). Some call this "PROI" (power return on power invested), and I think providing this calculation would add insight.
 - i. My understanding (you can correct me) is that you are NOT attempting to do this calculation of "PROI", and this is OK, but you are calculating an "annualized PROI-type" metric since it has units of (power output)/(annualized power input) = (energy output per year)/(annualized energy input per year) rather than (energy output per year/energy input per year).
- g. I am indicating the issues laid out in these two (among other) publications:
 - i. [You already cite this]: Murphy et al. (2011) *Sustainability* (<https://www.mdpi.com/2071-1050/3/10/1888>) Figure 4 showing that CED is the integral of power flows each year, and you can see in Equations (1) and (2) of this paper that the authors define the acronym EROI using two methods. Their Equation (2) is essentially "PROI".
 - ii. King et al (2015) *Energies* Part 1 that explicitly shows the connection between "power return ratios" and "energy return ratios" (<https://www.mdpi.com/1996-1073/8/11/12346>) to avoid the confusion. Figures 1-4 show how this plays out for a simple system.
 - iii. My interpretation of your denominator of your Equation (2) is that the inputs if represented in Figure 4 of Murphy et al (2011) and Figures 1 & 2 of King et al (2015) would simply be a constant value for each technology you add to the system, rather than fluctuating from construction-operation-decommission phases over a technology lifetime as indicated in those publications.
- h. To me, the most accurate representation of the system-wide EROI calculation you are studying is two use both of the following approaches:
 - i. (X) "PROI_{system}" = (annual energy output)/(annual energy invested).
 - ii. (XX) EROI_{system} = (integral of annual energy generation to date)/(integral of annual energy investment to date)

- iii. Since the system of “the grid” does not have a lifetime (i.e., we expect it to continue indefinitely), both approaches avoid the assumption of how a power plant lifetime affects the energy input by simply counting the energy inputs and outputs each year. The system is simply composed of parts that continuously generate electricity (or get decommissioned) and continuously have energy investment, and you track this over time. So including the value of the lifetimes (e.g., 30 years) of individual plants in your Equation (2) becomes unnecessary.
 - iv. Yes, there are inherent difficulties with the approach (XX), primarily understanding the initial conditions (this is OK for me). But your current methodology already needs to explicitly describe how you calculate the initial EROI calculation for your first year of 2015 and/or 2020 (since it seemingly depends on the EROI of all technologies that exist now, as based on when they were installed) and maybe this becomes obvious after you address my comments on your Equation (2) more generally.
- 2) “A further analysis of the EROI relation with system levelised cost of electricity (LCOE) (Fig. 1j) shows that low-cost solutions correlate to low EROI.”
- a. Per the units of LCOE (money input/kWh output) and EROI (kwh output/energy input), they are inversely related, so that we might normally expect high EROI and low LCOE, and vice versa. Example see King and Hall (2011), <https://www.mdpi.com/2071-1050/3/10/1810>, applied to U.S. oil and gas to oil prices (not a levelized cost, but a price) but similar idea.
 - b. But you are showing declining EROI can be associated with also declining LCOE.
 - c. Also, I think your Figure 1(j) can simply be a 2-dimensional plot of EROI vs. LCOE, and perhaps label each data point with a year (or the first and last data points with the first and last years so we can see how any given trajectory changes over time).
 - d. REQUEST: Please explain in more detail why you see this opposite trend (from what I’m stating) as you state in the text. This answer might come from addressing my first “main” comment.
 - e. QUESTION: In each year, do you adjust your EROI and LCOE calculations based upon the capacity factor of each power plant or category of power plants (i.e., since I presume capacity factor of coal and/or natural gas plants go down over time in your scenarios)? Perhaps you can see the conundrum here, the same power plant might calculate an EROI of 20 in year 2025 with capacity factor 90% would have a lower EROI calculation if using its dispatched capacity factor of 50% in 2040.
 - f. NOTE: If my interpretation of your Equation (2) is correct, then I do think that your direct comparison of LCOE and $EROI_{system}$ is possibly consistent, since LCOE can also be put into terminology of $LCOE = (\text{annualized money inputs})/(\text{annualized electricity output})$ but it then assumes the SAME annualized money inputs each year. In reality, the full cost of money to construct and install the power plant was paid (to the builders) at the beginning of the project just as all of the energy inputs for construction were consumed at the beginning. So we know LCOE is an abstraction (a summary) of the real cash flows, just as you seem to be creating the similar abstraction of energy flows. I

don't think it is useful to promote this type of abstraction of energy flows (say, average annual power = energy/year) that net energy analysis is trying to illuminate.

- 3) "The ecoinvent database provides CED values by separating them into eight sub-categories according to the energy resource type as MJ equivalents. The same category logic presented by ecoinvent is applied for the determination of conversion factors, and the MJ equivalent unit is converted to MJ-electricity on a technology level."
 - a. Please write as simple equation as possible to express what you are saying in words. It is hard to follow the conversions from words, so be explicit using an equation of how you convert between "MJ equivalents" and "MJ-electricity".

MINOR COMMENTS ON ARTICLE CONTENT:

- 4) "Some of these studies questioned the plausibility of 100% RE in terms of net energy production."
 - a. Please repeat the citations (already existing) the relate to this statement.
- 5) "Hereby, this study expands the newly developed LUT-EROI ..."
 - a. I think you still need to spell out LUT the first time ...
- 6) "one of the two most used tools for highly RE system analyses ..."
 - a. "high" instead "highly"?
- 7) The Solomon references 25, 27, and 33 don't have any publication name associated with them. Are they in journals? Are they white papers? Something else?
- 8) "Revisiting the foregoing discussion of EROI trends by combining Fig. 1a to Fig. 1d reveals that, despite of scenario dependent differences,"
 - a. Make it "... despite scenario-dependent differences"

MINOR COMMENTS ON SUPPLEMENTAL

- 9) Section 2.4, "in Table S1Hata! Başvuru kaynağı bulunamadı,..." probably needs to be corrected.
- 10) For clarity, just show the use of a conversion factor in Table S1. That is to say, write something such as " $MJ_{el} = (\text{conversion factor})MJ_{pe-eq}$ " (but ensure to write the equation as you use it). Also specify if one needs to use the conversion of 3.6 MJ = 1 kWh.
- 11) Figure S4. Should it read "Phase III Calculating Technology Level Invested Technology"
 - a. That is it is "Inverted" or "Invested"?

Reviewer #1	Response to Reviewer #1
Reviewer comments: This paper offers potentially an important contribution to the literature on the energy transition. All the results summarised in the abstract are important and significant for the transition from fossil fuels to renewable energy.	Thank you for your valuable comments and suggestions. Kindly note that the changes are already reflected in the revised manuscript via the track change module. However, for the sake of clarity and to save time, we have also indicated the direct changes made in accordance with Reviewers' comments are given in blue. The words/phrases in bold and underlined refer to specific changes. Apart from the moderate/minor changes requested by the reviewers, the major ones are presented in the revised manuscript.
But, without reading the SI (and very few readers will), most readers of the manuscript will be unaware of several key assumptions of this research and will not understand important aspects of the problem.	We agree. Unfortunately, we deal with a very large dataset both at the input and output level. Just considering the input, this study brings together large LCA dataset, modelled energy transition scenario results and datasets to perform a comprehensive analysis, the detail of which can hardly be compressed into the main paper. To provide full perspective, the issues related to the development of the CED database are elaborated in the SI and the energy transition scenarios are already published separately in Aghahosseini et al.¹. Thus, the paper is focused on presenting the main key findings. This secures the uniformity and flow of the main paper. Instead of including all information in the main text, we believe that the importance of the SI has been emphasized in the paper several times, which will direct readers to the SI. Not to forget, the word limit of the journal is a tough constraint for presenting information in the main paper. We intend to be as transparent as possible, therefore, the SI is also quite comprehensive.
[] For example, in its attempt to be concise, the paper fails to explain the dynamic EROI problem that the LUT-EROI model apparently addresses (among others): namely, the more rapid the energy transition, the smaller the EROI of the system, because new energy technologies are being built and operated before existing energy technologies have generated the energy needed to build themselves. This research has a long history, e.g. the excellent mathematical analysis by John H. Price in Lovins & Price (1975) merits citation. This issue should be explained in any treatment of dynamic EROI.	Thank you. The recommended reference is added in the Discussion section together with following revision: The findings on EROI clarify that the replacement of fossil fuel technologies with RE results in lower EROI values supporting the arguments of Lovins and Price⁴⁷.
The abstract has the important statement of result "Shortening the transition period leads to a sharper	The relevant statements can be found in the third paragraph of the subsection "Impact of the energy transition on the global EROI" and is copied below.

declining of EROI, which stabilizes after achieving 100% renewables”, but strangely this is missing from the results (Fig. 1) and the discussion of the results. Fig.1 needs an additional graph: EROI as a function of transition rate.	The decline after 2025 (Fig. 2a) is associated with the expansion of both solar photovoltaics (PV) and wind power capacities and of the enabling technologies, mainly batteries and gas storage towards the end of the ET period. The upsurge of renewables and emerging technologies' installed capacities brings an additional burden to total energy invested (EI), leading to a decline in EROI. However, the rate of decline changes considerably with scenarios. Three scenarios, namely BPS-NWF, BPS-WF, and BPS-plus2040 show a gradual decline with small differences between their trends. The differences can be credited to changes in the energy mix, which can be seen from the differences of BPS-NWF and BPS-WF, and the difference in storage build-up, especially as observed in the last period. On the other hand, the BPS-plus2030 and BPS-plus2035 scenarios show a sharp decline between the 2025-30 and 2030-35 periods, respectively. As the transition in these BPS-plus scenarios is ambitiously carried out in the specified periods, EROIs of these scenarios decline more slowly in the later periods, showing a sign of EROI stabilisation after achieving 100% RE. The underlying reason for the sharp decline is the very fast-growing capacity need for solar PV systems and their complementary technologies. But we erroneously linked the discussion to Fig. 1 rather than Fig. 2, which we have now corrected. Also, the indication of the importance of this result is again emphasized in the first paragraph of discussion section as: (iii) restricting ET time leads to sharp declines in EROIs temporarily, but as time proceeds, EROIs are stabilised. At this point, we would like to emphasize that the analysis of the speed of the energy transition is only limited to BPS-plus scenarios and the analysis is only done by comparing their EROI trends already given in Fig. 2a. Thus, we believe that an additional graph is not necessary as the relationship between EROI trends and transition period can be seen from Fig. 2. But we apologise for causing the confusion by linking the discussion to Fig. 1 than Fig. 2. Many thanks for noticing.
Most of the scenarios considered in this manuscript (e.g. IEA) fall short of the requirements of climate science. For the rapid transitions demanded by climate science, values of EROI could fall to much less than 10 until the transition has been completed. Only then could positive net energy be obtained. It's the price we may have to pay for past delays to rapid, effective action.	Thank you for the comments. As regards to the scenarios' relevance to climate requirement, we added the following clarification to the manuscript. Nine global energy transition scenarios form the ground of this net energy study. Out of nine transition scenarios, three (BPS-plus scenarios) aim at achieving zero CO₂ emissions before 2050, while four scenarios (the two BPS and two Teske/DLR scenarios) target the same by 2050 and the remaining two IEA scenarios achieve the goal after 2050. Thus, the scenarios selection was motivated to get a diverse representation of climate change targets together with an associated change in the energy mix and system cost.

	The LUT Energy System Transition Model (LUT-ESTM) is a cost optimization model, thereof, the scalability to cover all SSPs (including socio-economic challenges and factors) is smaller and restricted as in other cost-optimization energy modelling tools. Additionally, it aims to small-medium term targets (by 2050) rather than setting up long-term targets (up to 2100). Therefore, it differs from the IAM and the Global Energy and Climate (GEC) models in terms of model structure and objective of the model. None of scenarios presented in here are designed considering RCP and/or SSP, which is also not the aim of this paper or LUT models. The new IEA scenario - Net Zero Emissions by 2050 – could not be examined due to: (i) as the modelling is finalised before this scenario is published, and (ii) lack of data to run LUT-ESTM at regional level. The issue of lack of data and transparency of the Net Zero Emissions by 2050 scenario is emphasized in Aghahosseini et al.¹. We used the Aghahosseini et al.¹ study outputs to execute a physical EROI analysis. The current EROI literature handles RCP/SSP scenarios, of course, they provide remarkable insights, but none of them see the decarbonisation by 2050. This is one of the main gaps in the EROI literature. There are also key methodological differences in addressing the current shortcomings of EROI estimation. We agree that the EROI declines. However, the possibility of falling below 10 depends on various factors such as modelling tool, assumptions of the model, selected unit costs of technologies, expected improvements in technologies, preferences for setting the scenarios, setting system boundaries of LCA, estimating CED values, etc. In this study, the EROI of all scenarios remain above 10. We also show that the EROI declines, and then stabilises after achieving 100% RE. These issues are presented in detail in the manuscript.
Another important point that merits inclusion in the results and discussion is the dependence of the EROI results on the choice of storage. As pointed out by Diesendorf & Wiedmann (2020), types of storage that are part of a generator (e.g. hydro with a single dam; CSP with thermal storage) may increase EROI of the whole system, while storage that is not part of a generator (e.g. battery; pumped hydro; compressed air) will decrease system EROI.	We thank the Reviewer for the well-pointed comment. Our findings are consistent with the Reviewer#1 statement, and this issue is highlighted in the 4th paragraph of the Results section, as follows: Evidently, storage technologies that are a part of power generators (e.g., TES, hydropower with reservoir etc.) support the increase in system EROIs, conversely complementary storage technologies (such as batteries, hydrogen, and methane storages) that are integrated into electricity generators lead to a substantial drop in overall EROIs as correspond to the findings of Diesendorf and Wiedmann²¹.

Although the manuscript cites the meta-analysis of Hall et al. (2014), it omits to recognise that their results for static EROIs of renewables must be discarded because they are averages over several decades and therefore grossly underestimate today's EROIs of solar PV and wind.	Thank you for the recommendation. We included this paper to provide more variety in the EROI. However, considering the comment, we decided to exclude this reference.
As the authors of the manuscript are no doubt aware, there has been considerable debate in the literature as to whether the output energy E_{out} should be defined in terms of electricity generated or primary energy equivalent (see papers by Raugei and colleagues). Perhaps I have read the manuscript, too quickly, but the choice of E_{out} and the reasons for the choice are unclear. Is that discussion buried in “the standardisation of energy quality”?	Yes, we are aware of the debate regarding energy quality. Two of the authors (namely: A. A. Solomon and Christian Breyer) have already explored the subject on a separate paper. The paper, which is currently under review has investigated the matter in detail and concluded that standardizing energy quality or the unit joule is the only solution to the associated discrepancies. They also found that electricity stands as the best option to solve the present problem, until further unifying solutions are presented. This paper thus converts all energy forms to electricity before doing any calculation as described in the SI, without this it is impossible to achieve a proper comparability. We believe this is logical and a basic scientific approach.
If the authors feel that such necessary discussions would make their manuscript too long for Nature Communications, then they could submit it to Energy or Energies or, with a discussion of the policy implications, to Energy Policy.	Thank you for the reviewer's recommendation, but we strongly believe that Nature Communications is the best fit for this paper. Also, we extended the manuscript content according to the Reviewers' comments as relevant.

Reviewer #2	Response to Reviewer #2
Reviewer comments: Overall, I found the article well-presented and well written.	Thank you for the valuable comments and suggestions. Kindly note that the changes are already reflected in the revised manuscript via the track change module. However, for the sake of clarity and to save time, we have also indicated the direct changes requested by the reviewers in apostrophes in the blue sentences. The words/sentences in bold and underlined refer to specific changes. Apart from the moderate/minor changes requested by the reviewers, the major ones are presented in the revised manuscript.
The analysis resulting from the assessment of a new system-wide EROI in energy transition scenarios is very interesting and timely. I specifically like the discussion of the sensitivity of global EROIs to the embodied energy of oil and gas, which are important features of this work. However, the manuscript has several flaws that make me recommend that it be accepted with major revisions. In particular, I would like to point out that three important references are not yet available, even as preprints, and that there is no comparison between the EROI results (overall and by technology) and the existing literature in the manuscript.	As Reviewer #2 stated three important references are not available as preprints. These papers are under peer review process in well-recognized journals, and unfortunately, it takes time to be published. However, we would like to point out that the core of the methodology, derived equations, used conversion factors as well as any fundamental information is also summarized in the main paper as well as in SI to provide a clearer view of the developed methodology and the implementation of systemwide EROI analysis. We will respond to the issue of comparison later, in association with related specific comment given below.
Abstract	
p.1, l.1 I do not think that the net energy analysis should aim at "selecting the right energy mix", but rather at assessing the potentiality and practicality of the transition scenarios. One can also object that there is no "good" or "bad" energy mix (in reference to an ethical standard) independently of the use made of the energy.	Selecting the right energy mix is revised as specifying the optimal system development strategy. We agree with this remark but reiterate that the energy mix is one of the main factors affecting the trends of the EROI.
p.1, l.3 It might be useful for the reader to add the climate target to which the nine global energy transition scenarios relate, as otherwise essential information is missing.	The sentence is corrected as: This study examines the net energy performance of nine decarbonisation global energy transition scenarios until 2050 by applying a newly developed systemwide energy return on investment (EROI) model. Further details on the scenarios are explained in detail in the sub-subsection of “An overview on nine global energy transition scenarios”.

p.1, l.4 (here and elsewhere in the manuscript) the net energy cliff is a useful concept, I fear however that the upper and lower bounds are questionable. See Brandt, A.R. How Does Energy Resource Depletion Affect Prosperity? Mathematics of a Minimum Energy Return on Investment (EROI). Biophys Econ Resour Qual 2, 2 (2017). https://doi.org/10.1007/s41247-017-0019-y or Fizaine, F., & Court, V. (2016). Energy expenditure, economic growth, and the minimum EROI of society. In Energy Policy (Vol. 95, pp. 172–186). Elsevier BV. https://doi.org/10.1016/j.enpol.2016.04.039.	We agree with Reviewer#2's point that several studies touch on this issue and have not reached a consensus about the lower and/or upper limits for the net energy cliff. The first time, a 10:1 ratio is brought forward in Lee's study (1969)³ and then has been referred to in many studies by Lambert et al.^{4,5}. This ratio is actually meaningful when we look at recent studies, such as Brockway et al.⁶, which states a sharp boundary for both fossil fuels and renewables; under this value, the born risks start to appear to society, and below 5 means that the system is not favourable anymore, holding high risks. Brockway et al.⁶ also explain this ratio as 90% of obtained energy can be useful to society. On the other hand, the study in Capellán-Pérez et al.⁷ provided risk levels collected from EROI studies following as 10-15:1 is no risk, 5-10:1 is low risk, and below 5:1 is dangerous whereas 2-3:1 is not feasible. Indeed, a few studies, including one's stated by Reviewer#2, offer different minimum limits for the net energy cliff, still, the definition of limits are changing due to the variances in methodology and scope of the studies. Clearly the boundary condition of that study is not the same as ours as we study different systems and use different tools. We know that a single discrepancy in energy quality affects EROI values of fossil fuels and renewables differently. Thus, no one has the correct answer as to what will be the minimum as of now. More interestingly, for two important reasons we used the upper limits of the energy cliff. First, we calculate the physical EROI for which going below the upper limits is linked to a loss in efficiency. Thus, we know that there is no advantage of going below 10. Second, this study involves future systems, thus carries uncertainties of its own. Therefore, instead of following the lower bound of the net energy cliff, it is appropriate to use the upper limit of the net energy cliff, as this value shows no risk based on studies of Brockway et al.⁶ and Capellán-Pérez et al.⁷. The lower limit should be studied separately by considering the complexity of estimating the physical EROI and the impact of falling below the upper limit of the energy cliff. But this is out of the scope of this study and should be handled from the methodological perspective.
p.1, l.5-7 I find the sentence "The EROI trends, etc." not a major scientific milestone, and I think it might depreciate the impact of the next sentence "Once achieving etc.". The last sentence of the abstract is perhaps the most interesting element of the work done. I suggest that you reword the last three sentences and	Thank you for the nice recommendation. However, the statement is the direct result of our findings, which links the change in EROI trends to the energy transition pathways, thereof, the order of the sentences is set to be consistent with the flow of results. At the same time, this is rarely reported and thus required to be emphasized. If it is convenient for Reviewer#2, we would like to leave it as it is.

better emphasize the last one, and its implications for energy transition research.	
Introduction	
p.1, para 1, l.6 Dependence on fossil fuels can mainly be reduced by consuming less, let's not forget.	A small addition is made considering Reviewer#3 statement. The added expression is: The Russian war in Ukraine has clearly shown that despite all efforts, fossil fuel dependency is still a sore point of present power systems considering the immense turbulence in commodity markets due to the gas crisis and despite the depotentiation of fossil fuel consumption!
p.1, para 1, l.22 Generating consensus is, in my opinion, a difficult wish to achieve. I also find this addition difficult to grasp and feel that the sentence reads more clearly without it.	We excluded and generate consensus to provide more clarity as suggested.
P.1, para 2 The literature review provided is efficient, but lacks three major studies whose results should later be compared to the findings of the present work in the Results section: Slameršak, A., Kallis, G., & O'Neill, D. W. (2022). Energy requirements and carbon emissions for a low-carbon energy transition. In Nature Communications (Vol. 13, Issue 1). Springer Science and Business Media LLC. https://doi.org/10.1038/s41467-022-33976-5 ; King, L. C., & van den Bergh, J. C. J. M. (2018). Implications of net energy-return-on-investment for a low-carbon energy transition. In Nature Energy (Vol. 3, Issue 4, pp. 334–340). Springer Science and Business Media LLC. https://doi.org/10.1038/s41560-018-0116-1 ; Sers, M. R. (2022). Ecological macroeconomic assessment of meeting a carbon budget without negative emissions. In Global Sustainability (Vol. 5). Cambridge University Press (CUP). https://doi.org/10.1017/sus.2022.2	The relevant references are added to the introductory part and the results of these studies are compared to this paper's findings in the Results section. We thank Reviewer#2 for this nice contribution. For the introduction part: The pioneering EROI studies present diverse concepts due to methodological inconsistencies^{7,8}, with some deriving varieties of EROI concepts⁹⁻¹², implementing different boundary conditions¹³⁻¹⁵, comparing fossil fuel and RE technologies with or without enabling technologies¹⁶⁻¹⁹. However, recent studies²⁰⁻²³ diverted this attention towards EROI analysis for ETs^{24,25} to foresee the feasibility of 100% RE systems²⁶. References numbers:  19. King, L. C., C.J.M., J. & van den Bergh, M. Implications of net energy-return-on-investment for a low carbon energy transition. Nature Energy 3, 334–340 (2018). 24. Sers, M. R. Ecological macroeconomic assessment of meeting a carbon budget without negative emissions. Global Sustainability 5, e6 (2022). 25. Slameršak, A., Kallis, G. & Neill, D. W. O. Energy requirements and carbon emissions for a low-carbon energy transition. Nature Communications 13, 1–15 (2022).

p.1, para 2, l.14 Correct me if I'm wrong but I find it very surprising that reference 25, “Solomon, A. A., Sahin, H. & Breyer, C. The pitfall in designing future energy system without considering energy return on investment in planning. (2023)” is not yet available, even as a preprint. This lack prevents us from knowing how this work “has challenged these [other] studies conclusions and suggested the fundamental weaknesses of existing EROI estimation techniques”.	Unfortunately, this paper is still in the process of peer-review in a recognized journal in the field and will take a while to be published. However, the core of the methodology, derived equations, conversion factors used, and all basic information are provided in both the main paper and the SI. In this paper, the newly developed systemwide EROI methodology is implemented at the country level and a comprehensive sensitivity analysis is performed for storage technologies.
p.1, para 2, l.19 The system-wide decline of EROI as the transition accelerates is not a point on which, to my knowledge, the NEA authors disagree, and as such I would not invoke the existence of "foregoing confusion."	It is revised considering Reviewer#2 comments as follows: Thus, applying an improved systemwide EROI tool to a global transition scenario can eliminate these salient issues while simultaneously contributing to enhancing the ET path selection.
p.1, para 3, l.1 This is the first occurrence of the LUT-EROI model. It seems to come a bit late, so I suggest that the authors explain in the previous paragraph how this model compares to the existing literature, and explicitly detail how refs 6 and 25 interact, if at all. Please also clarify whether the LUTEROI model is an AML, and if so, add Pehl et al. study in the literature review as they tried to incorporate EROI analysis into integrated assessment models for the power sector. (https://www.nature.com/articles/s41560-017-0032-9).	We are grateful for Reviewer#2's suggestion to clarify this issue. LUT-ESTM is a cost optimization energy modelling tool built using MATLAB as a compiler and Mosek as a solver. The current version of LUT-ESTM offers a techno-economic perspective. It is no IAM model, but optimised for high temporal and spatial resolution for answering research questions how to reach high shares of renewables for full system stability – and aspect for what IAMs fall short. On the other hand, LUT-EROI is an Excel-based tool that we have been developing over the past years and is also still under development. LUT-EROI uses the output of LUT-ESTM and a CED database that we created as discussed in the SI. Aghahosseini et al.¹ (Ref.6 in the manuscript) presents the detailed analysis of the energy transition scenarios used in this study. The scenarios were also reproduced using LUT-ESTM. Aghahosseini et al.¹ (Ref.6) and Slameršak et al.² (Ref. 25 in the manuscript) does not interact with each other at all. Some explanations are added to clarify this situation. The text on LUT-EROI is now “Excel-based LUT-EROI (LUT stands for Lappeenranta-Lahti University of Technology)”. We also specifically refer to LUT-ESTM as cost optimization model (not an IAM) in the “Perspectives of systemwide EROI” section.
p.1, para 3, l.4 See the previous comment about the existence of a supposed "weakness of corresponding EROI studies." I think a rephrasing is necessary.	We have corrected this word as gap and/or shortcoming in the main paper. The revisions are:

	A recent study that implemented an advanced systemwide EROI approach has challenged these studies' conclusions²⁷ and suggested that fundamental gaps of existing EROI estimation techniques, such as the inability to capture the impact of optimal interoperability of multi-processes, and the typical methodological gaps of EROI²⁸, may lead to such conclusions. “The overall modelling framework improved on the existing shortcomings of corresponding EROI studies and enhances the representativeness of the estimated physical EROI values by implementing a holistic approach for estimating primary energy quality at electricity level²⁹, creating a broader cumulative energy demand (CED) database for technologies based on life cycle assessment (LCA) databases²⁸, and integrating EROI estimation with energy system model output²⁷.
p.1, para 3, l.8 As ref. 25, I am unable to access the reference 27 “Solomon, A. A., Breyer, C. & Manjong, N. B. The necessity to standardize primary energy quality in achieving a meaningful quantification of related indicators.”	That paper explores the subject of energy quality and recommends standardization of energy quality or the unit “Joule” to solve related discrepancies in broader energy area including EROI. Including this manuscript, three manuscripts cited in this manuscript are undergoing peer-reviewing in different journals, and as a result they are not yet in public use. However, as mentioned, the fundamental information for methodology and the derivation of equations, and the implementation of EROI analysis is given both in the main paper as well as SI. Thus, we wrote this paper in a self- explanatory way but cited those manuscripts for exploring further details. These other details have another context that should stand on its own. It is very overwhelming to put various independent ideas in one paper and the message could simply get lost. Thus, we have no option than to present them independently. The limited words per publication is a further challenge.
Results	
Perspective of systemwide EROI	
p.1, para 1 l.17 Please highlight, succinctly, the advantages AND disadvantages of the LUT model – with particular reference to the dedicated section at the end of the document.	The advantages and disadvantages of LUT-ESTM with particular references are added to at the end of the related paragraph. Kindly see the additional sentences in below: LUT-ESTM is ranked as a leading cost-optimisation energy system model³⁶ and one of the two most used tools for highly RE system analyses²⁶. Dissimilar to Integrated Assessment Models (IAMs), LUT-ESTM is designed for analysing short- and medium-term goals to achieve 100% net-zero CO₂ power systems, thus, the software tool architecture is structured around this scope rather than targeting the long-term climate goals, as these can be set as constraints. It is a multi-scale energy modelling tool that allows flexible implementation at national, regional, and global scales³¹. As with all cost-based energy modelling tools,

	techno-economic parameters are the underpinnings of the structure, so any major modification of them gradually leads to large deviations. Using reliable and widely accepted references and iterating micro- and macro-scale energy systems in different studies is one way to overcome this disadvantage^{26,37}. LUT-ESTM uses updated and internationally recognised references clarified in Aghahosseini et al.⁶ and Bogdanov et al.³⁰.
p.1, para 1, l.21 I would like the authors to detail how the selected scenarios "provide an excellent representation of the variety of transition pathways discussed" and analyze how they relate to the SSPs (or at least to the climate goals). A reference to the dedicated section at the end of the paper would also be an appropriate addition.	The following information further clarify this issue:  ▪ The BPS scenarios (BPS No Wind Force and Wind Force) are designed to anticipate the changing weight of solar and wind technologies in the energy mix when wind repowering is implemented. ▪ BPS-plus scenarios aim to shorten the time to complete the energy transition without applying restrictions as applied in BPS scenarios. BPS-plus2030 refers to energy systems that will achieve carbon neutrality in 2030, while the suffix of BPS-plus scenarios (as a year) also represents the final year of the ET transition. The energy mix of these scenarios shows that the share of renewable energies, the phasing out of fossil fuels, and nuclear energy occur in different periods. This helps us to understand whether or not there is a clear benefit in shortening the ET period from an EROI perspective, or whether there are bottlenecks that we might encounter. ▪ On the other hand, IEA and Teske/DLR scenarios are simulated to provide a fair platform for comparing BPS and BPS-plus scenarios with their results. Notably, IEA scenarios do not aim for full decarbonisation of the power sector by 2050, and Teske/DLR scenarios are targeting full decarbonisation considering socio-economic restrictions. Please see the modification that we have made to the manuscript to address this and your other related comment above. The following statements are added to the manuscript. Nine global energy transition scenarios form the ground of this net energy study. Out of nine transition scenarios, three (BPS-plus scenarios) aim at achieving zero CO₂ emissions before 2050, while four scenarios (the two BPS and two Teske/DLR scenarios) target the same by 2050 and the remaining two IEA scenarios achieve the goal after 2050. Thus, the scenarios selection was motivated to get a diverse representation of climate change targets together with an associated change in the energy mix and system cost.

The LUT Energy System Transition Model (LUT-ESTM) is a cost optimization model, thereof, the scalability to cover all SSPs (including socio-economic challenges and factors) is smaller and restricted as in other cost-optimization energy modelling tools. Additionally, it aims to small-medium term targets (by 2050) rather than setting up long-term targets (up to 2100). Therefore, it differs from the IAM and the Global Energy and Climate (GEC) models in terms of model structure and objective of the model. None of scenarios presented in here are designed considering RCP and/or SSP, which is also not the aim of this paper or LUT models.

The new IEA scenario - Net Zero Emissions by 2050 – could not be examined due to: (i) as the modelling is finalised before this scenario is published, and (ii) lack of data to run LUT-ESTM at regional level. The issue of lack of data and transparency of the Net Zero Emissions by 2050 scenario is emphasized in Aghahosseini et al.¹. We used the Aghahosseini et al.¹ study outputs to execute a physical EROI analysis. The current EROI literature handles RCP/SSP scenarios, of course, they provide remarkable insights, but none of them see the decarbonisation by 2050. This is one of the main gaps in the EROI literature. There are also key methodological differences in addressing the current shortcomings of EROI estimation.

Having seen the directed questions as authors of this paper, an attempt was made to clarify this question. We would like to thank Reviewer #2 for indicating this issue. The following modification was made to the statement:

Even though the choice of these scenarios were primarily motivated by the availability of the required detailed data for the EROI calculation, they also provide **a representation of the variety** of discussed transition paths.

The key summary of scenario differences with their results are elaborated in “An overview on nine global energy transition scenarios” sub-section, and due to the request, we added a brief summary, but readers can still see detailed result and comparison in without in Aghahosseini et al.¹.

Additionally, scenario differences in Table 1 are revised. All these revisions are moved to the Results section to improve the readability of the paper. For further details, we believe that the best source will be Aghahosseini et al.¹ which is stated more than once in the paper.

Impacts of the energy transition on global EROI	
p.2, para 1, l.4 As previously mentioned, setting upper and lower bounds to the net energy cliff is questionable.	Kindly see the detailed explanation presented for comments related to p.1, l.4 for the Abstract section. We did not apply any upper or lower bounds; we simply give a reference to establish a fair comparison so as to show that the scenarios avoid risks safely as detailed in there. The “upper limit” in this spot is for the energy cliff, not EROI, which could serve as appropriate reference to identify the risk zone for systemwide EROI.
p.2, para 3, l.1 It is difficult for the reader to understand how the results differ from one scenario to another without first presenting the energy mix, energy demand and socio-economic dynamics by scenario in more detail (although these can be retrieved from the SI and the corresponding literature). A detailed reference to the corresponding section in the Data and Methods section should suffice.	We agree with the Reviewer. As the reviewer noted, there is substantial data to be discussed in connection with this paper. Luckily, the energy transition scenarios are already published (Aghahosseini et al.¹). Thus, we opted to using Aghahosseini et al.¹ together with a summary of the fundamental differences and key findings of the scenarios in the Result section, and extensive regional results are detailed in the SI. However, based on your feedback, we partly expanded the subsection “An overview of nine global energy transition scenarios”. As previously mentioned, Table 1 is also changed to emphasize the key differences in scenarios and is moved into the Results section. The electricity generation mix given in SI Fig. S7 is also moved into the main paper. Beyond this, we believe that the Results section should focus on the EROI analysis, of course while showing in connection with the modelling results. The main source of the modelling outputs (Aghahosseini et al.¹) is emphasized more than once in the main paper and the general outlook of electricity generation profiles, future installed capacities, and diversification of energy technologies are also included in the SI of this paper.
p.2, para 3 What about the deployment of carbon capture and storage (CCS) when these technologies result in a significant net decrease in energy? See Fajardy, M., & Mac Dowell, N. (2018). The energy return on investment of BECCS: is BECCS a threat to energy security? Energy & Environmental Science (Vol. 11, Issue 6, pp. 1581–1594). Royal Society of Chemistry (RSC). https://doi.org/10.1039/c7ee03610h	As indicated in the main paper, the cumulative energy demand (CED) values of CCS integrated into fossil-fuelled power plants (CCGT and hard coal) show a slight increase. Fossil CCS is mainly used in IEA scenarios and have a low capacity compared to other fossil fuel technologies. Therefore, their impact on the overall EROI is negligible. BECCS are not part of the LUT-ESTM technologies. Recently published results further emphasize that BECCS has a multitude of issues, so that a growing number of researchers decreases its relevance. A central aspect is the limited sustainable bioenergy, among others. https://www.nature.com/articles/s41558-023-01697-2
p.2 para 3 + 5 These paragraphs lack, in my opinion, an effective purpose. I advise the authors to take back	We fully understand Reviewer #2's concerns. It might not be necessary for any researcher specializing in EROI analysis to include these paragraphs; however, energy modelling studies have

in particular the end of paragraph 3 and the whole of paragraph 5 which do not seem to me to be superfluous, but less important with regard to the rest.	been continuing to include EROI analysis. We modified paragraph 3 by removing the part suggested by Reviewer, but we would like to leave most of the text in the 5th paragraph for the benefit of other readers except the exclusion of its last statement. Here are the excluded parts: Page 2, paragraph 3: The output variables of LUT-ESTM are (i) installed capacities (segregated as newly installed, decommissioned, gross, and cumulative on technology-level), (ii) annual electricity generation (total electricity demand and final electricity consumption), (iii) full load hours of technologies, and (iv) lifetimes of technologies, and are embedded into the LUT-EROI model as main input parameters in equation (2) and (3). Page 2, paragraph 7: Further details about energy consumption and generation profiles, future installed capacities, and the diversification of energy technologies are provided in the SI Note 4.
p. 2, para 5 I also cannot find the reference 33 “Sahin, H., Solomon, A. A., Aghahosseini, A. & Breyer, C. The impact of spatial representation in energy transition modelling on system-wide energy return on investment. (2023).”	This paper analyses the impact of spatial representation on systemwide EROI using Europe as a case study. As mentioned, this paper is still under review and is not public. The paper was also cited to only recognize the present of data regarding the uncertainty related to spatial representation of the model. At the same time, the analysis was presented at the 17th SDEWES Conference Paphos 2022 by the same authors. The conference abstract of the previous analysis is attached along with the Revised Manuscript for the Reviewers.
p.3, fig. 1 The figures are really interesting but I am afraid that they are not of sufficient quality in dpi. Figures (f) and (g) are also a bit blurry and difficult to interpret.	The figure is updated according to the Reviewers’ comments, and we increased the 900 dpi. The figures in the manuscript also provided as a separate file for Reviewers. The compression of the submitted file for the review process may be a challenge, hopefully not.
Effects of PV – storage - curtailment paradigm on the global EROI trends	
p.4, para 3, l.9 I am surprised by the authors assertion that “nuclear power [...] achieves a lower EROI performance compared to wind and solar PV” as this goes against the emerging consensus. I would like the authors to explain why, and carry a brief analysis on how their EROI numbers compare to the existing	CED values are subject to change based on system boundaries, technology types, share of the market, and data. In this analysis, we calculated nuclear CED values based on ecoinvent sub-datasets. CED values for different types of nuclear power plants (PWR, BWR, EPR, AP 1000, SMR, AHWR, GT-MHR) are derived. Their capacities are segregated according to the current situation: construction, operation, and future planned ones. Then using capacity shares, we estimated a reference CED value for the nuclear power plant. CED values through its lifetime are estimated by summing the reference

literature, taking for instance Murphy, D.J.; Raugei, M.; Carbajales-Dale, M.; Rubio Estrada, B. Energy Return on Investment of Major Energy Carriers: Review and Harmonization. Sustainability 2022, 14, 7098. <https://doi.org/10.3390/su14127098> as a reference.

CED value (structure CED value), and fuel CED value (energy invested in the fuel chain multiplied by full load hours and lifetime, i.e., operational CED value). The founded total CED value is then divided by the nuclear power plant lifetime to get an annualised energy investment, which is later on again multiplied by the nuclear capacities of that scenario. The technology-specific data of the PWR nuclear power plant was used to estimate the CED value of the nuclear fuel since its share of total nuclear capacity corresponds to over 80% in 2050. The relative inventory is provided in the SI in Supplementary Note 3 in Section 3.6. All estimates related to operational aspects of LUT-ESTM and fuel CED do not contain the primary use of uranium (as resource) and is not included in all calculations. Coming to your question, nuclear EROI could be as high as 30 and lower than 10 depending on technology and fuel enrichment techniques, our estimation is based on global average quantities. Thus, comparison should keep that in mind.

As regards to our overall analysis, the presented systemwide EROI analysis approach in this paper does not use static EROI values since they change every 5 years depending on the technology mix, etc. For instance, in the IEA-SDS scenario, where nuclear is more preferred compared to other scenarios, the EROI of nuclear power plants stays around 17, on the other hand, for PV, the EROI values significantly change. As presented in the following table, the renewables EROI values are increasing due to (i) advancement in the technology, which is presented by energetic learning curves for PV, (ii) energy-intensive material consumption (e.g., steel, cement, etc.), and (iii) energy invested in the fuel chain (without fuel energy content). For PV, wind, and hydro, energy invested in the fuel chain is zero, therefore, does not produce operational CED values. However, the operational CED value is affecting the overall CED value during the lifetime of nuclear power plants, and even though 0.2 MJ_e/kWh is quite a conservative value compared to fossil fuels that is expected to increase in the future.

We would like to remind that we do not follow the traditional technology-based EROI calculation. The presented systemwide EROI calculation covers all technologies and have system level boundary conditions. However, a separate technology-by-technology EROI calculation was roughly performed to check the credibility of our results and to detect any errors in the technology-level CED estimates.

Table 1. Technology level EROI estimation of IEA-SDS scenario, including renewable and non-renewable technologies

Technology/Year	2015	2020	2025	2030	2035	2040	2045	2050
PV fixed tilted	11.7	14.0	16.3	19.5	22.3	24.8	26.7	28.5
PV prosumers	11.5	12.1	15.6	17.6	19.6	24.8	31.0	32.7
Wind onshore	44.8	46.5	39.6	42.0	44.5	46.0	47.6	49.8
Wind offshore	50.9	62.1	54.4	59.7	65.0	68.4	72.8	77.5
Hydro run-of-river	57.5	57.7	51.3	51.7	52.1	52.4	52.7	52.9
Hydro reservoir (dam)	94.1	94.6	84.1	84.8	85.4	85.9	86.4	86.7
Biomass solid	12.9	13.2	12.9	13.0	13.0	13.0	13.0	13.0
Geothermal electricity	4.2	9.0	6.8	7.0	7.1	7.3	7.3	7.4
CSP ST	33.3	58.0	17.7	19.7	23.7	24.6	25.0	25.3
ST	0.0	0.0	0.0	0.0	0.0	0.0	0.0	0.0
Wave	0.0	0.0	0.0	0.0	0.0	0.0	0.0	0.0
CCGT	12.5	12.0	11.9	11.6	11.3	11.0	10.8	10.5
CCGT +CCS	0.0	0.0	0.0	0.0	0.0	0.0	0.0	0.0
OCCGT	12.6	11.8	11.4	11.1	10.9	10.7	10.3	10.1
ICE	12.2	10.5	10.9	10.4	9.9	9.5	9.0	0.0
Multifuel ICE	0.0	0.0	0.0	0.0	0.0	0.0	0.0	0.0
Coal power plant	24.2	23.6	22.2	21.3	20.4	19.2	0.0	0.0
Coal power plant +CCS	0.0	0.0	0.0	0.0	0.0	0.0	0.0	0.0
Nuclear power plant	17.2	17.3	17.2	17.3	17.3	17.3	17.3	17.3
Biomass CHP	13.1	13.2	11.6	11.6	11.6	11.6	0.0	0.0
Waste-to-energy CHP	51.0	53.5	55.5	55.5	55.5	55.5	55.5	55.5
Biogas CHP	13.3	14.0	13.9	13.9	13.9	13.9	13.9	13.9
Methane CHP	103.3	77.7	100.8	105.4	84.4	69.1	54.8	0.0
Oil CHP	10.6	8.5	10.0	9.6	9.0	8.9	8.1	0.0
Coal CHP	22.6	20.9	22.1	21.5	21.0	20.5	0.0	0.0
Fuel Cell CHP	0.0	0.0	0.0	0.0	0.0	0.0	0.0	0.0

p.4, para 3, l.17 The last sentence is quite obvious, you can delete it.

It is deleted.

Significant variations in EROIs due to regional differences

p.4 I think the authors' findings have implications for regional inequality in the transition. An interesting additional analysis could explore (or at least mention) this. Sensitivity of the global EROIs to the embodied energy requirements for natural gas and oil sources

Thanks for the nice comment. We agree that regional inequality in transition should be analysed from the EROI perspective, which will be presented in another article that we are currently working on. However, due to massive data and discrepancies in both the energy mix and the capacities and levelized cost of electricity (LCOE) of the regions, it is difficult to go into such detail and explain

	the results of 9 regions together with the global ones in this paper. Thus, a key summary of regional EROI trends is given in this paper.
p. 4 This section provides really interesting analysis which, I believe, should be further highlighted per region.	Unfortunately, our regional EROI analysis is still ongoing. The typical trends are what is given in this paper. Thus, we believe that such a summary will suffice for the readers as the focus of this paper is on analysis at a global level. Thank you again for your interest in our research and your constructive comments. Because of the detail in the massive data that required additional analysis to present detailed regional differences and its link to EROI, we can only promise to return to this subject in another paper.
Discussions	
p.5, para 1, l.18 Beware that all scenarios are only feasible from a technical point of view, and that the drop in the systemwide EROI leads to consequent adverse economic consequences. See Jacques, P., Delannoy, L., Andrieu, B., Yilmaz, D., Jeanmart, H., & Godin, A. (2022). Assessing the Economic Consequences of an Energy Transition Through a Biophysical Stock-Flow Consistent Model. In SSRN Electronic Journal. Elsevier BV. https://doi.org/10.2139/ssrn.4174917 in this regard. You may also rely on Rye, C. D., & Jackson, T. (2018). A review of EROEI-dynamics energy-transition models. In Energy Policy (Vol. 122, pp. 260–272). Elsevier BV. https://doi.org/10.1016/j.enpol.2018.06.041 (which you use in the SI) and Jackson, A., & Jackson, T. (2021). Modelling energy transition risk: The impact of declining energy return on investment (EROI). In Ecological Economics (Vol. 185, p. 107023). Elsevier BV. https://doi.org/10.1016/j.ecolecon.2021.107023	Thanks for your constructive comments. All references are added along with the additional statement in the discussion section while highlighting these issues. The revision is executed adding a short statement in the Discussion part: Note that taking all sustainability aspects into account, a steep drop in the systemwide EROI might causes irreparable economic consequences⁴⁶⁻⁴⁸. References numbers: 46. Rye, C. D. & Jackson, T. A review of EROEI-dynamics energy-transition models. Energy Policy 122, 260–272 (2018). 47. Jackson, A. & Jackson, T. Modelling energy transition risk: The impact of declining energy return on investment (EROI). Ecological Economics 185, 107023 (2021). 48. Jacques, P. et al. Assessing the economic consequences of an energy transition through a biophysical stock-flow consistent model. Ecological Economics 209, 107832 (2023)
p.5 I would really like the authors to compare their findings with the work of Slamersak et al. and King and van den Bergh (2018), who estimated a 10-34%/24–31% reduction in net energy per capita for	As requested, we have compared our results with the outcomes of these studies in the Results section. However, we would like to remind that these studies differ from ours in terms of approach, methodology, modelling differences as well as the way they interpret the EROI for an energy system. The added paragraph to the manuscript is also given in below.

the IPCC 1.5°C SR/IEA low-carbon transition pathway, respectively.	Low-carbon pathways generated by IAMs differ from cost-optimisation energy modelling in terms of methodology, and conceptualisation of the scenarios, and could yield different EROI outcomes. Together with the divergence of our EROI methodology, developed by Solomon et al.^{27,29}, a careful comparison is necessary. The typical results of our study are consistent with the findings of King et al.¹⁹ that applied IAMs to investigate EROI and indicated a decrease in net energy for society. The technology level EROIs for solar and wind technologies are increasing over time while the reverse situation is observed for fossil fuelled power plants¹⁹. At this point, gas and nuclear power plants mostly maintain their low EROI values compared to renewables or show minor change over years. This is also founded in the study by Sers²⁴ according to which the EROI of fossil fuels shows a precipitous drop by 2050 due to the depletion of reserves. Additionally, the renewable EROI has been set at 40 and increases as the cumulative greenhouse gas emissions decrease²⁴. However, in this study, the EROI of renewable technologies varies depending on operating patterns, energy learning rate, and the storage technologies reduce the systemwide EROI, in contrast to the study of Sers²⁴, where storage EROI is estimated separately from renewable EROI. Another study presented by Slameršak et al.²⁵ notes that the early years of the transition could result in an increase in GHG emissions due to more fossil fuel usage. This partially overlaps with our findings in an indirect way. By shortening the ET period, the systemwide EROI will be drastically reduced due to the deployment of renewable power plants and complementary technologies. This study does not apply energy flow techniques as in the case of PROI, but this data suggests that construction phase energy consumption may present a challenge for those scenarios. We leave such matters to further studies. Note that IAMs studies discussed here aim to model the entire energy system, rather than just focusing on the power sector, and follow a broader energy modelling framework, including socio-economic relationships, indirect GHG estimates and energy efficiency implications, that differs from the study of Aghahosseini et al.⁶ as regards to the contextualisation of scenarios and modelling approach.
Data and Methods	
An overview of nine global energy transition scenarios	
p.5, para 1, l.11 I feel like the socio-economic conditions of all scenarios need to be critically compared, as the information provided here are not sufficient to estimate how they compare between each other and relate to SSPs.	This section is moved to the Results part, and Table 1 is revised considering your comments. As mentioned previously, LUT-ESTM is not an IAM, so it is structured with the purpose of achieving net-zero power systems by 2050. The scalability to cover all SSPs (including socio-economic challenges and factors) is smaller and restricted as in other cost-optimization tool. Kindly see the previous explanation for this part (p.1, para 1, l.21). We are thankful to Reviewer#2, for pointing out this issue.
CED database for LUT-EROI	
p.6, para 2, l.5 Not having access to the ref. 27 is critical here as I am unable to check how the uncertainty of	Unfortunately, the study of Solomon et al.⁸ (Ref. 27 in the manuscript) is still being reviewed in a recognized journal. Due to its novel approach, we cannot present it as a preprint. However, the

estimating primary energy at the same energy quality has been solved (i.e., applying conversion factors).	relevant information on unifying energy quality, and the process to implement it in the LUT-EROI analysis are explained in detail in both the main paper and the SI. Estimating primary energy quality at same energy level does not introduce uncertainty, which should be logically clear. Unfortunately, the traditional practice does not unify energy quality and that introduces uncertainty. We cite the paper because the superiority of unifying energy quality over any of the traditional approaches was discussed there. But it is subject of several studies that the approach introduces error, the suggested solutions are pragmatic fixes.
Supplementary Information	
I have trouble figuring out the share of several factors (increasing recycling capabilities, technological improvements, etc.) in the evolution of the EROI of the technologies. Could the authors elaborate and indicate how these results are challenged?	Recycling of the materials and wastes are not considered in this study due to a lack of data related to (i) the requirement of specific data for each material on how much of them can be recycled, (ii) the recycling process can be done in several methods, which also require energy consumption and should be included as indirect energy consumption. To provide more information, we added a few statements and a new sub-section in the main paper, touching on these issues. These statements are too long to be put in this file, thereof, we suggest the Reviewer to check the manuscripts methodology section. Additionally, technological improvements are explained in the SI as an adaptation of CED values (also given as separate tabs in the CED database Excel document). In our overall research, doing a serious work in accounting impact of recycling in systemwide EROI will require a substantial time of research. An approach applied to a single technology will not help to understand system level issues. We would like to take this opportunity to emphasize that the systemwide EROI and technology level EROI has significant differences in terms of the level of detail. Systemwide EROI is the one critically needed. This paper is the first detailed work, which has explored the subject while addressing existing shortcomings in the area of EROI and the vast data requirement by bringing expertise from various field of studies into one. We do not expect it to solve everything at once. We believe that we present serious piece of work for your evaluation.

Reviewer #3	Response to Reviewer #3
Reviewer comments:	Thank you for your valuable comments and suggestions. Kindly note that the changes are already reflected in the revised manuscript via the track change module. However, for the sake of clarity and to save time, we have also indicated the direct changes requested by the reviewers in apostrophes in the blue sentences. The words/phrases in bold and underlined refer to specific changes. Apart from the moderate/minor changes requested by the reviewers, the major ones are presented in the revised manuscript.
The EROI over time of the electricity sector for a set of 9 future scenarios. This type of result is fine. This paper cites mostly sufficient studies, and I've mentioned a few more that provide insights into some of my feedback. The authors should also discuss how their study relates to a recent Nature Communications paper by Aljoša Slameršak et al (2022) that also calculates economy-wide EROI over time for some scenarios: "Energy requirements and carbon emissions for a low-carbon energy transition," https://www.nature.com/articles/s41467-022-33976-5	We are thankful for the comment. As requested, we have compared our results with the results of these studies in the Results section.
Major Comments #1	
It is not clear to me that this methodology represents the full dynamics of EROI in relation to when installation occurs, causing the majority of CED at the beginning of the life of most renewable technologies, versus when electricity is generated relatively uniformly over the lifetime of the technology. I take your equation (2) as representing the calculations you are plotting in Figure 1. However, my interpretation of the method is that it is calculating a value every 5 years that is not a number we should want to know because it provides a confusing interpretation about the state of the energy system for any given year. The method	Thank you for this comment. First, this study does not follow annual energy flows, which is used in PROI estimations. Additionally, EROI is not the same as PROI. Mathematically, the same EROI value can be calculated using two approaches. First approach is the well-known lifetime approach but that applies to some reference year. The second one is the annual approach by assuming static energy input per year with reference to the specified year. When dealing with the dynamic energy transition, we run into some inconsistency to apply lifetime technique. For example, the lifetimes of technologies are different thus it is difficult to get the system level lifetime to calculate lifetime system energy output, etc. Thus, the annual version of the calculation solves several such challenges. Thus, we applied that appropriately. Therefore, EROI for year 2030 takes all operational technologies (new and old) by then and the corresponding static energy input of the full system. Thus, the calculation is methodologically consistent at all years. The EROI result will not be affected by the modelling time step, be it 5-year or 1-year.

seems to be calculating “For a given year, e.g., 2030, the weighted average lifetime EROI of the technologies that are installed in 2030 is EROI = X.” which is specifically different from calculating “For a given year, e.g., 2030, the electricity system has an EROI = X calculated by dividing annual electricity output by annual energy inputs.”.	The 5-year time step is applied because traditionally the energy transition modelling follows such a large step. Because this methodological detail and associated justification was presented in another paper, which is under peer-review, but we provided a short but sufficient summary in this paper. Unfortunately, that may have partly contributed to this confusion. We added further explanation regarding our methodological approach and responded to the remaining part of this comment below. We also added the following statement that shows that PROI and EROI are to distinct tools in the main paper. The systemwide EROI was implemented by covering all technologies by a given year (new and existing). We simply don't calculate the EROI of some technology and then use that EROI in the energy mix to determine the overall EROI for a system, which is common in EROI literature. Instead, we create a boundary condition that fits to the system as one system, which is challenging and data intensive. Thus, we generated structural CED values (construction and decommissioning phases) and operational CED (using full load hours, lifetime, and fuel CED values) for each technology. Then, we estimated the CED requirement through the lifetime of technology, and then we annualised this estimated value by dividing technology lifetimes. For key technologies, this value shows the difference every 5 years. And they are later multiplied by capacity values taken from modelling outputs (considering the old/existing capacities, newly installed capacities, decommissioned capacities). After the calculation of the total energy invested for the overall system at the given year, the estimated total electricity generation for those years is divided by the total energy invested for the overall system and finally, we have systemwide EROI values for every 5 years. Our methods rely on a cumulative approach rather than using one-year capacities and specific technology EROI values.
a. Your Equation (2): $EROI_{sys,year} = \frac{SG_{year}}{\sum_t \frac{(\sum_{n=1}^{N_t} E_{inv_{t,n}} \cdot Capacity_t)}{LT_t}}$	As we tried to point out above, EROI lifetime takes lifetime output in the numerator and lifetime input in the denominator. Because we apply the annual approach, the formulation should be exactly as given in Equation (2). Referring to equation (2), if we don't divide the denominator by LT. We should multiply the numerator by system LT (as indicated above, we don't have that number) to get the EROI. It may be difficult to see at system level, but we assume the following technology level

b. Equation (3) defines E_{inv} as the entire lifetime energy investment, since it adds $E_{capacity}$ to $E_{operation} \cdot FLH \cdot LT$.

c. Thus, the denominator of Equation (2) is = (total lifetime energy of existing generation fleet)/LT = annualized energy investment regardless of when the energy investment occurs.

d. Thus, since the denominator of Equation (2) divides E_{inv} by LT, it is dividing the annual generation (SG/year) by a quantity of energy investment that does not relate to a quantity of invested energy (to date) that provides an insight that is useful.

There are at least 2 problems with the approach of Equation (2):

i. Construction energy investment: Equation (2) is calculating the energy investment of construction by the lifetime of the plant, but this is not an accurate description of what is actually happening. 100% of the construction occurs at the beginning and it is not spread over the life of the project, so this makes the EROI calculation seem higher than what comes from an accurate (I think) definition of the input dynamics. This is not a realistic representation of real-world energy flows. IN SHORT: This biases EROI upward for all years (I suppose until decommissioning).

ii. Decommissioning energy investment: Equation (2) is dividing generation in a current year by energy investment for decommissioning that has not yet occurred. This is not a realistic representation of real-world energy flows. IN

mathematical formulation may clarify the equivalence of the lifetime and the annual approach to estimate EROI.

$$EROI = \frac{EO}{EI} = \frac{EO_{an} \cdot LT}{EI} = \frac{\left(\frac{EO_{an} \cdot LT}{LT}\right)}{\frac{EI}{LT}} = \frac{EO_{an}}{\left(\frac{EI}{LT}\right)}$$

where EO – lifetime output, EI-lifetime input and the subscript “an” represents annual values.

As noted in the above comment of the reviewer and the one coming below, the reviewer assumes that EROI requires to account for real world energy flow. The two algorithms (EROI and PROI) are different but unfortunately many literatures erroneously interpret them as one thing despite their fundamental difference (see further comment below). Thus, we would like to request to evaluate our work as distinct from PROI.

We are aware of each issue raised by each reviewer and have provided appropriate solutions to all issues beforehand. Unfortunately, we must define a scope and limit our writings in that scope to present our material in a logically manageable way, both for us and readers, given the challenge that we are dealing with. We would like to give the following summary on the difference of our work and PROI.

- The provided equation presents one of the two ways of calculating EROI. The study does not rely on annual new capacities, but on cumulative capacities at every 5-year step (i.e., all operational technology capacity at a given year are included). In addition, the LT of certain technologies is changing for certain periods.
- Also, we would like to emphasize once again that the presented methodology is not related to the PROI.
- PROI follows an energy flow analysis while EROI relies on impact assessment (ISO 14040⁹ and ISO 14044¹⁰).

The major difference between PROI and EROI lies in identifying temporal and spatial boundaries as pointed out by Carbajales-Dale¹¹ and Murphy et al.¹².

In the studies of Carbajales-Dale¹¹ and Murphy et al.¹², net energy analysis (NEA) is engaged with PROI, which is generated using energy balances by input-output approach and/or stated technology-

SHORT: This biases EROI downward until decommissioning.

level EROIs in the literature. At the core of conceptualisation, it becomes an energy flow analysis rather than an impact assessment as it tries to incorporate the typical flows presented in Fig.1 given in the study by Carbajales-Dale¹¹. On the other hand, for our EROI estimations, we rely on life cycle inventories (LCIs) - basically stay on the impact assessment side - (energy requirements for material/substances consumptions, specific processes (drilling, excavation, transportation)) and did not use any energy balances as well as energy flows. Thus, our approach to EROI estimation differs from their PROI perspective. At the same time, our approach complies with the basic concept of EROI, which focuses on understanding how much net energy a given energy technology or system can provide over its lifetime (as compared to the total energy input). Thus, PROI should be used for purposed other than the one EROI is designed for as the two tools have a different meaning and approach (see below).

Figure 1. Energy investments (red) and outputs (green) for an energy project over three life cycle phases: (1) construction, (2) operation, and (3) decommission explained by Carbajales-Dale¹¹.

We know that the energy invested may change each year depending on location, materials and sources consumed, and processes (drilling, excavation, etc.). However, taking the analysis to system level on a global scale requires massive and tailored data, which currently does not exist in LCA databases. Thus, calculating system level PROI at global level considering proper energy flow could be challenging due to associated uncertainties despite its irrelevance for the stated objective of this

study. Thus, though we understand the interest on the side of the reviewer, we believe that such subjects should be left for future studies.

Summing up, PROI and EROI should not be used with the same meaning because they don't deliver equivalent values. PROI calculations also change with time even for the same technology due to the consideration of energy flow. And, thus, follows its own specific meaning and interpretation. The case of systemwide PROI is even more complex as stated above. Generally, PROI may fit more as a tool of evaluating possible planning bottlenecks. Also, primary energy quality issues remain an unsolved issue for PROI. However, interpretation of the result could even be challenging as values may face significant uncertainties. Thus, the two indicators (EROI and PROI) should be separated but not replace each other. Yet, systemwide PROI should be left for a future study as we see more challenges in the evaluation and interpretation. The goal of this study is to evaluate the sustainability risk due to system evolution, to which the use of the physical EROI fits correctly.

I will point out: Many papers calculate EROI of a system as = (annual energy output)/(annual energy invested). Some call this “PROI” (power return on power invested), and I think providing this calculation would add insight. i. My understanding (you can correct me) is that you are NOT attempting to do this calculation of “PROI”, and this is OK, but you are calculating an “annualized PROI-type” metric since it has units of (power output)/(annualized power input) = (energy output per year)/(annualized energy input per year) rather than (energy output per year/energy input per year).

I am indicating the issues laid out in these two (among other) publications:

i. [You already cite this]: Murphy et al. (2011) Sustainability (https://www.mdpi.com/2071-1050/3/10/1888) Figure 4 showing that CED is the integral of power flows each year, and you can see in Equations (1) and (2) of this paper that the authors define the acronym EROI using two methods. Their Equation (2) is essentially “PROI”.

King et al (2015) Energies Part 1 that explicitly shows the connection between “power return ratios” and “energy return ratios” (https://www.mdpi.com/1996-1073/8/11/12346) to avoid the confusion. Figures 1-4 show how this plays out for a simple system.

As we pointed out above, we are aware of this literatures and associated issues, but we wanted to avoid the subject because the associated confusion requires a separate treatment. It is confusing and challenging to compare these approaches and again present the analysis of such a large dataset in a single paper. And yet, EROI is not the same as PROI given the facts that we described above. Thus, we added the following clarification in our revised manuscript:

The above calculation should not be confused with the power return on invested (PROI) metric, which enforces an energy flow technique and yet get confused with EROI that implements no energy flow ^{14,15}. The annual approach implemented in this study is a direct mathematical equivalent to a lifetime EROI calculation.

We leave the rest for future research, but we would like to note two things regarding these references. First, these analyses are mostly technology level. Second, they follow traditional approaches which at least differ from our work in their treatment of energy quality, for which definite reference energy quality is a must as in our study. Our study is systemwide and uses energy modelling result thus includes various issues related to systems design and operation. Thus, in some sense comparison may not be appropriate. Technology level EROI estimation in our study also exist but it is only used for validation purpose and/or detect errors. But definitely, we know system level values differ from technology level estimates.

My interpretation of your denominator of your Equation (2) is that the inputs if represented in Figure 4 of Murphy et al (2011) and Figures 1 & 2 of King et al (2015) would simply be a constant value for each technology you add to the system, rather than fluctuating from construction-operation-decommission phases over a technology lifetime as indicated in those publications.	First of all, as also indicated above, our denominator includes all technologies that are operational on that year. Thus, includes old and newly added technologies. In short, we treat the system as one. Second, denominator input can be seen as a static if we take one reference year. But if we take various years, the input energy related to a given technology may vary because several things change with time. For example, there is energy learning for some technologies, lifetime change, dispatch change, etc. Technology capacities (considering both new and old, and decommissioned capacities) change and CED values also change over time. In short, the model is designed to follow the system evolution in all its consequences and opportunities. All those assumptions are in the SI. Yet, the changes are not related to the energy flow. Even though, we treat the system as one. To simplify the discussion, let us take a specific technology X, which may have A old capacity, B new capacity at a specific year. Any decommissioned capacity will be excluded as it is not operational on that year. The corresponding input energy will be the sum of the value for A and B. For the old capacity the initial input energy may need to be updated depending on its dispatch, the new capacity will follow the condition at the specified year. Thus, the model follows a mixture of these factors and create an aggregate input relevant to the year of interest taking all technologies. A simple way to understand this approach is that each year's system will have its own EROI value depending on that system composition and dispatch. It also has some lifetime depending on that system composition but difficult to estimate. Thus, if we keep the same system and operation over that lifetime, the system EROI and thus denominator remains constant. But since the system continuously change both will change.
To me, the most accurate representation of the system-wide EROI calculation you are studying is two use both of the following approaches:  i. (X) “PROI_{system}” = (annual energy output)/(annual energy invested). ii. (XX) EROI_{system} = (integral of annual energy generation to date)/(integral of annual energy investment to date) iii. Since the system of “the grid” does not have a lifetime (i.e., we expect it to continue 	We are grateful for reviewer #3's suggestions. As we have explained in previous comments, our approach is the EROI calculation for the whole system. We agree that the energy system is continuously expanded. But that is a different concept from the lifetime perspective. The system is based on a composition of technology that has a limited duration of operation. Thus, has a lifetime but the lifetime changes from one year to another because the system composition continuously changes. Just because humans reproduce, we don't say there is no life expectancy. That is an analogy for energy system lifetime. Thus, EROI calculations should be related to lifetime if we want to maintain the required scientific meaning and its quality to bear scientific meaning. Coming to the suggested tools:

indefinitely), both approaches avoid the assumption of how a power plant lifetime affects the energy input by simply counting the energy inputs and outputs each year. The system is simply composed of parts that continuously generate electricity (or get decommissioned) and continuously have energy investment, and you track this over time. So including the value of the lifetimes (e.g., 30 years) of individual plants in your Equation (2) becomes unnecessary. iv. Yes, there are inherent difficulties with the approach (XX), primarily understanding the initial conditions (this is OK for me). But your current methodology already needs to explicitly describe how you calculate the initial EROI calculation for your first year of 2015 and/or 2020 (since it seemingly depends on the EROI of all technologies that exist now, as based on when they were installed) and maybe this becomes obvious after you address my comments on your Equation (2) more generally.	PROI system (X)- we are not implementing energy flow thus this cannot be applied to our study. If the energy flow is not considered for a given year as we do, then it should be named $EROI_{system}$ because that is exactly the same as lifetime EROI. $EROI_{system}$ (XX) – the suggested calculation does not qualify as an EROI. It is a new concept. Thus, it should be first tested and verified so that it can provide the same scientific meaning as the lifetime EROI. But testing that is not easy because the data required to do what the reviewer is suggesting is quite massive. Because this algorithm requires the same data quality that we briefly discussed above when we discussed the issue in relation to PROI. And that data does not exist in LCA databases because LCA is lifetime based. Even if we go for simplifying assumptions, the likelihood of the results showing unpredictable variations due to data quality is high. Thus, year to year comparison could be challenging due to that assumptions. Yet, this cannot be interpreted as an EROI without verification. The difficulty that the reviewer mentioned in relation to this approach is the simplest if the method really had the suitable meaning as an EROI. Thus, further research is required to verify his suggestion and it should be done outside this paper. The real-world energy system has continuously changing lifetimes, thus a proper interpretation of XX as an EROI can be challenging without some discretising. While various time ranges could be possible for discretising, the most logical and simple step is again annual. That comes back to the methodology applied in this paper. Other time steps should be left for future study. While we appreciate the reviewer for his comment, we have to leave these issues as a subject for future research.
Major Comments #2	
A further analysis of the EROI relation with system levelised cost of electricity (LCOE) (Fig. 1j) shows that low-cost solutions correlate to low EROI.” Per the units of LCOE (money input/kWh output) and EROI (kwh output/energy input), they are inversely related, so that we might normally expect high EROI and low LCOE, and vice versa. Example see,	We are grateful for Reviewer #3's comment on this issue. Thus, we've revised the associated graph (2D) and label the data points. Also, a brief explanation is added in the Result section. The revision is as follows: A further analysis of the EROI relation with system LCOE (Fig. 2h) shows that low-cost solutions correlate to low EROI. Replacing fossil and nuclear energy with renewable energy reduces the system's

https://www.mdpi.com/2071-1050/3/10/1810, applied to U.S. King and Hall (2011) oil and gas to oil prices (not a levelized cost, but a price) but similar idea. But you are showing declining EROI can be associated with also declining LCOE. Also, I think your Figure 1(j) can simply be a 2-dimensional plot of EROI vs. LCOE, and perhaps label each data point with a year (or the first and last data points with the first and last years so we can see how any given trajectory changes over time).	LCOE by lowering fuel and CO₂ costs. Also, the level of reduction in the LCOE and EROI is directly correlated to the system composition and the choice of technology.
REQUEST: Please explain in more detail why you see this opposite trend (from what I'm stating) as you state in the text. This answer might come from addressing my first "main" comment.	The fundamental reason is in the system dynamics, which makes it much different from the situation of oil industry. The system is constructed based on various technologies that enjoy different EROI values (e.g., high EROI for wind and hydro, low for PV). At the same time, when the energy transition proceeds the system adds more solar and enabling technologies such as batteries. The impact of EROI is increasing in the earlier years of the transition, then it switches trends and decline as enabling technology increasing. But at the same time, LCOE decreases because the capital cost and operational cost decreases as the energy system goes more renewable. Due to these dynamics, the trend may not be the same. But it is good to bear in mind that this study is based on modelled future circumstance and LCOE (not price). The study of King and Hall (2011) is based on historical data.
QUESTION: In each year, do you adjust your EROI and LCOE calculations based upon the capacity factor of each power plant or category of power plants (i.e., since I presume capacity factor of coal and/or natural gas plants go down over time in your scenarios)? Perhaps you can see the conundrum here, the same power plant might calculate an EROI of 20 in year 2025 with capacity factor 90% would have a lower EROI calculation if using its dispatched capacity factor of 50% in 2040.	Systemwide EROI and LCOE are system level, thus it follows system level condition on that year. Yes, we do not adjust system level EROI or LCOE calculations for technical differences related to power plants. These technical aspects are already taken into account in the modelling phase, as explained in Aghahosseini et al.¹. All capacities, FLH, and LCOE are the outputs of Aghahosseini et al.¹. And we used these outputs as the inputs of the systemwide EROI calculations. Also, we would like to highlight that we do not follow the traditional technology-based EROI calculation. Our systemwide EROI calculation covers all technologies due to the reasons we stated before treats them as one. However, a separate technology-by-technology EROI calculation was roughly performed to check the credibility of our results and to detect any errors in the technology-level CED estimates. In those technology-level calculations, some technology EROI may change depending on its dispatch. But we only use those technology level data for validation and/or detecting

errors. Giving independent technology level EROI without enabling technologies are misleading. This is why we did not present it separately.

As already mentioned, the results of the modelling are used to estimate the EROI. As given in below, Table 1 refers to the technology level EROI values of the IEA SDS scenario for generators, which are roughly estimated. Table 2 refers to the IEA STEPS scenario and again contains EROI estimates at the technology level. Table 3 represents the sensitivity analysis for IEA STEPS scenario where fuel CED values are changed.

A comparison of Table 1 and Table 2 shows that dispatching and technology mix are the main causes for the change in EROI. However, a low systemwide EROI does not mean that the EROI at the technology level should also be low. For the IEA SDS scenario, the EROI increases after 2025 as fossil fuel consumption decreases and this gap is filled by solar PV, nuclear and gas technologies. For example, the EROI of solar PV increases, but considering together with enabling technologies (e.g., battery) can pull down the technology-level EROI at a lower level. Therefore, technology interactions, disposition, and technology selection are critical to EROI calculation, which also supports the holistic concept (a systemwide EROI) is necessary.

Looking at Table 2 and Table 3, although we used the same capacities, FLH and methodology, the fossil fuel EROI is very sensitive to changes in the CED of the fuel, which represents the upper production chain (production to transport to power plants) without considering the energy content of the fuel.

Table 1. Technology level EROI estimation of IEA SDS scenario, including renewable and non-renewable technologies

Technology/Year	2015	2020	2025	2030	2035	2040	2045	2050
PV fixed tilted	11.7	14.0	16.3	19.5	22.3	24.8	26.7	28.5
PV prosumers	11.5	12.1	15.6	17.6	19.6	24.8	31.0	32.7
Wind onshore	44.8	46.5	39.6	42.0	44.5	46.0	47.6	49.8
Wind offshore	50.9	62.1	54.4	59.7	65.0	68.4	72.8	77.5
Hydro run-of-river	57.5	57.7	51.3	51.7	52.1	52.4	52.7	52.9
Hydro reservoir (dam)	94.1	94.6	84.1	84.8	85.4	85.9	86.4	86.7
Biomass solid	12.9	13.2	12.9	13.0	13.0	13.0	13.0	13.0
Geothermal electricity	4.2	9.0	6.8	7.0	7.1	7.3	7.3	7.4

OCGT	12.6	11.8	11.4	11.1	10.9	10.7	10.5	10.4
ICE	12.2	10.5	11.0	10.6	10.2	9.9	9.6	0.0
Multifuel ICE	0.0	0.0	0.0	0.0	0.0	0.0	0.0	0.0
Coal power plant	24.2	23.6	22.6	21.9	21.4	20.9	20.5	20.1
Coal power plant +CCS	0.0	0.0	0.0	0.0	0.0	0.0	0.0	0.0
Nuclear power plant	17.2	17.3	17.2	17.3	17.3	17.3	17.3	17.3
Biomass CHP	13.1	13.2	11.6	11.6	11.6	11.6	0.0	0.0
Waste-to-energy CHP	51.0	53.5	55.5	55.5	55.5	55.5	55.5	55.5
Biogas CHP	13.3	14.0	13.9	13.9	13.9	13.9	13.9	13.9
Methane CHP	103.3	77.7	95.2	94.2	98.2	101.9	105.4	0.0
Oil CHP	10.6	8.5	10.0	9.7	9.3	9.0	8.7	0.0
Coal CHP	22.6	20.9	19.6	15.0	15.0	15.2	15.3	15.4
Fuel Cell CHP	0.0	0.0	0.0	0.0	0.0	0.0	0.0	0.0

Table 3. Technology level EROI estimation of IEA STEPS scenario when fuel CED is changed (sensitivity analysis)

Technology/Year	2015	2020	2025	2030	2035	2040	2045	2050
PV fixed tilted	11.7	14.0	15.2	18.0	21.1	23.8	25.9	27.4
PV prosumers	11.5	12.1	15.6	17.6	19.6	24.8	31.0	32.7
Wind onshore	44.8	46.5	39.9	42.1	44.5	46.1	47.3	48.1
Wind offshore	50.9	62.1	54.1	59.6	64.1	67.8	70.4	72.9
Hydro run-of-river	57.5	57.7	51.6	52.2	52.6	53.0	53.3	53.6
Hydro reservoir (dam)	94.1	94.6	84.5	85.5	86.4	87.1	87.8	88.3
Biomass solid	12.9	13.2	12.9	13.0	13.0	13.0	13.0	13.0
Geothermal electricity	4.2	9.0	6.9	7.1	7.2	7.3	7.3	7.4
CSP ST	33.3	58.0	17.8	19.5	22.7	24.56	25.2	25.7
ST	0.0	0.0	0.0	0.0	0.0	0.0	0.0	0.0
Wave	0.0	0.0	0.0	0.0	0.0	0.0	0.0	0.0
CCGT	20.7	16.5	13.8	11.1	8.9	7.1	5.7	4.9
CCGT +CCS	0.0	0.0	0.0	0.0	0.0	0.0	0.0	0.0
OCGT	20.9	16.0	13.1	10.7	8.6	6.9	5.6	4.8
ICE	8.6	6.7	5.8	4.8	4.1	3.6	3.1	0.0
Multifuel ICE	0.0	0.0	0.0	0.0	0.0	0.0	0.0	0.0
Coal power plant	24.2	23.5	22.5	21.9	21.4	20.9	20.5	20.1
Coal power plant +CCS	0.0	0.0	0.0	0.0	0.0	0.0	0.0	0.0
Nuclear power plant	17.2	17.3	17.2	17.3	17.3	17.3	17.3	17.3
Biomass CHP	13.1	13.2	11.6	11.6	11.6	11.6	0.0	0.0

Waste-to-energy CHP	51.0	53.5	55.5	55.5	55.5	55.5	55.5	55.5	55.5
Biogas CHP	13.3	14.0	13.9	13.9	13.9	13.9	13.9	13.9	13.9
Methane CHP	103.3	77.7	95.2	94.2	98.2	101.9	105.4	0.0	
Oil CHP	7.8	5.8	5.5	4.6	4.0	3.4	3.0	0.0	
Coal CHP	22.6	20.9	19.6	14.9	15.0	15.1	15.3	15.4	
Fuel Cell CHP	0.0	0.0	0.0	0.0	0.0	0.0	0.0	0.0	

NOTE: If my interpretation of your Equation (2) is correct, then I do think that your direct comparison of LCOE and EROI system is possibly consistent, since LCOE can also be put into terminology of LCOE = (annualized money inputs)/(annualized electricity output) but it then assumes the SAME annualized money inputs each year. In reality, the full cost of money to construct and install the power plant was paid (to the builders) at the beginning of the project just as all of the energy inputs for construction were consumed at the beginning. So we know LCOE is an abstraction (a summary) of the real cash flows, just as you seem to be creating the similar abstraction of energy flows. I don't think it is useful to promote this type of abstraction of energy flows (say, average annual power = energy/year) that net energy analysis is trying to illuminate.

We thank for the nice note from Reviewer#3. This comment is another way of point to the perspectives.

First of all, this annual approach is mathematically equivalent to traditional EROI calculation. Thus, its abstraction is the same as EROI. We copy again the technology level, equation for clarification of the issue:

$$EROI = \frac{EO}{EI} = \frac{EO_{an} \cdot LT}{EI} = \frac{\left(\frac{EO_{an} \cdot LT}{LT}\right)}{\frac{EI}{LT}} = \frac{EO_{an}}{\left(\frac{EI}{LT}\right)}$$

where EO – lifetime output, EI-lifetime input, and the subscript “an” represents annual values.

EROI has also a physical meaning in terms of its ability to inform decision makers regarding the system ability to maintain net energy required for economic development. Thus, the comparison with LCOE somehow loses the point. LCOE is an abstraction we need because tracking financial flow is somehow challenging in large analysis. But EROI is not a tool we need to replace something, but a must developed tool to achieve its scientific purpose.

As we noted above, PROI is a different tool and should not be confused with EROI. EROI is used to measure sustainability of the energy system while PROI is an energy flow analysis. PROI cannot replace EROI, but may be used to identify possible bottlenecks.

Major Comments #3	
The ecoinvent database provides CED values by separating them into eight sub-categories according to the energy resource type as MJ equivalents. The same category logic presented by ecoinvent is applied for the determination of conversion factors, and the MJ equivalent unit is converted to MJ-electricity on a technology level.” Please write as simple equation as possible to express what you are saying in words. It is hard to follow the conversions from words, so be explicit using an equation of how you convert between “MJ equivalents” and “MJ-electricity”.	Considering the Reviewer#3 comment, a simple equation for the conversion is given in the main paper, and small additions are made in the SI Note 2. In the revised manuscript, the following sentences and equation is added. The ecoinvent database provides CED values by separating them into eight sub-categories (renewables: biomass, geothermal, solar, potential- barrage water, kinetic-wind, and non-renewables: fossil, nuclear, and primary forest) according to the energy resource type as MJ-equivalents²². The same category logic presented by ecoinvent is applied for the determination of conversion factors, and the MJ-equivalent unit is converted to MJ-electricity on a technology level, that is represented in equation (1). MJ_{el,c} for each category (c) estimated by multiplying the primary energy (MJ_{pe-cq}) and the related electricity conversion factor (f). The conversion factors are explicitly given in SI Note 2. $MJ_{el,c} = MJ_{pe-cq} \cdot f_c \quad (1)$
Minor Comments on Article Content	
“Some of these studies questioned the plausibility of 100% RE in terms of net energy production.” Please repeat the citations (already existing) the relate to this statement.	The citations are repeated as requested. Some of these studies questioned the plausibility of 100% RE in terms of net energy production²⁰⁻²³
“Hereby, this study expands the newly developed LUT-EROI ...” I think you still need to spell out LUT the first time ...	It is revised considering Reviewer#3 comments. Hereby, this study expands the newly developed Excel-based LUT-EROI model (LUT stands for Lappeenranta-Lahti University of Technology) to study global systemwide EROI using the nine global transition scenarios presented in Aghahosseini et al.⁶.
“one of the two most used tools for highly RE system analyses ...” “high” instead “highly”?	We kept the phrase as this is a standardly used term in the near 100% renewable energy community.
The Solomon references 25, 27, and 33 don’t have any publication name associated with them. Are they in journals? Are they white papers? Something else?	These papers are still under peer-reviewing process in well-recognized journals, and unfortunately, it takes time to be published. However, we would like to point out that the core of the methodology, derived equations, used conversion factors as well as any fundamental information is also provided in the main paper as well as in the SI to provide a clearer view of the developed methodology and implementation way of systemwide EROI analysis.
“Revisiting the foregoing discussion of EROI trends by combining Fig. 1a to Fig. 1d reveals that,	The sentence is revised according to the comment.

despite of scenario dependent differences,” Make it “... despite scenario-dependent differences”	Revisiting the foregoing discussion of EROI trends by combining Fig. 2a to Fig. 2d reveals that, despite scenario-dependent differences, all trends can be linked to the corresponding VRE penetration.
Minor Comments on Supplemental	
Section 2.4, “in Table S1Hata! Başvuru kaynağı bulunamadı,..” probably needs to be corrected	We are grateful for this correction as we missed it during submission. The statement is corrected in the revised version of the SI. It is a consequence of doc to pdf conversion and should be fine now. The revision is as follows: Hereby, the primary electricity conversion factors³²⁻³⁵ are presented in Table S1, where MJ_{pe-eq} units are converted to MJ_{el} according to conversion factors ($MJ_{el} = MJ_{pe-eq} \cdot (\text{conversion factor})$) corresponding to each resource types.
For clarity, just show the use of a conversion factor in Table S1. That is to say, write something such as “MJ_{el} = (conversion factor)MJ_{pe-eq}” (but ensure to write the equation as you use it). Also specify if one needs to use the conversion of 3.6 MJ = 1 kWh.	The relevant small equation of the conversion factor is added to page 14 of the SI. The explanation for converting kWh to MJ is also included on page 15 of the SI in subsection, “2.5 Unit Conversion Factors”. The addition is given as follows: 1 kWh of electricity is assumed to be 3.6 MJ_{el}.
Figure S4. Should it read “Phase III Calculating Technology Level Invested Technology” a. That is it is “Inverted” or “Invested”?	The mistyping is corrected. We thank Reviewer#3 for pointing out this issue.

References:

1. Aghahosseini, A. *et al.* Energy system transition pathways to meet the global electricity demand for ambitious climate targets and cost competitiveness. *Applied Energy* **331**, 120401 (2023).
2. Slameršak, A., Kallis, G. & Neill, D. W. O. Energy requirements and carbon emissions for a low-carbon energy transition. *Nature Communications* **13**, 1–15 (2022).
3. RB, L. Kung bushmen subsistence: an input-output analysis. in *Environment and Cultural Behavior* (ed. édité par AP Vayda) 47–79 (1969).
4. Lambert, J. G., Hall, C. A. S., Balogh, S., Gupta, A. & Arnold, M. Energy, EROI and quality of life. *Energy Policy* **64**, 153–167 (2014).
5. Lambert, J. G., Hall, C. A. S. & Balogh, S. B. *EROI of Global Energy Resources. Status, Trends and Social Implications.* (2013).
6. Brockway, P. E., Owen, A., Brand-Correa, L. I. & Hardt, L. Estimation of global final-stage energy-return-on-investment for fossil fuels with comparison to renewable energy sources. *Nature Energy* **4**, 612–621 (2019).
7. Capellán-Pérez, I., de Castro, C. & Miguel González, L. J. Dynamic Energy Return on Energy Investment (EROI) and material requirements in scenarios of global transition to renewable energies. *Energy Strategy Reviews* **26**, 100399 (2019).
8. Solomon, A. A., Sahin, H. & Breyer, C. The pitfall in designing future energy system without considering energy return on investment in planning. (2023).
9. International Organisation for Standardisation (ISO). ISO 14040:2006 - Environmental management - Life cycle assessment -Principles and framework. (2006). Available at: <https://www.iso.org/standard/37456.html>. (Accessed: 27th February 2022)
10. International Organization for Standardization (ISO). ISO 14044:2006 - Environmental management - Life cycle assessment - Requirements and guidelines. (2006). Available at: <https://www.iso.org/standard/38498.html>. (Accessed: 27th February 2022)
11. Carbajales-Dale, M. When is EROI Not EROI? *BioPhysical Economics and Resource Quality* **4**, 1–4 (2019).
12. Murphy, D. J., Carbajales-Dale, M. & Moeller, D. Comparing apples to apples: Why the net energy analysis community needs to adopt the life-cycle analysis framework. *Energies* **9**, 1–16 (2016).

REVIEWER COMMENTS

Reviewer #1 (Remarks to the Author):

The revised manuscript is a definite improvement. However, I've noticed some careless referencing. On page 1, two of the references to the statement "Some of these studies questioned the plausibility of 100% RE in terms of net energy production^{20–23}" misrepresent the papers cited. Neither ref. 21 nor ref. 23 (which is incomplete) question the plausibility of 100% renewable energy. These gross misrepresentations suggest that the authors must check the other references.

In comparing scenarios with different end-point demands, the EROI pathway will depend on electricity demand in 2050. The higher the demand, the more rapid the rate of energy transition and hence the greater the temporary reduction in system EROI. Perhaps I have overlooked it, but it seems that the manuscript needs some discussion of the effect of different final demands.

Reviewer #2 (Remarks to the Author):

The corrections made to the manuscript, in terms of both form and content, met my expectations. I am particularly pleased with (i) the inclusion of a description of the LUT-ESTM model in the body of the text (ii) the addition of a comparison of the authors' results with the work of Slamersak et al. and Sers. As these two changes have led to the appearance of a link with IAMs, I simply invite the authors to take note of the paper linking the emerging consensus of net energy and integrated assessment models of Delannoy et al*, and if they wish to add a sentence on IAMs' limits, perhaps at the end of the section "Impacts of the energy transition on global EROI". I would also like to congratulate the authors on providing clear and dense responses to our comments, much appreciated. I therefore recommend the publication of this article.

*Delannoy, Louis and Auzanneau, Matthieu and Andrieu, Baptiste and Vidal, Olivier and Longaretti, Pierre-Yves and Prados, Emmanuel and Murphy, David J. and Bentley, Roger and Carbajales-Dale, Mik and Raugei, Marco and Höök, Mikael and Court, Victor and King, Carey and Fizaine, Florian and Jacques, Pierre and Heun, Matthew Kuperus and Jackson, Andrew and Guay, Charles and Aramendia, Emmanuel and Wang, Jianliang and Le-Boulzec, Hugo and Hall, Charlie, Emerging Consensus on Net Energy Paves the Way for Improved Integrated Assessment Modeling (July 4, 2023). Available at SSRN: <https://ssrn.com/abstract=4500020> or <http://dx.doi.org/10.2139/ssrn.4500020>

Reviewer #3 (Remarks to the Author):

GENERAL COMMENTS

Thank you for a thorough set of statement addressing my and the other reviewers' comments on your initial submission. I only refer to a couple of outstanding issues.

MAJOR COMMENTS

1) Based on your responses, I believe I fully understand your algorithm and mathematics behind your system EROI calculation. I also agree with your stated description of the differences between your system EROI calculation and a calculation of power return on power invested (PROI). What I disagree with is that there is no value in performing the PROI calculation. I understand you think it should be left for a different paper. To me, you should have all of the data and it seems relatively trivial for you do perform the PROI calculation (you can correct me as needed), aside from some uncertainty in knowing how to set the initial condition (value for 2015), but I'm thinking that just uses the same data that you're already using for EROI. Just doing the PROI of these scenarios is not worthy of another paper (in my opinion), and only entails 1 more figure in this paper and a short paragraph (or 2-4 sentences) of explanation.

a. Are against doing the PROI calculation because

- i. It is just too many calculations for 1 paper? If yes, don't you think that a Nature Communications paper is worthy of a full analysis and comparison of similar indicators?
- ii. You fear or know that the result will point to a less favorable view of energy transition feasibility due to highlighting up-front energy flows?

b. Can you not perform the PROI calculation because

- i. You actually don't have all of the necessary input or output data? (If "yes", I don't understand why this would be the case)?

2) With regard to comparing EROI and LCOE, you wrote new text as: "A further analysis of the EROI relation with system LCOE (Fig. 2h) shows that low-cost solutions correlate to low EROI. Replacing fossil and nuclear energy with renewable energy reduces the system's LCOE by lowering fuel and CO2 costs. Also, the level of reduction in the LCOE and EROI is directly correlated to the system composition and the choice of technology."

a. The final sentence conveys not meaning (obviously the mix of technologies input into your algorithm affects the results), so can be removed.

b. Your response to my review (not the manuscript) makes the following 2 statements: (1) "EROI has also a physical meaning in terms of its ability to inform decision makers regarding the system ability to maintain net energy required for economic development." And (2) "Thus, calculating system level PROI at global level considering proper energy flow could be challenging due to associated uncertainties

despite its irrelevance for the stated objective of this study.”

i. Two immediate questions are:

ii. But what are decision makers supposed to think when you tell them there will be less net energy in our scenarios (which is normally interpreted as a higher cost outcome for development) at same time as lower monetary cost?

iii. Where is the stated objective of this study? I don't see it in the manuscript.

c. I think you need to elaborate, with new text in manuscript, on the possible explanations of why your EROI and LCOE both decrease over time in your scenarios, even though with logic (as I imply below) we'd expect them to be inversely related. I believe this points to a VERY CORE principle as to whether EROI analyses can be useful in decision making in addition to using monetary metrics like LCOE. That is to say, I think it is very important for you to interpret your EROI calculation to LCOE, especially given that you simply used the results of a model that produced the LCOE calculations such that your comparison is relatively self-consistent (still not 100% so since the EROI calculations are not fully integrated into the simulation of Aghahosseini, A. et al. (2023), and monetary and energy input assumptions are not linked in any fundamental way).

d. Given basic definitions (you mentioned King and Hall (2011) looked at price to PROI, but this concept also holds for cost compared to EROI on a lifetime basis).

i. Say: $EROI = \text{MWh}_{el\ out} / E_{invested}$

ii. Say: $LCOE = \text{euros cost} / \text{MWh}_{el\ out}$

iii. Because “MWh_el” is the same in each equation, the differences can only be in “euros cost” and “E_invested”. Since both EROI and LCOE decline in later years of all scenarios per your algorithm and method, how much of the reason why both EROI and LCOE can decrease with the transition is one or more of:

1. 1) CO2 costs that only show up in LCOE (You mention this, but I don't think this is an explanation since LCOE can only increase from CO2 monetary costs relative to no CO2 cost)?

2. 2) Discounting (time value of money) in LCOE that is not in EROI?

3. 3) Euro cost assumptions are too low relative to Energy investment assumptions? (Possibly assumptions for electrolysis and methanation (for gas storage) are assumed too optimistic in terms of euros.)

4. 4) Energy investment assumptions (fromecoinvent) are too large relative to euro cost?

5. 5) This ‘inverse relationship reasoning’ is too simple for reasons(s) A, B, C ... one reason could be that monetary costs decline faster than energy costs rise because they are associated with non-physical costs (e.g., engineering, lawyer, and other soft costs that don't consume much energy) that were assumed to decline very quickly.

iv. In short, based on your results someone could legitimately argue something akin to either of the following:

1. (believe EROI, don't believe LCOE): “By showing that EROI declines in high decarbonation scenarios, and because EROI and LCOE are fundamentally inversely related, this indicates that the monetary cost assumptions are likely too optimistic.”

2. (believe LCOE, don't believe EROI): “The fact that both LCOE and EROI decrease for high decarbonization scenarios, and because EROI and LCOE are fundamentally inversely related, the most

likely explanation is that the energy investments will likely be less than what the authors have assumed, and possibly the life cycle inventory data for cumulative energy demand are too high.”

v. I’m not claiming you have to solve my conundrum in your manuscript, or that I know you have made a mistake. I am claiming that the interpretation of your result is confusing, thus not of clear value, with regard the LCOE vs. EROI trends. And, it is important to tell the reader (and the community) that we should understand what to expect in terms of LCOE vs. EROI trends. Thus, you should acknowledge that your trend in Figure 2(h) defies basic mathematical logic, and that there are some options to understand why (in future research). Of course you can correct me (that LCOE and EROI don’t have to be inversely related), but I need convincing.

e) To me, the comparison of LCOE to EROI is a core issue because the net energy community is essentially saying “EROI” and such metrics are important, but to convince ourselves and others that net energy metrics are useful, we need to explain how they should or should not relate to monetary or other metrics. My opinion in this review is that your current manuscript is exactly the place to discuss if you can or cannot make a meaningful comparison of EROI to LCOE, and the reason is that you are performing these calculations using computational tools from your research group of which you are the experts. No one else should be more able to answer this question, and the question for your tools is not enough for an additional manuscript. I cannot recommend this paper without a reasonable comparison, explanation, or discussion of why both EROI and LCOE decline in your future scenarios, and (in addition to believing PROI type calculation is valuable,) I’ve suggested several uncertainties for inputs that could be part of the explanation, but of course there could be others.

MINOR COMMENTS ON ARTICLE CONTENT:

3) Apologies I did not suggest a change here in version 1, but you write on page 1 “Fundamentally, it is a ratio that measures the amount of energy invested to deliver the energy need of the society.”
a. Strictly, EROI does not say anything about “the energy need of society”. We are trying to use studies such as yours to understand what society “needs” in a general sense (e.g., what are the tradeoffs at low EROI?). So I suggest a reword that is more related to the strict mathematical definition, such as “... energy output of an energy system or technology relative to the energy invested to build and operate said system or technology.”

4) State (in 1 sentence if possible, or in an existing sentence) how CO2 costs are included in LCOE (on emissions of power plants only?) from the Aghahosseini, A. et al. (2023) study, and if there are any “energy investments” associated with CO2 costs. The reason is that (1) the reader of this manuscript does not necessarily know that the existing study includes a price or cost on CO2 and (2) I can't tell if CO2 monetary cost translates to any energy investments (I understand the monetary costs dictate the mix of power plants via the cost-minimization in Aghahosseini, A. et al. (2023).)

Reviewer #1	Response to Reviewer #1
Reviewer comments:	Thank you for your valuable comments and suggestions. Kindly note that the changes are already reflected in the revised R2 manuscript via the track change mode. However, for the sake of clarity and to save time, we have also provided the direct changes made in accordance with Reviewers' comments in blue. The words/phrases in bold and underlined refer to specific changes. Apart from the moderate/minor changes requested by the reviewers, the major ones are presented in the revised R2 manuscript.
The revised manuscript is a definite improvement. However, I've noticed some careless referencing. On page 1, two of the references to the statement "Some of these studies questioned the plausibility of 100% RE in terms of net energy production²⁰⁻²³" misrepresent the papers cited. Neither ref. 21 nor ref. 23 (which is incomplete) question the plausibility of 100% renewable energy. These gross misrepresentations suggest that the authors must check the other references.	Thank you for pointing out this mistake. We have corrected the references. Some of these studies questioned the plausibility of 100% RE in terms of net energy production^{20,22}.
In comparing scenarios with different end-point demands, the EROI pathway will depend on electricity demand in 2050. The higher the demand, the more rapid the rate of energy transition and hence the greater the temporary reduction in system EROI. Perhaps I have overlooked it, but it seems that the manuscript needs some discussion of the effect of different final demands.	Thank you for this comment. In the subsection "Impact of the energy transition on global EROI", we have added a brief discussion, also included below: Note that the five BPS scenarios created with a slightly higher demand projection that reaches 48.38 PWh in 2050 as compared to the IEA and Teske/DLR scenarios, which remain in the range of 45-46.5 PWh. However, no clear evidence was found that demand influenced the EROI trends such as BPS-WF scenarios show a slower decline compared to Teske/DLR scenarios. Further in-depth examination of the data also reveals that the system composition and the technology choice have more impact on the trend.

Reviewer #2	Response to Reviewer #2
Reviewer comments:	Thank you for your valuable comments and suggestions. Kindly note that the changes are already reflected in the revised R2 manuscript via the track change mode. However, for the sake of clarity and to save time, we have also provided the direct changes made in accordance with Reviewers' comments in blue. The words/phrases in bold and underlined refer to specific changes. Apart from the moderate/minor changes requested by the reviewers, the major ones are presented in the revised R2 manuscript.
The corrections made to the manuscript, in terms of both form and content, met my expectations. I am particularly pleased with (i) the inclusion of a description of the LUT-ESTM model in the body of the text (ii) the addition of a comparison of the authors' results with the work of Slamersak et al. and Sers. As these two changes have led to the appearance of a link with IAMs, I simply invite the authors to take note of the paper linking the emerging consensus of net energy and integrated assessment models of Delannoy et al*, and if they wish to add a sentence on IAMs' limits, perhaps at the end of the section "Impacts of the energy transition on global EROI". I would also like to congratulate the authors on providing clear and dense responses to our comments, much appreciated. I therefore recommend the publication of this article. *Delannoy, Louis and Auzanneau, Matthieu and Andrieu, Baptiste and Vidal, Olivier and Longaretti, Pierre-Yves and Prados, Emmanuel and Murphy, David J. and	Thank you very much for the constructive comments. We believe the recommended paper highlights important points we discussed in the R1 discussions. Thus, in the last part of the subsection "Impacts of the energy transition on global EROI" we added a sentence as follows: Further information on the key differences in IAMs implemented in EROI studies is explicitly discussed in a recent study by Delannoy et al.⁴⁰ Ref:40: Delannoy, L. et al. Emerging Consensus on Net Energy Paves the Way for Improved Integrated Assessment Modeling. SSRN Electronic Journal 1–17 (2023). doi:10.2139/ssrn.4500020 In light of the R1 discussions, a few sentences are added to the Discussion section, which is given below: These important findings support the claims of technological optimism⁴⁷ that the transition to 100% RE systems does not result in a significant disruption from a physical EROI perspective. The impacts of high electrification and energy efficiency should be further investigated to anticipate the changes in systemwide EROI trends. Ref:47: Raugei, M. Addressing a Counterproductive Dichotomy in the Energy Transition Debate. Biophysical Economics and Sustainability 8, 1–6 (2023).

Bentley, Roger and Carbajales-Dale, Mik and Raugei, Marco and Höök, Mikael and Court, Victor and King, Carey and Fizaine, Florian and Jacques, Pierre and Heun, Matthew Kuperus and Jackson, Andrew and Guay, Charles and Aramendia, Emmanuel and Wang, Jianliang and Le-Boulzec, Hugo and Hall, Charlie, Emerging Consensus on Net Energy Paves the Way for Improved Integrated Assessment Modeling (July 4, 2023). Available at SSRN: <https://ssrn.com/abstract=4500020> or <http://dx.doi.org/10.2139/ssrn.4500020>

Reviewer #3	Response to Reviewer #3
Reviewer comments:	Thank you for your valuable comments and suggestions. Kindly note that the changes are already reflected in the revised R2 manuscript via the track change mode. However, for the sake of clarity and to save time, we have also provided the direct changes made in accordance with Reviewers' comments in blue. The words/phrases in bold and underlined refer to specific changes. Apart from the moderate/minor changes requested by the reviewers, the major ones are presented in the revised R2 manuscript.
General Comments	
Thank you for a through set of statement addressing my and the other reviewers' comments on your initial submission. I only refer to a couple of outstanding issues.	Thank you for the constructive comments. We are also glad to know that the reviewer agrees with the distinction we made between EROI and PROI. The remaining issues also correspond to matters that we did not explain in the previous versions. We hope that we have answered your questions with the clarification that we have provided regarding these issues below.
Major Comments #1	
1) Based on your responses, I believe I fully understand your algorithm and mathematics behind your system EROI calculation. I also agree with your stated description of the differences between your system EROI calculation and a calculation of power return on power invested (PROI). What I disagree with is that there is no value in performing the PROI calculation. I understand you think it should be left for a different paper. To me, you should have all of the data and it seems relatively trivial for you do perform the PROI calculation (you can correct me as needed), aside from some uncertainty in knowing how to set the initial condition (value for 2015), but I'm thinking that just uses the same data that you're already using for EROI. Just doing the PROI of these scenarios is not worthy of	Thank you for the comments and for letting us know the clarity of our previous explanation on issues of EROI and PROI. We hope we do the same in this case too. We understand your suggestions. However, this paper focuses on the systemwide implementation of physical EROI. The leading challenge as regards to the inclusion of PROI is due to an answered question of: how one can estimate systemwide PROI without reliable and proper systemwide methodology? Even if we had tried to conduct the PROI estimation, the question remains whether the comparison would be logical and meaningful. Scientifically, two distinct parameters, used for different purposes, that require different methodology for their evaluation will logically require different references for their interpretation and thus direct comparison requires further work. Thus, we strongly believe that the comparison of the two indicators with different methodologies (EROI is estimated based on impact assessment data while PROI analysis relies on the dynamic annual energy flow of the system) should not simply be presented here, but rather should be examined in detail through a separate case study, including differences in their purpose, divergence in their methodologies, advantages, and disadvantages, and the ways of the implementation on the energy modelling tools. Even with such detailed data that we have gathered and the systemwide EROI analysis that we developed; we believe that achieving a reliable estimate of PROI is a non-trivial task at present time. To our knowledge, we have not found a

another paper (in my opinion), and only entails 1 more figure in this paper and a short paragraph (or 2-4 sentences) of explanation.

comprehensive system-level PROI methodology including various technologies, particularly the enabling technologies (DAC, methanation unit, electrolyser, and various storage options). To logically implement system-level PROI estimation, we need to develop a systemwide PROI methodology and find a way to incorporate these technologies. This requires independent and further comprehensive analysis.

The present systemwide EROI relies on the LUT-ESTM result, which is modelled in a 5-year step, meaning if we take a capacity installed in 2025. That capacity is assumed to generate electricity starting on 1 January that year. All construction works should be completed and connected to the grid by 31 December 2024 (in reality the capacities are built and commissioned about 2.5 years before and after that point in time). The annual historical capacity growth data of renewable technologies could provide good information on how it is distributed over time for various technologies¹. Thus, the lack of overlap between energy generation and energy investment (EI) for construction/manufacturing/installation is clear in this format. The corresponding EI occurs around 2025 for these systems. Thus, it is not possible to perform a correct PROI estimation since it involves time transfer of energy flow. However, the aggregate of a 5-years period would be possible. We capture the energy flow correctly only when we convert everything to an annual time step (or the 5-years average for one annual time step).

To provide a better understanding of the divergence in the systemwide EROI and PROI methodologies, we used the systemwide energy investment flow (5-year period) as a basis for an annual energy investment flow while categorised the energy investment types according to their intended use, i.e., upstream supply chain for fuel production, operation, and investment. The same calculation method as the systemwide EROI is used, but the CED values of the technologies spread over their lifetime are converted to CED values per year and instead annual net capacities are taken. Although what we have presented is a simplified version of the annual energy investments for our methodology, still, it is not correct to define it as PROI estimation since the PROI is implemented without taking into account the overlapping lifespans of the technologies. The consideration of the time lag in the transition period is mostly not considered (distributing the energy requirements over the following next years). Furthermore, the dynamics of electricity generation and energy requirements are captured on a micro-level in PROI at technology level, which is different than system-level analysis where the interactions between technologies are present and reflected in the systemwide EROI analysis.

The annual energy investment flow analysis together with detailed graphs is presented in the Supplementary Information in the Supplementary Note 6, which explains energy investment flows in the energy transition. A brief analysis is provided, and two energy investment flow graphs (for the systemwide energy investment

	flow and the annual energy investment flow during the energy transition) are detailed by the mentioned energy investment types. In addition, considering the Reviewer’s suggestion, we added a paragraph in the revised R2 manuscript at the end of the subsection “Impact of the energy transition on global EROI” as follows: Furthermore, the sustainability risk of the systemwide EROI is reanalysed by the annual energy investment flows for each scenario. The profound analysis signifies that none of the scenarios has exceeded 16% of the final energy consumption of the respective year (Fig. S20) while the upper limit for the systemwide energy investment flow is estimated as 7% (Fig. S19). In both cases, the energy need for the upstream supply chain of fuel production (excluding energy content of fuels) approaches zero during the transition as this is gradually substituted by the energy necessity of RE investments in the defossilisation scenarios. However, this shift occurs more slowly in the IEA scenarios because of the lower share of RE and continued utilisation of fossil and nuclear power plants. Also, the shortening of the ET triggers the need for more energy for RE investment in the BPS-plus scenarios, as observed in the annual energy investment flow (Fig. S19). Considering the CED in the year of the investment creates greater fluctuations in the annual energy investment flow during the ET (Fig. S21). Despite of a high expansion of RE, the estimated annual energy return on investment value stays in the low-risk range, as defined by Capellán-Pérez et al.²². Further details are presented in the SI Note 6.
a. Are against doing the PROI calculation because i. It is just too many calculations for 1 paper? If yes, don’t you think that a Nature Communications paper is worthy of a full analysis and comparison of similar indicators? ii. You fear or know that the result will point to a less favorable view of energy transition feasibility due to highlighting up-front energy flows?	We are not against doing a PROI calculation, as we fully understand the importance of the concept itself. We have clarified the scientific ground of why PROI should be studied independently in previous questions. Instead of this, we have provided annual energy investment flows for all scenarios alongside with a brief analysis. We hope the Reviewer can see that we need to limit the scope of the study for scientific reasons. It has nothing to do with the amount of calculation that we need to perform. Deriving a new methodology for the systemwide PROI is still one of the key challenges to performing this analysis. In order not to compromise the integrity of our article and not to disregard the aim of the presented study, we strongly believe that it is not correct to present two different analyses having two different methodologies in one article. Also, performing proper PROI estimation is a separate research project requiring much more rigor. Yet, considering the Reviewer concerns, we perform an annual energy investment flow analysis for all scenarios. The analysis shows that the annual energy return on investment values remain in the low-risk range, as defined by Capellán-Pérez et al.², and not fall below 5.

	The data we provide is a complete and comprehensive work. To our knowledge, we have not found any other study that provides a systemwide EROI analysis while addressing existing shortcomings of EROI as we did in this study. Hence, we strongly believe that Nature Communications is best suited for this paper. Neither the provision of adding maybe incomplete PROI methodology will bring benefit to the journal or satisfy the academic community, nor it will make a significant contribution to the EROI and PROI literature. For this reason, we suggest that this topic, together with the methodological differences, be dealt with separately in another study. We trust that the Reviewer will consider our genuine care for the integrity of what we present to the scientific community.
b. Can you not perform the PROI calculation because i. You actually don't have all of the necessary input or output data? (If "yes", I don't understand why this would be the case)?	Our recommendation to postpone the PROI study was due to the fundamental issues discussed above, data availability being one of them. Thus, we hope that the Reviewer has already seen our explanation. Instead of PROI, we have provided the annual energy investment flows for all scenarios along with a concise analysis in the SI and added a paragraph on this topic in the revised R2 manuscript.
2) With regard to comparing EROI and LCOE, you wrote new text as: "A further analysis of the EROI relation with system LCOE (Fig. 2h) shows that low-cost solutions correlate to low EROI. Replacing fossil and nuclear energy with renewable energy reduces the system's LCOE by lowering fuel and CO₂ costs. Also, the level of reduction in the LCOE and EROI is directly correlated to the system composition and the choice of technology." a. The final sentence conveys not meaning (obviously the mix of technologies input into your algorithm affects the results), so can be removed. b. Your response to my review (not the manuscript) makes the following 2	This is one of the findings we presented in our manuscript. The discussion on EROI and LCOE is not based on econometric analysis or covers the case of fossil fuel dominated systems. However, we present an analysis for the relative trend of the two indicators as the economic and energy input assumptions are independent is logically a requirement for the present case (see further explanation below). (a) The last sentence, which is based on data, presents clear reminders to readers that LCOE and EROI trends may change depending on system composition, unit cost, and technology selection. Critically, this study is based on 9 energy transition scenarios, the energy mix was forced only in the case of the IEA and Teske/DLR scenarios, as it is not possible to reproduce those results without forcing it. Based on the current cost set, there is no possibility that the model could produce results that are not similar to the BPS scenarios. Further technical issues are discussed in the study of Aghahosseini et al.³. Additionally, we also understand that such a link is rarely reported, because such detailed EROI studies covering energy transition are not common. The result of EROI and LCOE trend as in this study should not be a surprise due to reasons to be discussed below. Thus, we would like to leave it that way for researchers who would like to examine it further, since we strongly believe this aspect requires further investigation.

statements: (1) “EROI has also a physical meaning in terms of its ability to inform decision makers regarding the system ability to maintain net energy required for economic development.” And (2) “Thus, calculating system level PROI at global level considering proper energy flow could be challenging due to associated uncertainties despite its irrelevance for the stated objective of this study.”

i. Two immediate questions are:

ii. But what are decision makers supposed to think when you tell them there will be less net energy in our scenarios (which is normally interpreted as a higher cost outcome for development) at same time as lower monetary cost?

(b) We understand your concerns. We have provided our answers to these questions, particularly methodological issues, and the lack of data. The aim of this study is also stated in the Introduction section, which is explicitly given in one of the targeted questions. Before starting to explain, we would like to clarify that the system LCOE includes the following components:

- LCOE primary (main investment costs)
- LCOS (levelised cost of storage)
- LCOC (levelised cost of curtailment)
- LCOT (levelised cost of transmission)
- Fuel cost
- CO₂ cost

(2b-i) Thank you for the specific questions. We provided the point-by-point answers below.

(2b-ii) Decision-makers should focus on whether the scenarios can avoid falling into the energy cliff and which scenarios preserve their vulnerability to the dependence on finite hydrocarbon resources, which is the case here, and this is explicitly stated in the last paragraph of the Discussion section.

For EROI above 10, the concerns about the difference in net energy are insignificant as seen in Figure 1. A high EROI from a net energy perspective does not lead to a significant sustainability risk that might arise in energy transition scenarios. However, the decreasing trend in EROI should be a concern only when it drops below the threshold of the net energy cliff, in that case, it could have significant implications. Furthermore, our modelling results overlap with the findings of Way et al.⁴, clearly highly RE systems can be achievable at lower costs. Review concerns may follow intuitive logic that links system cost to EROI, which could be true when EROI falls below 10. Thus, decision-makers should understand the link between physical EROI and cost perspective depending on suitable context.

[REDACTED]

Figure 1. Net Energy Cliff with EROI⁵.

The presented study examines EROI for different energy transition pathways, which takes the system from a fossil to a renewable dominated system. Fossil consumption takes a notable share of both the required energy investment and LCOE from the present system, in the future this component will significantly diminish or fully be eradicated depending on the scenario. Thus, we strongly believe that it is not right to apply the present and estimated fossil-based theories directly to a RE-based system. In RE-based systems both the energy investment and technology cost is expected to improve with time, thus, such comparison should be made with the full understanding of each systems characteristics because each EROI value change over the years related to different systems even for the same scenario.

Based on ongoing discussions, we decided to put a brief explanation related to the limitations and uncertainties arising from the nature of the models on how these decisive factors may affect the relationship between the system LCOE and EROI trends.

The relationship between EROI and LCOE is compared by a relative trend analysis of the corresponding modelled LCOE results and physical EROI estimations. The two indicators change during the energy transition in response to a change in the energy mix and technology selection. The reported relationship

	between EROI and LCOE may not follow similar conditions as the present system, where fossil fuel consumption takes a notable share of the required energy investment and LCOE. Systemwide ET studies with higher fossil fuel versus RE shares may reveal a more complex relationship of these two key energy system metrics, such as a time delay in the response or structural differences due to cost-price aspects or the role of supporting technologies among others. Future data will enrich the nature of this relationship. Importantly, this relationship should only be examined in ET studies at the system level, where multiple interactions between technologies occur.
iii. Where is the stated objective of this study? I don't see it in the manuscript.	The aim of this study is stated in the Introduction section and stated as follows. Thanks to your important comment, we edited the section to clarify it: Hereby, this study expands the newly developed Excel-based LUT-EROI model (LUT stands for Lappeenranta-Lahti University of Technology) in order to study global systemwide EROI based on the nine global transition scenarios presented in Aghahosseini et al.⁶, to evaluate the sustainability risk of these transition scenarios from the perspective of physical EROI. The overall modelling framework improved on the existing shortcomings of corresponding EROI studies and enhances the representativeness of the estimated physical EROI values by implementing a holistic approach for estimating primary energy quality at electricity level²⁹, creating a broader cumulative energy demand (CED) database for technologies based on life cycle assessment (LCA) databases²⁸, and integrating EROI estimation with energy system model output²⁷. Notably, the potential impacts of the electricity generation mix on EROI and significant factors contributing to these impacts are deeply analysed. Finally, this study also presents how EROI links to systemwide levelised cost of electricity (LCOE) and CO₂ emissions.
c. I think you need to elaborate, with new text in manuscript, on the possible explanations of why your EROI and LCOE both decrease over time in your scenarios, even though with logic (as I imply below) we'd expect them to be inversely related. I believe this points to a VERY CORE principle as to whether EROI analyses can be useful in decision making in addition to using monetary metrics like LCOE. That is to say, I think it is very important for you to interpret your EROI calculation to LCOE, especially given that you	Thank you for the constructive comment. We have expanded the respective section explaining the system LCOE and EROI relationship in the results section and added the following explanation to clarify this issue. However, we would like to recall that the studies showing this inverse relationship only considered fossil fuel-based systems and not highly RE systems. There are several differences in methodology, system boundaries, assumptions, and technology types that may alter this relationship. This of course also applies to our modelling and EROI estimates as well as the trends given in the manuscript. The other issue is that this study is presenting EROI and LCOE relation associated to a changing system (as also discussed below). The characteristics in the current system will not reflect this change properly. Thus, the given current data is the only solution until future data provides further evidence. Further analysis of the EROI relation with system LCOE (Fig. 2h) shows that low-cost solutions correlate to low EROI. Replacing fossil and nuclear power with RE reduces the system's LCOE by lowering fuel and CO₂

simply used the results of a model that produced the LCOE calculations such that your comparison is relatively self-consistent (still not 100% so since the EROI calculations are not fully integrated into the simulation of Aghahosseini, A. et al. (2023), and monetary and energy input assumptions are not linked in any fundamental way).	costs. The EI for the upstream processes of the fuel chain initially account for a significant percentage of the total system EI. As the transition progresses, this share is replaced by the EI of the RE technologies and their enabling systems, and the addition of enabling technologies leads to a further decrease in EROI values. The level of reduction in the LCOE and EROI is related to the key decisive factors, which are system composition, technology selection, and the unit cost of technology. The observed trend may change depending on these decisive factors, and the interpretation here is presented by a relative trend analysis of two different indicators. This conclusion may change in future depending on the system types (predominantly fossil-nuclear powered or highly RE with enabling technologies), as the investigation of system level ET studies increases, and thus, the complex nature of this relationship can be better elucidated. In the added text, we also clarified that the comparison of LCOE and EROI is based on a relative trend analysis of the two indicators. Yes, economic and energy input assumptions are independent. We cannot think of better methodology due the requirement of the specific study. A correct net energy calculation requires implementation of physical EROI as we did in this manuscript, part of the reason can be seen in Solomon et al.⁶. The inputs and outputs should thus conform to basic quality requirement in order to achieve dimension-lessness and comparability of the EROI in the transition period. Since this study is about the future system, the economic data should be based on the corresponding LCOE (as discussed below). Thus, the result cannot be compared to the past studies, which may not consider the issues that we consider in this study.
d. Given basic definitions (you mentioned King and Hall (2011) looked at price to PROI, but this concept also holds for cost compared to EROI on a lifetime basis). i. Say: $EROI = \frac{MWh_{el\ out}}{E_{invested}}$ ii. Say: $LCOE = \frac{euros\ cost}{MWh_{el\ out}}$ iii. Because “MWh_{el}” is the same in each equation, the differences can only be in “euros cost” and “E_{invested}”. Since both EROI and LCOE decline in later years of all scenarios per your algorithm and method, how much of	(d) The paper in question focused its argument on the US oil and gas sector. Based on the case study, there is an inverse relationship between EROI and energy prices as EROI decreases as energy prices increase due to depletion of fossil fuel production. The key question is whether this generalisation applies to highly RE systems, which are different than fossil fuels and, more importantly, do not rely on resource depletion. This requires another comprehensive study in this field. In this study, we did not go for an in-depth economic analysis, rather we simply performed a relative trend analysis with the system LCOE value and the corresponding EROI value for the selected year. The price of energy is determined by market dynamics. Also, wind fall profits from fossil fuels could be the reason for observing this inverse relationship between LCOE and EROI systems, where the cost of extracting fossil fuels (oil and gas) is lower than the market price (just have in mind that oil extraction in the Middle East costs about 5-10 USD/barrel and in Siberia about 20-30 USD/barrel,

the reason why both EROI and LCOE can decrease with the transition is one or more of:

- 1. 1) CO2 costs that only show up in LCOE (You mention this, but I don't think this is an explanation since LCOE can only increase from CO2 monetary costs relative to no CO2 cost)?**
- 2. 2) Discounting (time value of money) in LCOE that is not in EROI?**
- 3. 3) Euro cost assumptions are too low relative to Energy investment assumptions? (Possibly assumptions for electrolysis and methanation (for gas storage) are assumed too optimistic in terms of euros.)**
- 4. 4) Energy investment assumptions (from ecoinvent) are too large relative to euro cost?**
- 5. 5) This 'inverse relationship reasoning' is too simple for reasons(s) A, B, C ... one reason could be that monetary costs decline faster than energy costs rise because they are associated with non-physical costs (e.g., engineering, lawyer, and other soft costs that don't consume much energy) that were assumed to decline very quickly.**

which deliver very high volumes and are sold for much higher prices). The assumptions and values used in the referred study are built on energy prices and could lead to different results if the study were based on energy costs (as mentioned on the case of crude oil extraction costs in major world regions). It is important to note that this study does not follow PROI, or integrated to any econometric analysis during EROI estimates. Furthermore, the mentioned study by the Reviewer is more an explanation of the past, where our study is about the future and relies on LCOE as a result. It also presents the case of evolving systems as compared to a uniform system. A comparison should be made when scientific context matches.

(2d-iii) The systemwide EROI analysis is created based on the basic description of EROI. During the EROI estimation phase, we did not include econometric analysis, which is also described in the methods section. Thus, we can say that the cost assumptions in the modelling are not directly 100% integrated into EROI estimates. There are a few reasons why LCOE and EROI have this relationship:

- 1) **LCOE:** CO₂ and fuel costs are one of the major drivers of the decrease in system LCOE, which is explicitly presented both in the manuscript and Aghahosseini et al.³ Furthermore, new installations of RE and enabling technologies expansion over time does not bring as much as excessive burden compared to fossil-fuel based system, so it ends up with positive savings. That is why we have low LCOEs in highly RE-based systems. Note that large-scale renewables (CSP and geothermal) could increase the LCOE of the system due to the larger investment as seen in Teske/DLR scenarios.

EROI: The EIs for the extraction and operation phases are higher in the past period (2015-2020) due to the high consumption of fossil fuels and nuclear fuels. This corresponds to slightly over 50% of the total energy invested in these years. As the system decarbonises, this ratio drops to an almost considerable level due to the phase-out of fossil fuels and the restriction of nuclear power use, so that the loss share is replaced by EI in the expansion of RE and enabling technologies. While this decline is greater in the BPS, BPS-plus, and Teske/DLR scenarios, the IEA scenarios still have a significant share of it by 2050 (a change of 25% to 50% of total EI). This reflects the use of gas and nuclear technologies in the IEA-SDS and, furthermore, the continuation of coal use in the IEA-STEPS.

Thus, the upstream supply chain of fuel production (not including the energy content of the fuel) has a strong influence on EROI trends. For this reason, we have presented a sensitivity analysis by relying on the recent study presented by Delannoy et al.[7] to show that changes in this additional energy consumption in the supply chain processes could reduce the systemwide EROI values if the system still has a significant share of fossil fuels and/or nuclear power. Note that the CEDs for nuclear fuel (not including the energy content of the fuel) are still very conservative and are derived based on the ecoinvent database. When these values are increased, the EROIs of systems dominated by fossil fuels decrease even more, as seen in the IEA-STEPS – as pointed out by the Reviewer -, and might lead to the inverse relationship between LCOE and EROI.

For highly RE systems, where the system's LCOE is lower due to low-cost technologies and certainly the positive benefits arising from $CO_2$ and fuel costs, yet the systemwide EROI decreases due to capacity expansion in the enabling technologies and large expansion of RE. Notably, the sum of EI for enabling technologies (including the power to e-methane, batteries, and other storage) corresponds to the range of 10-23% of total system EI in the decarbonisation scenarios. Exceptionally, this value is below 0.5% in both IEA scenarios. Therefore, several factors could alter EROI trends.

2) Discounting is only applied to the LCOE during energy modelling phase. No econometric analysis, discounting, or simplified cost analysis is incorporated into the systemwide EROI calculation.

3) The cost assumptions are taken from reliable studies and the institute reports, which are listed below along with the cost assumption of these technologies given in Aghahosseini et al.[3]:

**Table 2.** Cost of some emerging technologies presented in SI of Aghahosseini et al.[3].

| Technology | | Unit | 2015 | 2020 | 2025 | 2030 | 2035 | 2040 | 2045 | 2050 |
|---|---|---|---|---|---|---|---|---|---|---|
| **Electrolysis**[8,9] | **CAPEX** | **€/kW$_{H2}$** | 938 | 803 | 586 | 446 | 381 | 347 | 313 | 291 |
| | **OPEX** | **€/(kW$_{H2}$ a)** | 37.5 | 28.1 | 20.5 | 15.6 | 13.3 | 12.1 | 11.0 | 10.2 |
| | **OPEX $_{var}$** | **€/kW$_{H2}$** | 0.001 | 0.001 | 0.001 | 0.001 | 0.001 | 0.001 | 0.001 | 0.001 |
| **Methanation**[8,9] | **CAPEX** | **€/kW$_{SNG}$** | 607 | 558 | 409 | 309 | 274 | 251 | 227 | 211 |
| | **OPEX** | **€/(kW$_{SNG}$ a)** | 27.9 | 25.7 | 18.8 | 14.2 | 12.6 | 11.5 | 10.4 | 9.7 |
| | **OPEX $_{var}$** | **€/MWh$_{SNG}$** | 0.002 | 0.002 | 0.002 | 0.002 | 0.002 | 0.002 | 0.002 | 0.002 |
| **Gas storage (e-methane)**[10] | **CAPEX** | **€/kWh$_{th}$** | 0.05 | 0.05 | 0.05 | 0.05 | 0.05 | 0.05 | 0.05 | 0.05 |
| | **OPEX** | **€/(kW$_{th}$ a)** | 0.001 | 0.001 | 0.001 | 0.001 | 0.001 | 0.001 | 0.001 | 0.001 |

		OPEX_{var}	€/kW_{th}	0	0	0	0	0	0	0	0
Gas storage interface (e-methane)¹⁰	CAPEX	€/kW_{th}	100	100	100	100	100	100	100	100	100
	OPEX	€/kW_{th a}	4	4	4	4	4	4	4	4	4
	OPEX_{var}	€/kW_{th}	0	0	0	0	0	0	0	0	0
	4) Unfortunately, ecoinvent is the only comprehensive database for LCA analysis. Sometimes the datasets are generated based on rather old data and do not contain specific information about the technology unit cost for the presented year. The CED value of a technology is typically estimated using the linear extrapolation method, which likely contributes to this uncertainty. This shortcoming is already mentioned in the limitations section of the manuscript. 5) We have not included non-physical costs in our assumptions. We explained the key drivers behind the changes in EROI trends and compared them with the corresponding LCOE values from the modelling results. In addition, wind fall profits from fossil fuels could be one of the main reasons for observing this inverse relationship between LCOE and EROI systems, where the cost of extracting fossil fuels (especially oil and gas) is lower than the market price. For this reason, we would like to point out that the relationship between price, costs, and EROI should be evaluated separately.										
iv. In short, based on your results someone could legitimately argue something akin to either of the following:	We believe that the relationship of EROI and LCOE is more complex than typically expected, also as the meaning and intent of these indicators are completely different. Please compare our results to previous results available in literature to avoid confusion. As mentioned earlier, the LCOE and EROI trends largely depend on the decisive factors that might alter their relationships. Moreover, the impact of enabling technologies on EROI should not be disregarded, as well as the excessive energy investment requirement for more fossil and nuclear fuel extractions. Our concern lies in generalising this relationship without in-depth analysis. Although the interpretation of the LCOE and EROI relationship in our manuscript is a simplified version that is not linked to either a background econometric analysis or directly to the modelling phase, the comparison with the studies based only on fossil-fuelled systems or oil/gas/coal sectors does not mean that same trends should be observed. This conclusion may not apply to highly RE systems, nor should the meaning of these important indicators be compromised, and they should not refute each other meaning. Since the complex nature of EROI and LCOE interaction on a system level is not yet well understood we better report the findings and motivate other researchers to intensify their investigations on that system-level approach. Chances may be not low that time delays of the										
1. (believe EROI, don't believe LCOE): "By showing that EROI declines in high decarbonation scenarios, and because EROI and LCOE are fundamentally inversely related, this indicates that the monetary cost assumptions are likely too optimistic."											
2. (believe LCOE, don't believe EROI): "The fact that both LCOE and EROI decrease for high decarbonization scenarios, and because EROI and LCOE are fundamentally inversely related, the most likely explanation is that the energy investments will likely be less than											

what the authors have assumed, and possibly the life cycle inventory data for cumulative energy demand are too high.”	indicators may be found in future, since LCOE may be a pre-indicator of the EROI metric, at least the results in this study indicate the existence of such an effect.
v. I’m not claiming you have to solve my conundrum in your manuscript, or that I know you have made a mistake. I am claiming that the interpretation of your result is confusing, thus not of clear value, with regard the LCOE vs. EROI trends. And, it is important to tell the reader (and the community) that we should understand what to expect in terms of LCOE vs. EROI trends. Thus, you should acknowledge that your trend in Figure 2(h) defies basic mathematical logic, and that there are some options to understand why (in future research). Of course you can correct me (that LCOE and EROI don’t have to be inversely related), but I need convincing.	Conclusions of a study, examining only the oil and gas sector based on price estimation relying on econometric analysis should be considered carefully for generalising LCOE and EROI studies, despite clear differences in the conceptualisation, methodology, and system differences, in particular for highly RE systems. Here we offer new insights for this scientific question from the perspective of different energy transition pathways especially highlighting the role of highly RE systems. We may conclude for now that the relationship of the metrics EROI and LCOE is more complex than initially thought if investigation in ongoing energy transitions and for the different natures of fossil-nuclear fuels (with limited resources and deviating cost-price relations) and RE and their enabling technologies (with rather abundant resources and closely linked cost-price relations). In some years from now, with more system-level and transition studies tracing both key metrics we may better understand the complex nature of this relationship. We hope that our explanation in this response document and the revisions in the R2 manuscript are sufficient to clarify this ongoing confusion. We cannot document an inverse relationship between LCOE and EROI in the results, while this may change for longer time periods beyond 2050, and which may be a further aspect for more studies on this important case. We are interpreting this relationship by only providing a relative trend analysis of LCOE and corresponding EROI values based on all the methods, data and analysis as carried out in this study.
e) To me, the comparison of LCOE to EROI is a core issue because the net energy community is essentially saying “EROI” and such metrics are important, but to convince ourselves and others that net energy metrics are useful, we need to explain how they should or should not relate to monetary or other metrics. My opinion in this review is that your current manuscript is exactly the place to discuss if you can or cannot make a meaningful comparison of EROI to LCOE, and the reason is that you are performing these calculations	We understand the Reviewer's comments. The presented study is using the impact assessment method to estimate physical EROI. As assumptions and methodology change, their trend may also change. We simply present a relative trend analysis of these two key indicators. In addition, we strongly believe that comparing this study's results with the completely different system studies will not be appropriate. To reach such an argument, a similar study should be presented, at least incorporating several enabling technologies and implementing a systemwide EROI methodology, and considering changes during the transition. For the relationship between LCOE and EROI, we expanded the interpretation of the results and defined the limitations and uncertainties related to this issue in the corresponding section of the manuscript. We hope that this confusion is clarified. Finally, we may preliminarily conclude that the relationship of EROI and LCOE is more complex than thought before, and to find that systemwide transition studies comparing fossil fuels-dominated and highly RE systems may be required, as there may be time delay effects or other not yet well understood factors.

using computational tools from your research group of which you are the experts. No one else should be more able to answer this question, and the question for your tools is not enough for an additional manuscript. I cannot recommend this paper without a reasonable comparison, explanation, or discussion of why both EROI and LCOE decline in your future scenarios, and (in addition to believing PROI type calculation is valuable,) I've suggested several uncertainties for inputs that could be part of the explanation, but of course there could be others.	We see critical implementation issues related to PROI estimation, which requires further research into how we can define the systemwide methodology and redefine systemwide PROI including enabling technologies. As already explained, these are completely different analyses that this divergence could serve as inspiration for future research. We have tried to highlight the reasons why two studies should not be compared. Please also notice the previous explanations on EROI and LCOE. We clearly understood the Reviewers' points and made necessary revisions to the R2 manuscript.
Minor Comments on the Article Content:	
3) Apologies I did not suggest a change here in version 1, but you write on page 1 “Fundamentally, it is a ratio that measures the amount of energy invested to deliver the energy need of the society.” a. Strictly, EROI does not say anything about “the energy need of society”. We are trying to use studies such as yours to understand what society “needs” in a general sense (e.g., what are the trade offs at low EROI?). So I suggest a reword that is more related to the strict mathematical definition, such as “... energy output of an energy system or technology relative to the energy invested to build and operate said system or technology.”	Thank you for this aspect. We narrowed down the description as you recommended and revised the sentence in the R2 manuscript as follows: Fundamentally, it is the ratio of the energy output of a system or a technology to the energy invested in building and operating that system or technology.
4) State (in 1 sentence if possible, or in an existing sentence) how CO2 costs are included in LCOE (on emissions of power plants only?) from the Aghahosseini, A. et al. (2023) study,	Thank you for this clarification. Instead of adding the new sentence, we decided to add this information to the existing sentence on Page 2. To prevent confusion, we corrected the word “price” to “costs”.

and if there are any “energy investments” associated with CO2 costs. The reason is that (1) the reader of this manuscript does not necessarily know that the existing study includes a price or cost on CO2 and (2) I can't tell if CO2 monetary cost translates to any energy investments (I understand the monetary costs dictate the mix of power plants via the cost-minimization in Aghahosseini, A. et al. (2023).)

In the modelling, we set certain CO₂ cost for fossil-fuelled power plants. This cost is calculated during the modelling phase and are not directly linked to or included in the EROI calculation.

LCOE of all scenarios peaks in the period 2020-2030 as the shifting from fossil fuel systems to renewable systems accelerates and CO₂ cost for emissions of the fossil-fuelled power plants is precluded after 2020, which brings additional burden on overall costs.

References:

1. IRENA. Renewable Capacity Statistics 2023. (2023). Available at: <https://www.irena.org/Publications/2023/Mar/Renewable-capacity-statistics-2023>. (Accessed: 6th September 2023)
2. Capellán-Pérez, I., de Castro, C. & Miguel González, L. J. Dynamic Energy Return on Energy Investment (EROI) and material requirements in scenarios of global transition to renewable energies. *Energy Strategy Reviews* **26**, 100399 (2019).
3. Aghahosseini, A. *et al.* Energy system transition pathways to meet the global electricity demand for ambitious climate targets and cost competitiveness. *Applied Energy* **331**, 120401 (2023).
4. Way, R., Ives, M. C., Mealy, P. & Farmer, J. D. Empirically grounded technology forecasts and the energy transition. *Joule* **6**, 2057–2082 (2022).
5. Lambert, J. G., Hall, C. A. S., Balogh, S., Gupta, A. & Arnold, M. Energy, EROI and quality of life. *Energy Policy* **64**, 153–167 (2014).
6. Solomon, A. A., Manjong, N. B. & Breyer, C. The necessity to standardise primary energy quality in achieving a meaningful quantification of related indicators. *Smart Energy* **12**, 100115 (2023).
7. Delannoy, L., Longaretti, P. Y., Murphy, D. J. & Prados, E. Peak oil and the low-carbon energy transition: A net-energy perspective. *Applied Energy* **304**, 117843 (2021).
8. Agora Energiewende. *Stromspeicher in der Energiewende*. (2014).
9. Breyer, C., Tsupari, E., Tikka, V. & Vainikka, P. Power-to-gas as an emerging profitable business through creating an integrated value chain. *Energy Procedia* **73**, 182–189 (2015).
10. Michalski, J. *et al.* Hydrogen generation by electrolysis and storage in salt caverns: Potentials, economics and systems aspects with regard to the German energy transition. *International Journal of Hydrogen Energy* **42**, 13427–13443 (2017).

REVIEWERS' COMMENTS

Reviewer #1 (Remarks to the Author):

I am satisfied with the R2 version of the manuscript and recommend publication.

Reviewer #3 (Remarks to the Author):

GENERAL COMMENTS

Thank you for a thorough set of statement addressing my and the other reviewers' comments on your initial submission.

Mainly, thank you for creating the supplemental Section 6 (as demonstrated in Figures S20 and S21) that was an honest attempt to address my concerns. I think they are more insightful than your main "systemwide EROI" result, even though for reasons I discuss (and I think I understand correctly) the results in Figures S20 and S21 still are not the closest approximation of "annual energy investment flows" that could be made given the data discussed in your Supplemental. However, I'm not 100% sure of the mathematics leading to Figures S20 and S21, and am asking for clarification. Given this clarification, I could suggest to the editor to "accept" if the equivalent of Figure S21 (or the data, yes I know a single figure could get messier) is put into the main body of the paper.

1) My main summary and understanding of my reviews and your responses is as follows:

- a. Am I correct in stating that your LCOE and "systemwide EROI" relate only to the global scale infrastructure required for electricity generation? That is to say, while your "systemwide EROI" does include fossil fuels required to build renewable electricity systems, it does not include delivering 100% of liquid, solid, or gaseous fuels that are, for example, burned for heat (e.g., within industrial production) or consumed in conversion devices used for transportation (e.g. fuel cells, combustion engines).
- i. The terms such as "global-level energy model" make it seem like you are simulating more than only power generation as compared to you using terms like "net-zero CO2 emission power system", LCOE (with E = "electricity"), and "electricity" (in Figure 1 y-axes).
- b. We both agree that it is mathematically possible for you to calculate "systemwide EROI" as you have done, but I suppose it is "electricity system EROI" only, right? To my understanding, you have done this calculation correctly in terms of the mathematics you state.
- i. My interpretation of your "systemwide EROI" in year T is "a weighted average of the EROI of each electricity technology, calculated at the time when it was assumed installed, for all electricity technologies assumed operating in year T."
- c. I think we both agree that your new calculations in Section 6 of Supplemental (using "annual energy investment flow") are not an approximation of a PROI calculation, that I'm defining as: $PROI = (\text{annual energy flow output of energy system}) / (\text{annual energy investment flow input into energy system})$.

However, your Figures S20 and S21 in Section 6 of supplemental do add some additional beneficent insights.

d. On your “systemwide EROI” calculation (or method):

i. You think it introduces more insight and clarity into the EROI conversation.

ii. I think it introduces more confusion into the EROI conversation. It makes people believe that monetary amortization and energy flow amortization are both possible; the former is possible, but not the latter. We can amortize money, but we cannot amortize energy consumption (consumption implying a rate of energy flow, or power).

MAJOR COMMENTS RELATED TO YOUR RESPONSES TO MY COMMENTS ON YOUR REVISION 2

2) Your response to me states: “To our knowledge, we have not found any other study that provides a systemwide EROI analysis while addressing existing shortcomings of EROI as we did in this study.”

a. What we are disagreeing on is that (1) I don’t actually understand the shortcomings of EROI that you are addressing and (2) you have introduced what I view as a major shortcoming, which is that what you call “systemwide EROI” is really a calculation that spreads all energy investment (CED) equally over all years of operation of a power plant and this violates what actually happens in real life. In the real world, we can amortize monetary spending but we cannot amortize energy consumption (as you are doing in your calculation). You think it is insightful to calculate your “systemwide EROI” calculation that aggregates information from all electricity generation technologies in the system, but I do not think it is insightful.

b. So the question in my head is, why should a paper, that is supposed to inform how the world consumes energy, be allowed to be published if it doesn’t even follow basic laws (via timing) of conservation of energy and mass? The be clear, your violation is that

i. First, your EROI calculation assumes a power plant can be installed in “year 1” and operate for some number of years (N_{years}) after that.

ii. Second, your EROI calculation assumes you spread the energy consumption of manufacturing and installation of the power plant over all N_{years} .

c. In short, you perform the following steps for handling life cycle energy inputs (per your Equation 2) that, for my example, sum to 10 over 6 years:

i. Step 1: get data indicating CED for manufacturing + installation up front, and also CED for operations. Example annual energy input: 5,1,1,1,1,1

ii. Step 2: you sum all CED into one number. Example total energy input: = 10

iii. Step 3: Spread out annual energy inputs (in defining your “systemwide EROI”, not a PROI as we agree) equally over the project lifetime. Example annual energy input: 1.67, 1.67, 1.67, 1.67, 1.67, 1.67

d. New Step 4: Your figures S20 and S21 use an “annual energy flow” that to my understanding is the following energy input flow: 10,0,0,0,0,0

3) Your response to me, in discussing PROI, states “The consideration of the time lag in the transition period is mostly not considered (distributing the energy requirements over the following next years).”

- a. -- Of course, and this has been my main point. In the real world there ABSOLUTELY IS NO DISTRIBUTING THE ENERGY (input) REQUIREMENTS (for manufacturing and construction of power plants) OVER THE FOLLOWING NEXT YEARS. We cannot consume 1/20th of the manufacturing and installation energy inputs to install 1/20th of a wind turbine each year for 20 years and simultaneously count 100% of its annual electricity generation each of those 20 years. It is my understanding that this is what your EROI "system" calculation is doing.
- b. Electricity generation is 0 MWh until the power plant is 100% installed, which means after it has consumed 100% of its energy investment related to manufacturing and construction. This is basic conservation of energy and mass, and physical principles. We can only distribute the monetary payments over "the following next years" which might involve the creation of new money (e.g., via loans from a bank or payments from a government that issues a currency) associated with manufacturing and construction.
- c. I think you understand my point here but we disagree in that you believe your "systemwide EROI" calculation is informative and clarifying on of some energy fundamentals related to net energy, and I don't think it is informative.

4) Thank you very much for writing Supplemental Section 6. Your response to me per your Supplemental Section 6 includes: "Although what we have presented is a simplified version of the annual energy investments for our methodology, still, it is not correct to define it as PROI estimation since the PROI is implemented without taking into account the overlapping lifespans of the technologies. The consideration of the time lag in the transition period is mostly not considered (distributing the energy requirements over the following next years)."

- a. I believe you are referring to the results plotted in Fig. S21 since that is a ratio. I'm OK if you don't want to call it PROI in your paper, because I understand (I think) why.
- b. I think the reason why you don't want to call it a PROI is that when you write (in your response to me) that "... the CED values of the technologies spread over their lifetime are converted to CED values per year ...", DOES THIS mean that you are summing all three of (i) CED associated with manufacturing and installation, (ii) CED associated with operating costs, and (iii) CED associated with decommissioning, and moving all three energy inputs to (approximately) the year of installation of the power plant?
- c. You also state in your response to me that "Also, performing proper PROI estimation is a separate research project requiring much more rigor." And also "Our recommendation to postpone the PROI study was due to the fundamental issues discussed above, data availability being one of them."
- d. QUESTION: So what exactly do you mean when you write that without more "rigor" and "data availability" you cannot do a proper PROI calculation? Your Supplementary data show you have separated CED of "construction and decommissioning" from CED of "operation". This is sufficient (but not perfect if you can't separate out "decommissioning") to provide more insight into the "annual energy investment flow" trend than what you have done in the current version (it seems to me, but you can correct me here). In Figures S20 and S21 it seems to me (correct me as needed) you are summing 100% of the all CED together in "year 1" of each technology's installation. This is to say, there is a more realistic scenario that is actually between the two different versions of results you show in Fig. S20 and S21 (version 1: your first published "systemwide EROI"; version 2: summing all CED as "annual energy investment flow") because both plotted results violate the timing (per year at least) of when energy

investment (consumption) would occur for each technology that could then be aggregated to a system-wide energy investment (consumption) per year, as a PROI calculation would best approximate.

e. I agree with you that a PROI calculation should not account for overlapping lifespans of technologies because it is not an applicable concern. For any given year, a given power plant (or oil well) is operating or it isn't. Thus, if it is (assumed to be) operating, it has operating costs (and energy inputs) and it is generating electricity (and thus included in the system-wide PROI calculation). If the technology has been decommissioned, then it has no operating costs and no electricity generation because it no longer exists. The only reason one needs to know the lifetime is to calculate when to remove it from the simulation, as you are doing in your simulations (I'm assuming, with confidence).

5) In your response to me (within a section that introduced the "net energy cliff" concept), you wrote: "Thus, we strongly believe that it is not right to apply the present and estimated fossil-based theories directly to a RE-based system."

a. What are you calling a "fossil-based" theory? Are you saying the EROI or PROI (what you and I are calling PROI) analysis related to the "net energy cliff" idea is a "fossil-based theory" that is not applicable mathematically and theoretically to renewable energy? If this is what you are suggesting, then I strongly disagree, or at least don't understand your argument. I can calculate the LCOE or EROI of a single oil well, gas well, or coal mine just as can be done for a single wind turbine or solar PV farm.

b. As far as I'm concerned, any given net energy metric (EROI, PROI, net external energy power ratio, etc.) is simply a method of how to perform mathematical calculations that is very much independent of the energy technology. I need you to explain why you say this.

c. What theory is "fossil-based"? What are its mathematics that cannot be applied to renewable energy, and why? Is there a net energy theory (or metric) that can only be applied to renewable energy, but not to "fossil-based" energy?

6) In your response to me you wrote: "In RE-based systems both the energy investment and technology cost is expected to improve with time"

a. --- Of course one can make this assumption, but this is a distraction for our discussion. Certain technology costs will be expected to possibly decrease also the case for fossil energy (i.e., horizontal drilling and hydraulic fracturing). Humans in general are seeking ways to improve technologies and many of these have spillover effects across industries. So if you are implying that renewable energy has a certain type of future technological progress that does not exist for fossil technologies, then that is an unsubstantiated claim. Renewables can also have depletion effects, for example, as you install more wind and solar you will have fewer places to install them, and these new places can have more political/social resistance and/or worse resource quality in the same way as fossil fuels will deplete along with mineral resources we mine for all human technologies (renewable energy as well as cars, computers, etc.).

7) Your response wrote: "5) We have not included non-physical costs in our assumptions. We explained the key drivers behind the changes in EROI trends and compared them with the corresponding LCOE values from the modelling results. In addition, wind fall profits from fossil fuels could be one of the main

reasons for observing this inverse relationship between LCOE and EROI systems, where the cost of extracting fossil fuels (especially oil and gas) is lower than the market price. For this reason, we would like to point out that the relationship between price, costs, and EROI should be evaluated separately.”

a. I’m not sure of your point. Windfall profits come because prices go high for reasons unrelated to costs of production. It does not act in the opposite direction that windfall profits come first and then prices go higher. The King and Hall (2011) paper show that the 1970s and 2007 data points are outside of the normal trend with prices higher than the inverse relationship would expect, but the inverse relationship holds even without the 1970s and 2007 data points.

b. It is likely that oil and gas prices are usually higher than average variable costs (as you suggest), because that is the definition of how energy markets are set up to work. If we include capital costs, this is less obvious as you imply. However, I’m not asking you to compare electricity prices (which you are not calculating) and costs to EROI, I’ve been only asking about LCOE and EROI.

8) Your response includes: “We believe that the relationship of EROI and LCOE is more complex than typically expected, also as the meaning and intent of these indicators are completely different.”

a. Yes, their meaning and intent is different, but I think by amortizing the energy investment (or CED of installation) over several years for each technology in trying to demonstrate how a “systemwide EROI” changes over time, this is too far removed from the meaning of EROI and it provides the appearance of representing energy flows when you in fact are not doing that in a realistic manner.

b. The idea of an annual LCOE cost can at least be contemplated because it inherently assumes amortization of “present money costs” into “annualized money costs” over a few decades. This same idea does not hold for EROI as you are using it, which is to say you are conceptually using EROI with units of energy (or power integrated over lifetime) in numerator and denominator, and then spreading those energy inputs (in the denominator) over time in a way that is DIFFERENT than the most realistic approximation (which is a sizeable up-front energy consumption in ‘year 1’ followed by smaller annual energy consumption during the rest of the power plant lifetime).

MAJOR COMMENTS ON ARTICLE CONTENT:

9) Your article now states its purpose is “... to evaluate the sustainability risk of these transition scenarios from the perspective of physical EROI. The overall modelling framework improved on the existing shortcomings of corresponding EROI studies and enhances the representativeness of the estimated physical EROI values by implementing a holistic approach for estimating primary energy quality at electricity level²⁹, creating a broader cumulative energy demand (CED) database for technologies based on life cycle assessment (LCA) databases²⁸, and integrating EROI estimation with energy system model output²⁷.”

a. I don’t think your paper “enhances the representativeness of the estimated physical EROI values” because it confuses and goes too far from the physical meaning of tracking energy input and output flows within the energy system.

b. When is EROI not physical? Why have “physical” in front of EROI when your system-wide EROI actually violates the physical flows of energy behind the EROI calculation?

10) With regard to relating LCOE and your systemwide EROI, your article states:

a. "Finally, this study also presents how EROI links to systemwide levelised cost of electricity (LCOE) and CO₂ emissions." and ... your new paragraph is ... "Further analysis of the EROI relation with system LCOE (Fig. 2h) shows that low-cost solutions correlate to low EROI. Replacing fossil and nuclear power with RE reduces the system's LCOE by lowering fuel and CO₂ costs. The EI for the upstream processes of the fuel chain initially account for a significant percentage of the total system EI. As the transition progresses, this share is replaced by the EI of the RE technologies and their enabling systems, and the addition of enabling technologies leads to a further decrease in EROI values. The level of reduction in the LCOE and EROI is related to the key decisive factors, which are system composition, technology selection, and the unit cost of technology. The observed trend may change depending on these decisive factors, and the interpretation here is presented by a relative trend analysis of two different indicators. This conclusion may change in future depending on the system types (predominantly fossil-nuclear powered or highly RE with enabling technologies), as the investigation of system level ET studies increases, and thus, the complex nature of this relationship can be better elucidated."

b. My conclusion is that this is not a sufficient explanation. Your "decisive factors" are just these general statements:

i. "system composition, technology selection, and the unit cost of technology."

c. You assume learning rates for declining CED (for solar for example, via Supplemental Eq. 1 and Table S3 and batteries as in Table S25), but it is hard to know how much this assumption matches any historical data, or at least the values in the cited literature (even of co-author Breyer). I see no similar CED "learning curve" (decreasing CED values) for other technologies.

11) (Apologies for not asking sooner) Do you calculate CED of any necessary transmission and distribution (grid) expansion? I don't see this described anywhere (I might be missing it) as to whether grid energy investments were included. I see a discussion of grid energy (heat) dissipation losses.

MINOR COMMENTS ON ARTICLE CONTENT:

12) You write "The modelling results reveal that the LCOE for all scenarios starts at 70.9 €/MWh in 2015 and gradually decreases as the RE deployment increases. This decline ranges from 45.2-49.7 €/MWh for BPS scenarios, 53.9-54.1 €/MWh for Teske/DLR scenarios, and 59.2-69.5 €/MWh for IEA scenarios⁶."

a. "This decline" does not refer to the values you state. Instead, the values you state refer to the 2050 values of your calculated LCOE. For example, for the "TESKE/DLR 2.0oC" scenario, the LCOE declines about 17 €/MWh from 71 €/MWh (in 2015) to about 54 €/MWh (in 2050). So the "decline" is 17 €/MWh, not 54 €/MWh.

b. Need to reword this to either refer to the actual decline amount (2050 LCOE minus 2015 LCOE) or to the final 2050 LCOE value.

13) Per your new paragraph starting “Furthermore, the sustainability risk of the systemwide EROI is reanalyzed ...”, you wrote “Also, the shortening of the ET triggers the need for more energy for RE investment in the BPS-plus scenarios, as observed in the annual energy investment flow (Fig. S19).”

a. I think you mean to refer to Fig. S20. Correct?

14) Per your new paragraph starting “Furthermore, the sustainability risk of the systemwide EROI is reanalyzed ...”, you wrote “Considering the CED in the year of the investment creates greater fluctuations in the annual energy investment flow during the ET (Fig. S21). Despite of a high expansion of RE, the estimated annual energy return on investment values stay in the low-risk range, as defined by Capellán-Pérez et al.²²” and separately state “. The analysis shows that the annual energy return on investment values remain in the low-risk range, as defined by Capellán-Pérez et al.², and not fall below 5.”

a. My main point is you have misinterpreted Capellán-Pérez et al. statement in their paper. The risk you refer to is for “EROI standard” and they state that there is:

i. No risk: > 15:1

ii. Low risk: 10:1 < EROI_{st} < 15:1

iii. Dangerous: 5:1 < EROI_{st} < 10:1

iv. Very Dangerous: EROI_{st} < 5:1

v. Unfeasible: EROI_{st} < 2:1 to 3:1

b. Your ratio based on annual energy return on investment plotted in Fig. S21 have some values are less than 10:1 for some years in some of the scenarios, thus in the Capellán-Pérez et al. “dangerous” zone for that more comparable metric to their paper.

c. Regardless, MY SUGGESTION is that I think you can just remove this sentence that refers to Capellán-Pérez et al. since you are not modeling 100% of the energy system (you can correct me) but only the electricity system (and its energy outputs and required energy inputs) and thus is not a good comparison to Capellán-Pérez et al. Capellán-Pérez et al. is modeling the entire energy system (electricity, liquid fuels, gaseous fuels) full set of input and outputs (e.g., total extraction of oil for EROI_{st} calculation, and total output of gasoline+diesel+jet fuel+etc. for EROI_{pou} calculation). However, they assume EROI of oil, gas, and coal are each constant.

15) Minor typo error: Remove “of” in “Despite of a high expansion of RE ...”

16) You responded to me with “It has nothing to do with the amount of calculation that we need to perform. Deriving a new methodology for the systemwide PROI is still one of the key challenges to performing this analysis.”

a. As far as I’m concerned, you have mostly satisfied my request (or desire) for using annualized energy flow via your Fig. S20 and S21, whether you call it PROI or not. Thank you. However, I’m still not 100% sure if the annualized input flows are:

i. 100% of CED brought to year 1 (or divided by 5 per year 5-year time step)? or

ii. CED values spread out between 2 times of annualized flows, per what you have in your supplemental:

(time 1): CED of installation and decommission (?) divided by 5 years when creating data points every 5 years, and (time 2) CED of operation used each time step.

iii. Something else?

b. If your calculations in Figure S20 and S21 can be clarified for me, then I might agree to accept publication of your paper, but I'd require (of course up to the editor) that Fig. S21 be placed in the main body of the paper and the mathematics of that calculation described in an equation.

MAJOR COMMENTS ON SUPPLEMENTAL:

17) Your response to me per your Supplemental Section 6 includes: "Furthermore, the dynamics of electricity generation and energy requirements are captured on a micro-level in PROI at technology level, which is different than system-level analysis where the interactions between technologies are present and reflected in the systemwide EROI analysis."

a. I'm not sure of your point, but I think it is incorrect. The PROI-like calculation using "annual energy investment flow" you have performed to create Figure S20 and S21 BY DEFINITION takes into account all technologies within the system. Right? The "annual energy investment flow" is the sum of annual energy investment for every technology you are assessing. Right?

b. I think you are using a caricature, or strawman, definition of PROI that does not have to hold. That is to say, a PROI calculation can be at a single technology level OR a system wide level simply by summing up each technology as you are doing (if you had the different parts of life cycle energy inputs timed properly).

c. My main argument within my review is that PROI is much more informative and appropriate for system-wide assessment than EROI, mainly because your EROI calculation is amortizing up-front energy investments that by definition (of physical laws) cannot happen in real life. And you have been arguing the opposite is true. Regardless, I have agreed that you can of course do the mathematics of your EROI as you are doing, but I'm also stating it is not very meaningful because of the distortion of the timing of mass and energy flows (it violates physical conservation laws) that is not informative to policymakers, but misinformative.

MINOR COMMENTS ON SUPPLEMENTAL

18) Please change "The annual energy investment flow (Fig. S20), which limits the lifetime effect, ..." to what I think is a more accurate distinction, because "lifetime effect" is not a clear term and not used anywhere else in the manuscript or supplemental (as far as my search indicates). I think you can rewrite it as something like:

a. "The annual energy investment flow (Fig. S20), which does not amortize the up-front energy investment over the power plant lifetime after installation, ..."

19) Your response to me stated (with regard to data plotted in Fig. S20):

a. "The same calculation method as the systemwide EROI is used, but the CED values of the technologies spread over their lifetime are converted to CED values per year and instead annual net capacities are taken."

b. What is the meaning of "annual net capacities are taken"?

Reviewer #1	Response to Reviewer #1
Reviewer comments:	Thank you for your valuable comments and suggestions. Kindly note that the changes are already reflected in the revised R3 manuscript via the track change mode. However, for the sake of clarity and to save time, we have also provided the direct changes made in accordance with Reviewers' comments in blue. The words/phrases in bold and underlined refer to specific changes. Apart from the moderate/minor changes requested by the Reviewers, the major ones are presented in the revised R3 manuscript.
I am satisfied with the R2 version of the manuscript and recommend publication.	Thank you for the constructive comments to improve the quality of this study.

Reviewer #3	Response to Reviewer #3
Reviewer comments:	Thank you for your valuable comments and suggestions. Kindly note that the changes are already reflected in the revised R3 manuscript via the track change mode. However, for the sake of clarity and to save time, we have also provided the direct changes made in accordance with Reviewers' comments in blue. The words/phrases in bold and underlined refer to specific changes. Apart from the moderate/minor changes requested by the Reviewers, the major ones are presented in the revised R3 manuscript.
General Comments	
Thank you for a thorough set of statement addressing my and the other reviewers' comments on your initial submission. Mainly, thank you for creating the supplemental Section 6 (as demonstrated in Figures S20 and S21) that was an honest attempt to address my concerns. I think they are more insightful than your main "systemwide EROI" result, even though for reasons I discuss (and I think I understand correctly) the results in Figures S20 and S21 still are not the closest approximation of "annual energy investment flows" that could be made given the data discussed in your Supplemental. However, I'm not 100% sure of the mathematics leading to Figures S20 and S21, and am asking for clarification. Given this clarification, I could suggest to the editor to "accept" if the equivalent of Figure S21 (or the data, yes I know a single figure could get messier) is put into the main body of the paper.	Thank you for acknowledging our efforts regarding Figure S20 and Figure S21. We relate all our responses to the given context of the manuscript and the objective of this study. The provided annual energy investment flow is the best approximation that we can provide, and we clearly stated before that it is not a PROI estimation. The main reasons have been discussed in R1 and R2. We believe the systemwide EROI methodology provides detailed insights into energy transition modelling tools and researchers to execute respective EROI studies. The annual energy investment flow could be a complement to PROI, but running the energy modelling annually could require a lot of computational effort, time, and data. The explanation regarding the annual energy investment flow estimation is explained in this file. As requested from Reviewer #3, we put Figure S21 to the main manuscript. This will be helpful to readers. We would like to emphasise that the core aim of this paper is the systemwide EROI, neither the PROI nor annual energy flow investment analysis.
1) My main summary and understanding of my reviews and your responses is as follows: a. Am I correct in stating that your LCOE and "systemwide EROI" relate only to the global scale infrastructure required for electricity generation? That is to say, while your "systemwide EROI" does include fossil fuels required to build renewable electricity systems, it does not include delivering 100% of liquid,	All data related to the modelling phase refers to the power sector and does not cover the heating and transport sectors, as pointed out in the manuscript. With decarbonisation in the power sector, the use of fossil fuels is drastically decreasing as part of them are being replaced by e-methane, biomethane and hydrogen as fuels, especially for LUT and Teske/DLR scenarios. The detailed explanation of the scenarios can be found in Table 1 and the study of Aghahosseini et al.¹.

solid, or gaseous fuels that are, for example, burned for heat (e.g., within industrial production) or consumed in conversion devices used for transportation (e.g. fuel cells, combustion engines). i. The terms such as “global-level energy model” make it seem like you are simulating more than only power generation as compared to you using terms like “net-zero CO2 emission power system”, LCOE (with E = “electricity”), and “electricity” (in Figure 1 y-axes).	LUT-ESTM is a global-scale energy model that provides the flexibility to run models at global, regional, and country levels. The global energy transition scenarios were simulated for the World structured in nine major regions. Additionally, the model has a sector cascade system that allows modelling one sector at a time and/or multiple sectors together. However, to avoid further misunderstandings, we have corrected this term in the manuscript as follows: The global-level power sector model presents an aggregated version of all nine major regions.
b. We both agree that it is mathematically possible for you to calculate “systemwide EROI” as you have done, but I suppose it is “electricity system EROI” only, right? To my understanding, you have done this calculation correctly in terms of the mathematics you state. i. My interpretation of your “systemwide EROI” in year T is “a weighted average of the EROI of each electricity technology, calculated at the time when it was assumed installed, for all electricity technologies assumed operating in year T.”	Yes, the systemwide EROI analysis provided for the power sector. We explained this study design in R1 and R2. The systemwide EROI is implemented by covering all technologies by a given year (new and existing). We do not calculate the EROI of some technology and then use that EROI in the energy mix to determine the overall EROI for a system, which is common in EROI literature. Instead, we create a boundary condition that fits to the system as one to estimate the EROI. However, a separate technology-by-technology EROI calculation was roughly performed to check the credibility of our results. The technology-level EROIs are not used to estimate systemwide EROI. A technology-level EROI already reaches limitations in properly integrating storage. We generated CED values (construction and decommissioning phases) and operational CED (using full load hours, lifetime, and fuel CED values) for each technology. Then, we estimated the CED requirement through the lifetime of technology, and then we annualised this estimated value by dividing technology lifetimes. For key technologies, these values show the difference every 5 years. They are later multiplied by capacity values taken from energy system transition modelling outputs. After the calculation of the total energy invested for the overall system at the given year, the estimated total electricity generation for those years is divided by the total energy invested for the overall system and finally, we have systemwide EROI values for every 5 years.
c. I think we both agree that your new calculations in Section 6 of Supplemental (using “annual energy investment flow”) are not an approximation of a PROI calculation, that I’m defining as: PROI = (annual energy flow output of energy system)/(annual energy	We agree to the Reviewer #3 that what we have provided is not a PROI but the annualised energy investment flow, in particular when it comes to the temporal CED allocation. As explained in the R2, we face several challenges in implementing PROI, especially for the specific technologies (such as DAC, methanation unit, electrolyser,

investment flow input into energy system). However, your Figures S20 and S21 in Section 6 of supplemental do add some additional beneficent insights.	and different storage options), which are enabling technologies. Estimating the PROI of these technologies requires extensive analysis as their output differs from electricity. This issue is discussed in more detail in the comments below. Additionally, we thank Reviewer #3 for acknowledging our efforts in this matter.
d. On your “systemwide EROI” calculation (or method): i. You think it introduces more insight and clarity into the EROI conversation. ii. I think it introduces more confusion into the EROI conversation. It makes people believe that monetary amortization and energy flow amortization are both possible; the former is possible, but not the latter. We can amortize money, but we cannot amortize energy consumption (consumption implying a rate of energy flow, or power).	Yes, we strongly believe that this paper provides more insight into the EROI literature. It provides a holistic approach, and the reiteration can be executed for other modelling tools. Moreover, the CED database is structured specifically for technologies in the power sector, which can be used in other studies other than EROI. The Energy Payback Time (EPBT) approach is widely used in energetic considerations. The EPBT describes when an energy system delivers the amount of energy equal to the amount needed to establish the system. This documents that energetic amortisation is a standard approach in science. The study of Bhandari et al.² provides the relationship between EROI and EPBT. In this paper, EROI is defined in terms of EPBT.
Major Comments Related to R2 Revisions	
2) Your response to me states: “To our knowledge, we have not found any other study that provides a systemwide EROI analysis while addressing existing shortcomings of EROI as we did in this study.” a. What we are disagreeing on is that (1) I don’t actually understand the shortcomings of EROI that you are addressing and (2) you have introduced what I view as a major shortcoming, which is that what you call “systemwide EROI” is really a calculation that spreads all energy investment (CED) equally over all years of operation of a power plant and this violates what actually happens in real life. In the real world, we can amortize monetary spending but we cannot amortize energy consumption (as you are doing in your calculation). You think it is insightful to calculate your “systemwide EROI” calculation that aggregates information from all electricity generation technologies in the system, but I do not think it is insightful.	We disagree to the Reviewer#3. (1) The shortcomings of the EROI were discussed in R1 and R2. We simply don’t calculate the EROI of some technology and then use that EROI in the energy mix to determine the overall EROI for a system, which is common in EROI literature. Furthermore, this paper provides (i) estimating primary energy quality at electricity level, (ii) providing a comprehensive CED database based on LCA for the power sector, (iii) incorporates nascent technologies and integrating systemwide EROI estimation with energy system model output. Different perspectives are required, similar to LCA where all LCA components are framed on a performance unit, as an internationally and scientifically acknowledged too. We apply that approach on EROI with the functional unit of energy system service (here electricity). Our approach is aligned to the LCA perspective, using CED and electricity as final energy demand. The annual energy investment flow analysis adds insights for the case that the CED is allocated to the period of investment. These two considerations deliver a holistic view on the performance of the energy system from an energy investment and delivered services point of view. The shortcomings are also addressed in the manuscript, which is given in below.

	The pioneering EROI studies present diverse concepts due to methodological inconsistencies^{7,8}, with some deriving varieties of EROI concepts⁹⁻¹², implementing different boundary conditions¹³⁻¹⁵, comparing fossil fuel and RE technologies with or without enabling technologies¹⁶⁻¹⁹. However, recent studies²⁰⁻²³ diverted this attention towards EROI analysis for ETs^{24,25} to foresee the feasibility of 100% RE systems²⁶. Some of these studies questioned the plausibility of 100% RE in terms of net energy production^{20,22}. A recent study that implemented an advanced systemwide EROI approach has challenged these studies' conclusions²⁷ and suggested that fundamental gaps of existing EROI estimation techniques, such as the inability to capture the impact of optimal interoperability of multi-processes, and the typical methodological gaps of EROI²⁸, may lead to such conclusions. Thus, applying an improved systemwide EROI tool to a global transition scenario can eliminate these salient issues while simultaneously contributing to enhancing the ET path selection. Regarding the amortisation, kindly revisit the previous comment. PROI follows an energy flow analysis while EROI relies on impact assessment (ISO 140409 and ISO 140410). All of the methodology that we structure is based on the impact assessment methodology (LCIs). We are aiming for a systemwide EROI, not an annual energy investment flow or PROI. Notably, generated CED values that we are using are also taken from impact assessment database where the background estimations of those values rely on the impact assessment methodology as well. Our aim is to provide an insight about the designed energy transition scenarios from an EROI perspective. We understand the concern of Reviewer #3 that is why we annualised the energy investment flow of our system and put two different approaches in the manuscript. Nevertheless, the systemwide EROI methodology is fully based on the well acknowledged impact assessment methodology.
b. So the question in my head is, why should a paper, that is supposed to inform how the world consumes energy, be allowed to be published if it doesn't even follow basic laws (via timing) of conservation of energy and mass? The be clear, your violation is that i. First, your EROI calculation assumes a power plant can be installed in "year 1" and operate for some number of years (N_years) after that.	To make it clear, we follow the principles of impact assessment, as pointed out in the previous response. The transition is described in intervals of five years. All relevant technologies assumed are built on such a period, most in a considerably shorter period of time. For acknowledging energy investment flow considerations, we added such a perspective. The methodology of this paper, and regarding explanation related to construction, operation and decommissioning phases are discussed in the manuscript and explained in R1 and R2, kindly revisit the Major Comments in R2. We have made it clear that we remain on the side of impact assessment and added the annual energy flow analysis for

ii. Second, your EROI calculation assumes you spread the energy consumption of manufacturing and installation of the power plant over all N_years.	additional insights. We developed our methodology that can be easily integrated in energy modelling tools for the power sector. We strongly believe that we reflected a solid proximation. All energy modelling, EROI, PROI and LCA methods are or are not applicable in real life, as they all involve uncertainties, and all are built on assumptions and limited data.
c. In short, you perform the following steps for handling life cycle energy inputs (per your Equation 2) that, for my example, sum to 10 over 6 years: i. Step 1: get data indicating CED for manufacturing + installation up front, and also CED for operations. Example annual energy input: 5,1,1,1,1,1 ii. Step 2: you sum all CED into one number. Example total energy input: = 10 iii. Step 3: Spread out annual energy inputs (in defining your “systemwide EROI”, not a PROI as we agree) equally over the project lifetime. Example annual energy input: 1.67, 1.67, 1.67, 1.67, 1.67 d. New Step 4: Your figures S20 and S21 use an “annual energy flow” that to my understanding is the following energy input flow: 10,0,0,0,0,0	We discussed this issue in R2. Kindly revisit the relevant comments. We would like to reiterate that the spread-out annual energy inputs for the key technologies in the systemwide EROI are not the same because the lifetimes of these technologies and the energetic learning related CED for new installations change throughout the transition, in particular the CED for solar PV and battery systems. In Figure S20 and S21, we used the systemwide energy investment flow (5-year period) as a basis for an annual energy investment flow while categorised the energy investment types according to their intended use, i.e., upstream supply chain for fuel production, operation, and investment. The same calculation method as the systemwide EROI is used, but we converted to CED values per year. In other words, 100% of CED values for construction and decommissioning phases are divided by 5 per year for 5-year time step, so that annual values can be considered, the other energy flow values (such as for operations) remain on an annual level.
3) Your response to me, in discussing PROI, states “The consideration of the time lag in the transition period is mostly not considered (distributing the energy requirements over the following next years).” a. -- Of course, and this has been my main point. In the real world there ABSOLUTELY IS NO DISTRIBUTING THE ENERGY (input) REQUIREMENTS (for manufacturing and construction of power plants) OVER THE FOLLOWING NEXT YEARS. We cannot consume 1/20th of the manufacturing and installation energy inputs to install 1/20th of a wind turbine each year for 20 years and simultaneously count 100% of its annual electricity	This is also our point, because we are not looking for a PROI or an annual energy investment flow. We follow the impact assessment logic, as in LCA. Lifetime is important to us because in the modelling tool the lifetime of technologies changes, and we work with 5-year data intervals. This is one of the reasons that PROI methodology does not fit our model structure, kindly see the previous comments and discussions in R1 and R2. For an additional perspective we added the annual energy investment flow in the R2. The brief explanation from previous revision (R2) is given in below. The present systemwide EROI relies on the LUT-ESTM result, which is modelled in a 5-year step, meaning if we take a capacity installed in 2025. That capacity is assumed to generate electricity starting on 1 January that year. All construction works should be completed and connected to the grid by 31 December 2024 (in reality the capacities are built and commissioned about 2.5 years before and after that point in time). The annual historical capacity growth data of renewable

generation each of those 20 years. It is my understanding that this is what your EROI “system” calculation is doing.	technologies could provide good information on how it is distributed over time for various technologies³. Thus, the lack of overlap between energy generation and energy investment (EI) for construction/manufacturing/installation is clear in this format. The corresponding EI occurs around 2025 for these systems. Thus, it is not possible to perform a correct PROI estimation since it involves time transfer of energy flow.
b. Electricity generation is 0 MWh until the power plant is 100% installed, which means after it has consumed 100% of its energy investment related to manufacturing and construction. This is basic conservation of energy and mass, and physical principles. We can only distribute the monetary payments over “the following next years” which might involve the creation of new money (e.g., via loans from a bank or payments from a government that issues a currency) associated with manufacturing and construction.	Please see the discussion in R2 and previous comments regarding this issue.
c. I think you understand my point here but we disagree in that you believe your “systemwide EROI” calculation is informative and clarifying on of some energy fundamentals related to net energy, and I don’t think it is informative.	We respect the opinion of Reviewer #3, but we strongly believe that this study provides informative findings, and the proposed methodology is specifically suitable for energy modelling. We apply the impact assessment logic, as in LCA. The aim of this paper is neither to implement PROI methodology nor develop an alternative annual energy investment flow for energy modelling tools. It provides a systemwide EROI methodology.
4) Thank you very much for writing Supplemental Section 6. Your response to me per your Supplemental Section 6 includes: “Although what we have presented is a simplified version of the annual energy investments for our methodology, still, it is not correct to define it as PROI estimation since the PROI is implemented without taking into account the overlapping lifespans of the technologies. The consideration of the time lag in the transition period is mostly not considered (distributing the energy requirements over the following next years).” a. I believe you are referring to the results plotted in Fig. S21 since that is a ratio. I’m OK if you don’t want to call it PROI in your paper, because I understand (I think) why.	We explained this in the previous comments that it is not a PROI, it is the best approximation we can have. We annualised all CED values per technology and used the capacities to estimate total CED for the system for a year. This is executed for all phases (construction, operation, and decommissioning). After that EROI estimation is carried out. We do not intend to deliver technology-level values, but system-level values, in a way how systems are built, evolutionary over time combining older and new investments in a continued development. For that we apply a novel methodology that follows the impact assessment logic.

b. I think the reason why you don't want to call it a PROI is that when you write (in your response to me) that "... the CED values of the technologies spread over their lifetime are converted to CED values per year ...", DOES THIS mean that you are summing all three of (i) CED associated with manufacturing and installation, (ii) CED associated with operating costs, and (iii) CED associated with decommissioning, and moving all three energy inputs to (approximately) the year of installation of the power plant?	
c. You also state in your response to me that "Also, performing proper PROI estimation is a separate research project requiring much more rigor." And also "Our recommendation to postpone the PROI study was due to the fundamental issues discussed above, data availability being one of them." d. QUESTION: So what exactly do you mean when your write that without more "rigor" and "data availability" you cannot do a proper PROI calculation? Your Supplementary data show you have separated CED of "construction and decommissioning" from CED of "operation". This is sufficient (but not perfect if you can't separate out "decommissioning") to provide more insight into the "annual energy investment flow" trend than what you have done in the current version (it seems to me, but you can correct me here). In Figures S20 and S21 it seems to me (correct me as needed) you are summing 100% of the all CED together in "year 1" of each technology's installation. This is to say, there is a more realistic scenario that is actually between the two different versions of results you show in Fig. S20 and S21 (version 1: your first published "systemwide EROI"; version 2: summing all CED as "annual energy investment flow") because both plotted results violate the timing (per year at least) of when energy investment (consumption) would occur for each technology that could then be	PROI and EROI should not be used in the same meaning because they do not deliver equivalent values. EROI and PROI differ methodologically. They have different temporal and spatial boundaries. Thus, the two indicators (EROI and PROI) should be separated but not substituted for one another. Moreover, our system includes enabling technologies (DAC, methanation unit, electrolyser, and various storage options). To implement a system-level PROI estimation, we need to find a way to integrate these technologies and redefine the system boundaries for specific technologies. To the authors' knowledge, we have not encountered any study in the PROI literature that addresses these technologies or considers them as enabling technologies for power systems. Also, the primary energy quality issue remains and is an unsolved issue for PROI as well. Again, we would like to emphasise that we used the systemwide energy investment flow (5-year period) as a basis for an annual energy investment flow, yet it cannot be said that we captured the time transfer of energy flows correctly as implemented in a PROI. Furthermore, Figures S20 and S21 are the best option that we can provide. These figures surely provide additional insight, this is why all authors agreed to move the Fig. S21 to the R3 revised main manuscript. However, the conceptualisation of this paper is not developing an alternative option to PROI or implement a systemwide PROI; rather we intended to see the power system as one and implemented the systemwide EROI methodology while staying on the impact assessment side (LCA).

aggregated to a system-wide energy investment (consumption) per year, as a PROI calculation would best approximate.	
e. I agree with you that a PROI calculation should not account for overlapping lifespans of technologies because it is not an applicable concern. For any given year, a given power plant (or oil well) is operating or it isn't. Thus, if it is (assumed to be) operating, it has operating costs (and energy inputs) and it is generating electricity (and thus included in the system-wide PROI calculation). If the technology has been decommissioned, then it has no operating costs and no electricity generation because it no longer exists. The only reason one needs to know the lifetime is to calculate when to remove it from the simulation, as you are doing in your simulations (I'm assuming, with confidence).	Kindly see previous comments about why lifetime is important to us and how EROI methodology is better suited for comprehensive energy system modelling compared to PROI. For comprehensive systems with many dozens of technologies, implementing PROI may not be a good option due to the aforementioned challenges. We are confident about our methodology and implementation.
5) In your response to me (within a section that introduced the “net energy cliff” concept), you wrote: “Thus, we strongly believe that it is not right to apply the present and estimated fossil-based theories directly to a RE-based system.” a. What are you calling a “fossil-based” theory? Are you saying the EROI or PROI (what you and I are calling PROI) analysis related to the “net energy cliff” idea is a “fossil-based theory” that is not applicable mathematically and theoretically to renewable energy? If this is what you are suggesting, then I strongly disagree, or at least don't understand your argument. I can calculate the LCOE or EROI of a single oil well, gas well, or coal mine just as can be done for a single wind turbine or solar PV farm. b. As far as I'm concerned, any given net energy metric (EROI, PROI, net external energy power ratio, etc.) is simply a method of how to perform mathematical calculations that is very much independent of the energy technology. I need you to explain why you say this.	We would like to remind Reviewer #3 that this discussion is related to the arguments about the relationship between EROI and LCOE. The question as directed to us in the R2 is given in below: But what are decision makers supposed to think when you tell them there will be less net energy in our scenarios (which is normally interpreted as a higher cost outcome for development) at same time as lower monetary cost? We gave our response in the R2 about what decision-makers should focus and how the results should be interpreted using the net energy cliff concept. The suggested study shows the inverse relationship between EROI and LCOE by relying on a case study using fossil-fuel based systems for examining only US oil and gas sector. Our response is given in below. The presented study examines EROI for different energy transition pathways, which takes the system from a fossil to a renewable dominated system. Fossil consumption takes a notable share of both the required energy investment and LCOE from the present system, in the future this component will significantly diminish or fully be eradicated depending on the scenario. Thus, we strongly believe that it is not right to apply the present and estimated fossil-based theories directly

c. What theory is “fossil-based”? What are its mathematics that cannot be applied to renewable energy, and why? Is there a net energy theory (or metric) that can only be applied to renewable energy, but not to “fossil-based” energy?

to a RE-based system. In RE-based systems both the energy investment and technology cost is expected to improve with time, thus, such comparison should be made with the full understanding of each systems characteristics because each EROI value change over the years related to different systems even for the same scenario.

The discussion is not based on the not applicability of PROI or EROI to a fossil fuel based system. Simply, we pointed out that it would not be appropriate to generalise a result based on a case study built on a fossil-fuelled system to RE-based systems. As outlined below, the reported relationship may change over time and depends on several factors explained in the manuscript. Hereby, we quote the abstract of the mentioned paper in here, which was published in 2011. The abstract clearly states that (i) the background data is energy prices, not cost, and the trends are generated using the oil and gas production sector, not renewables.

*“In this paper we derive relations among the biophysical characteristic of an energy resource in relation to the businesses and technologies that exploit them. These relations include the energy return on energy investment (EROI), the price of energy, and the profit of an energy business. **Our analyses show that EROI and the price of energy are inherently inversely related such that as EROI decreases for depleting fossil fuel production, the corresponding energy prices increase dramatically. Using energy and financial data for the oil and gas production sector, we demonstrate that the equations sufficiently describe the fundamental trends between profit, price, and EROI. For example, in 2002 an EROI of 11:1 for US oil and gas translates to an oil price of 24 \$2005/barrel at a typical profit of 10%. This work sets the stage for proper EROI and price comparisons of individual fossil and renewable energy businesses as well as the electricity sector as a whole. Additionally, it presents a framework for incorporating EROI into larger economic systems models.**”⁴.*

In this study, we use life cycle inventories (LCIs) to derive CED values for technologies that depend on material type, size, material consumption and defined system boundaries. Any change in the type of technology LCI will result in different CED values and ultimately affects the EROI estimate. Furthermore, we used the expression of “fossil-fuel theory” in here to emphasise that findings of this paper using the background oil and gas sector prices should not be generalised to RE systems. This is valid for the present studies using similar background data. Additionally, the discussed article was

	published in 2011, and the used data differ from the starting value of our EROI estimations, which is 2015.
6) In your response to me you wrote: “In RE-based systems both the energy investment and technology cost is expected to improve with time” a. --- Of course one can make this assumption, but this is a distraction for our discussion. Certain technology costs will be expected to possibly decrease also the case for fossil energy (i.e., horizontal drilling and hydraulic fracturing). Humans in general are seeking ways to improve technologies and many of these have spillover effects across industries. So if you are implying that renewable energy has a certain type of future technological progress that does not exist for fossil technologies, then that is an unsubstantiated claim. Renewables can also have depletion effects, for example, as you install more wind and solar you will have fewer places to install them, and these new places can have more political/social resistance and/or worse resource quality in the same way as fossil fuels will deplete along with mineral resources we mine for all human technologies (renewable energy as well as cars, computers, etc.).	We would like to quote from a recent press release by IRENA⁵ that compares renewable power to fossil fuels. “Renewable Power Generation Costs in 2021, published by the International Renewable Energy Agency (IRENA) today, shows that almost two-thirds or 163 gigawatts (GW) of newly installed renewable power in 2021 had lower costs than the world’s cheapest coal-fired option in the G20. IRENA estimates that, given the current high fossil fuel prices, the renewable power added in 2021 saves around USD 55 billion from global energy generation costs in 2022”⁵. The previous discussion is based on the relationship between LCOE and EROI, where LCOE is estimated based on technology costs. We did not claim that fossil fuel technologies may not also improve over time. However, given market trends and investments, improvements in fossil fuel technologies may require more time, while overall available resources may determine the cost most. For instance, according to the World Energy Investment report published by the IEA, the investment in clean energy (24%) is risen much faster than the investment in fossil-fuels (15%) in the year of 2021-2023 ⁶. Although this does not entirely mean that there will be no progress. Another point is that 7 out of 9 scenarios in this study aim for the decarbonisation of the power sector by 2050. Only IEA scenarios still rely mostly on gas and nuclear technologies. On the other hand, the depletion effect for renewables may be observed locally, but not on a global scale. Such effects, however, are also part of the RE resources as used, as a maximum area potential per region is assumed and a weighted average for the better half of the resource sites is applied as can be found in the underlying studies, so that effectively a wider range of resource quality is considered. As we mentioned in the manuscript, we examine the system from a techno-economic perspective, not in terms of social context. We agree with Reviewer #3 that the depletion of mineral resources impacts not only fossil fuels but also renewables. Therefore, the next research will be based on this topic (critical raw material analysis) using the LCIs provided in this study.

7) Your response wrote: “5) We have not included non-physical costs in our assumptions. We explained the key drivers behind the changes in EROI trends and compared them with the corresponding LCOE values from the modelling results. In addition, wind fall profits from fossil fuels could be one of the main reasons for observing this inverse relationship between LCOE and EROI systems, where the cost of extracting fossil fuels (especially oil and gas) is lower than the market price. For this reason, we would like to point out that the relationship between price, costs, and EROI should be evaluated separately.” a. I’m not sure of your point. Windfall profits come because prices go high for reasons unrelated to costs of production. It does not act in the opposite direction that windfall profits come first and then prices go higher. The King and Hall (2011) paper show that the 1970s and 2007 data points are outside of the normal trend with prices higher than the inverse relationship would expect, but the inverse relationship holds even without the 1970s and 2007 data points.	This comment is given as response to the inverse relationship between LCOE and EROI based on the findings of King and Hall (2011)⁴. The mentioned study relies on “price” rather than “cost”, which is explicitly given in the abstract and in the figures in this paper. As price includes a significant portion of the profits generated by market conditions. This impact is higher in fossil fuels. For example, while the price of crude oil in the market is over \$80/bbl, the production cost is around \$5-20/bbl for the largest proportion of the market. Since this paper is based on gas and oil prices, it is inevitable that this large profit margin will have an impact on the trends. Importantly, the data used in the mentioned study is very old (1970s-2007) and only US oil and gas sector is examined whereas here we provide results at a global level. The system LCOE for a highly RE-based system phases out such enormous differences of costs and prices, as the weighted average cost of capital includes the profits for the investors are part of the return on equity.
b. It is likely that oil and gas prices are usually higher than average variable costs (as you suggest), because that is the definition of how energy markets are set up to work. If we include capital costs, this is less obvious as you imply. However, I’m not asking you to compare electricity prices (which you are not calculating) and costs to EROI, I’ve been only asking about LCOE and EROI.	Kindly we ask to revisit the previous comments. The price for fossil fuels is considered in LCOE and the higher the fossil energy share in the mix, the more the fossil fuel prices impact the LCOE, while the higher the RE share in the mix the less this effect matters.
8) Your response includes: “We believe that the relationship of EROI and LCOE is more complex than typically expected, also as the meaning and intent of these indicators are completely different.” a. Yes, their meaning and intent is different, but I think by amortizing the energy investment (or CED of installation) over several years for each technology in trying to demonstrate how a “systemwide EROI” changes over time, this is too far removed	Energy Payback Time (EPBT) is a clear example how energetic amortisation is considered a standard approach in energy analyses. Kindly see our previous comments. Few studies provides direct relationship between EPBT and EROI. Moreover, considering prolonging the discussion in the R1 and R2, we clearly explained our methodology. The aim of this study is developing and implementing a systemwide EROI based on the impact assessment logic, not an annual energy flow analysis or PROI. We clearly mentioned this in our manuscript, and previous discussion. Also, CED

from the meaning of EROI and it provides the appearance of representing energy flows when you in fact are not doing that in a realistic manner.	values (installation over years) are changing depending on energy learning curves, and we are using a 5-year period in our calculations to estimate the systemwide EROI. The lifetime of the technologies in the modelling tool also changes for key technologies in certain years, which affects the estimated CED values, too.
b. The idea of an annual LCOE cost can at least be contemplated because it inherently assumes amortization of “present money costs” into “annualized money costs” over a few decades. This same idea does not hold for EROI as you are using it, which is to say you are conceptually using EROI with units of energy (or power integrated over lifetime) in numerator and denominator, and then spreading those energy inputs (in the denominator) over time in a way that is DIFFERENT than the most realistic approximation (which is a sizeable up-front energy consumption in ‘year 1’ followed by smaller annual energy consumption during the rest of the power plant lifetime).	We would like to emphasise that the core aim of this paper is the systemwide EROI, neither the PROI nor annual energy flow investment analysis. If the accuracy or the best proximation is the major problem in here, we must first ask to what extent the energy modelling, EROI, PROI and LCA methods are or are not applicable in real life, as they all involve uncertainties, the assumptions and built on using the limited data. Notably, in PROI analysis, the background data mostly is taken from LCA studies' findings. Nevertheless, following the Reviewer #3 suggestion, we agreed to annualise our estimate to provide readers with more information. We thank the Reviewer #3 for this suggestion.
Major Comments on Article Content	
9) Your article now states its purpose is “... to evaluate the sustainability risk of these transition scenarios from the perspective of physical EROI. The overall modelling framework improved on the existing shortcomings of corresponding EROI studies and enhances the representativeness of the estimated physical EROI values by implementing a holistic approach for estimating primary energy quality at electricity level²⁹, creating a broader cumulative energy demand (CED) database for technologies based on life cycle assessment (LCA) databases²⁸, and integrating EROI estimation with energy system model output²⁷.” a. I don’t think your paper “enhances the representativeness of the estimated physical EROI values” because it confuses and goes too	We have clearly stated the purpose of our work and our methodology reflects the physical EROI concept. We strongly believe in capturing the concept of physical EROI. As previously mentioned, PROI and EROI are conceptually different from one another. The aim of this paper is the systemwide EROI methodology, based on the impact assessment logic, not the implementation of PROI. However, considering the prolonged discussion, we provided the annualised version of the systemwide EROI in the manuscript. Two versions of our estimates are now part of the manuscript. We agree with Reviewer #3 that these two perspectives enrich the discussion in the context of the revised manuscript.

far from the physical meaning of tracking energy input and output flows within the energy system.	
b. When is EROI not physical? Why have “physical” in front of EROI when your system-wide EROI actually violates the physical flows of energy behind the EROI calculation?	The methodology of this work was described in detail in the manuscript and the connection to the physical EROI concept as well as the derivation steps were given. We would like to emphasise again that our goal is not PROI, but EROI. The implementation of PROI and EROI differs in terms of the estimation methods and the data used. We explicitly point out this difference in the R1 and R2. PROI is neither the concept nor the goal of this study.
10) With regard to relating LCOE and your systemwide EROI, your article states: a. “Finally, this study also presents how EROI links to systemwide levelised cost of electricity (LCOE) and CO2 emissions.” and ... your new paragraph is ... “Further analysis of the EROI relation with system LCOE (Fig. 2h) shows that low-cost solutions correlate to low EROI. Replacing fossil and nuclear power with RE reduces the system's LCOE by lowering fuel and CO2 costs. The EI for the upstream processes of the fuel chain initially account for a significant percentage of the total system EI. As the transition progresses, this share is replaced by the EI of the RE technologies and their enabling systems, and the addition of enabling technologies leads to a further decrease in EROI values. The level of reduction in the LCOE and EROI is related to the key decisive factors, which are system composition, technology selection, and the unit cost of technology. The observed trend may change depending on these decisive factors, and the interpretation here is presented by a relative trend analysis of two different indicators. This conclusion may change in future depending on the system types (predominantly fossil-nuclear powered or highly RE with enabling technologies), as the investigation of system level ET studies increases, and thus, the complex nature of this relationship can be better elucidated.” b. My conclusion is that this is not a sufficient explanation. Your “decisive factors” are just these general statements:	We understand the Reviewer #3 points, so have therefore expanded “the decisive factors”. We hope that the R3 revised version reflects key decisive factors that we discussed in the R2. This sentence is revised in the manuscript as follows: The level of reduction in the LCOE and EROI is related to the key decisive factors, which are system composition, the type of technology, the costs associated with the technologies, the cost of capital, the capacity utilisation, and the required energy investments.

i. “system composition, technology selection, and the unit cost of technology.”	
c. You assume learning rates for declining CED (for solar for example, via Supplemental Eq. 1 and Table S3 and batteries as in Table S25), but it is hard to know how much this assumption matches any historical data, or at least the values in the cited literature (even of co-author Breyer). I see no similar CED “learning curve” (decreasing CED values) for other technologies.	Learning rates are applied based on three studies. These are:  ▪ For PV: Görig, M., & Breyer, C. (2016). Energy Learning Curves of PV Systems. Environmental Progress & Sustainable Energy, 35(3), 914–923. https://doi.org/10.1002/ep.12340. While the value taken from that study is independently confirmed by Louwen et al. (2016) Re-assessment of net energy production and greenhouse gas emissions avoidance after 40 years of photovoltaics development, Nature Communications, 7, 13728. https://www.nature.com/articles/ncomms13728. ▪ For Battery: Xu, C., Dai, Q., Gaines, L. et al. Future material demand for automotive lithium-based batteries. Commun Mater 1, 99 (2020). doi.org/10.1038/s43246-020-00095-x. Supplementary Material Excel, Battery Sales by Capacity, Sustainable Development Scenario, NCX battery roadmap scenario ▪ For Electrolyser: Böhm, H., Goers, S., & Zauner, A. (2019). Estimating future costs of power-to-gas—A component-based approach for technological learning. International Journal of Hydrogen Energy, 44(59), 30789-30805. (Table 2 and Fig. 7, page 30798-30800). We used learning rates (LR) if a study provides a detailed analysis for a technology. The detailed estimation of LRs for the battery and electrolyser are provided in the above references, especially in the supplementary materials. In addition, estimates of CED values based on LR are provided in the CED database with relevant references. For highlighting the PV LR: The aforementioned Louwen et al. (2016) study confirms the energy learning rate for PV at a range of 12-13% whereas in Görig and Breyer (2016)⁷ the LR estimation stays in the range of 11-14%. Thus, the findings of the two publication are very close, notably, Görig and Breyer (2016)⁷ have more data points compared to Louwen et al.⁸.
11) (Apologies for not asking sooner) Do you calculate CED of any necessary transmission and distribution (grid) expansion? I don’t see this described anywhere (I might be missing it) as to whether grid energy investments were included. I see a discussion of grid energy (heat) dissipation losses.	We thank the Reviewer #3 for this question. We did not include the CED values of the transmission and distribution (T&D) lines as we are working on global power sector modelling. To start such a calculation, we need detailed data about the network structure and the future planning activities of the network. Furthermore, the data in ecoinvent on this topic is very limited in terms of the cable types and interface structure requirements. The LCIs of the T&D network in ecoinvent are based on the Swiss case and use SF6

	switch stations, which is usually not the case in developing countries. We have conducted internal roughly CED estimates on few types of T&D lines and found that these values remain at negligible levels compared to energy technologies. Thus, we excluded energy investments related to the T&D network, but we included losses related to this network. This is applied for all scenarios. The overall T&D grid infrastructure may not change considerably among the scenarios, thus, should be of negligible impact on the conclusions. To clarify this situation, we would like to add a statement in the “Limitations and uncertainties arisen from the nature of the models” section of this study, which is given below. In addition to this limitations, recycling of materials and waste is not considered in this study due to lack of sufficient specific data. The same situation occurs with the CED estimations related to the expansion of the transmission and distribution networks, and their CED values are at negligible amounts based on the limited LCIs given in the ecoinvent database. Thus, the EIs associated with these networks are not involved in the EROI estimates.
Minor Comments on the Article Content:	
12) You write “The modelling results reveal that the LCOE for all scenarios starts at 70.9 €/MWh in 2015 and gradually decreases as the RE deployment increases. This decline ranges from 45.2-49.7 €/MWh for BPS scenarios, 53.9-54.1 €/MWh for Teske/DLR scenarios, and 59.2-69.5 €/MWh for IEA scenarios⁶.” a. “This decline” does not refer to the values you state. Instead, the values you state refer to the 2050 values of your calculated LCOE. For example, for the “TESKE/DLR 2.0oC” scenario, the LCOE declines about 17 €/MWh from 71 €/MWh (in 2015) to about 54 €/MWh (in 2050). So the “decline” is 17 €/MWh, not 54 €/MWh. b. Need to reword this to either refer to the actual decline amount (2050 LCOE minus 2015 LCOE) or to the final 2050 LCOE value	We thank Reviewer #3 for pointing out this wording mistake. As the specified ranges are defined for the scenarios, we corrected this statement as follows: The modelling results reveal that the LCOE for all scenarios starts at 70.9 €/MWh in 2015 and gradually decreases as the RE deployment increases. The estimated LCOE remains in the range for 45.2-49.7 €/MWh for the BPS scenarios, 53.9-54.1 €/MWh for the Teske/DLR scenarios, and 59.2-69.5 €/MWh for the IEA scenarios⁶.
13) Per your new paragraph starting “Furthermore, the sustainability risk of the systemwide EROI is reanalyzed ...”, you	Thank for the correction. This has been corrected in the revised R3 manuscript.

wrote “Also, the shortening of the ET triggers the need for more energy for RE investment in the BPS-plus scenarios, as observed in the annual energy investment flow (Fig. S19).” a. I think you mean to refer to Fig. S20. Correct?	Also, the shortening of the ET triggers the need for more energy for RE investment in the BPS-plus scenarios (Fig.4b), as observed in the annual energy investment flows (Supplementary Fig. 20).
14) Per your new paragraph starting “Furthermore, the sustainability risk of the systemwide EROI is reanalyzed ...”, you wrote “Considering the CED in the year of the investment creates greater fluctuations in the annual energy investment flow during the ET (Fig. S21). Despite of a high expansion of RE, the estimated annual energy return on investment values stay in the low-risk range, as defined by Capellán-Pérez et al.22.” and separately state “The analysis shows that the annual energy return on investment values remain in the low-risk range, as defined by Capellán-Pérez et al.2, and not fall below 5.” a. My main point is you have misinterpreted Capellán-Pérez et al. statement in their paper. The risk you refer to is for “EROI standard” and they state that there is:  i. No risk: > 15:1 ii. Low risk: 10:1 < EROIst < 15:1 iii. Dangerous: 5:1 < EROIst < 10:1 iv. Very Dangerous: EROIst < 5:1 v. Unfeasible: EROIst < 2:1 to 3:1 b. Your ratio based on annual energy return on investment plotted in Fig. S21 have some values are less than 10:1 for some years in some of the scenarios, thus in the Capellán-Pérez et al. “dangerous” zone for that more comparable metric to their paper. c. Regardless, MY SUGGESTION is that I think you can just remove this sentence that refers to Capellán-Pérez et al. since you are not modeling 100% of the energy system (you can correct me) but only the electricity system (and its energy outputs and required energy inputs) and thus is not a good comparison to Capellán-Pérez et al. Capellán-Pérez et al. is modeling the entire energy system (electricity, liquid fuels, gaseous fuels) full set of	We assumed that this would be a relative indicator for our analysis, even though this categorisation was derived for the entire energy system. Since the heating and transport sectors are included, there is a probability that the EROI will decrease, and this ranges might change. At this point, we agree with the Reviewer #3 that it does not fully fit for our system. This reference is removed from the manuscript. We thank Reviewer #3 for the suggestion. The following sentence is revised as follows: Despite a high expansion of RE, the estimated annual energy return on investment values do not fall below 5.

input and outputs (e.g., total extraction of oil for EROIst calculation, and total output of gasoline+diesel+jet fuel+etc. for EROI_{pou} calculation). However, they assume EROI of oil, gas, and coal are each constant.	
15) Minor typo error: Remove “of” in “Despite of a high expansion of RE ...”	It is corrected based on the Reviewer #3 suggestion. The revision is shown in the previous comment.
16) You responded to me with “It has nothing to do with the amount of calculation that we need to perform. Deriving a new methodology for the systemwide PROI is still one of the key challenges to performing this analysis.” a. As far as I’m concerned, you have mostly satisfied my request (or desire) for using annualized energy flow via your Fig. S20 and S21, whether you call it PROI or not. Thank you. However, I’m still not 100% sure if the annualized input flows are:	PROI applies to power technologies, and involves time transfer of energy flows. It requires management of extensive data which requires exhausting computational effort, and time. To execute PROI correctly, we need to involve in the energy modelling phase, and adjust our model to generate annual results for global, regional, and country level (as we provide insights for regional EROIs in this paper). Though, the used model is structured to provide capacities at 5-year intervals, and the authors do not know any model that has ever document an hourly resolution (which is needed for 100% RE systems and storage) on global level on a 1-year interval for decades until 2050. Also, in the outputs of power sector modelling, we have enabling technologies, such as methanation unit (output is methane) and DAC (output is CO₂). They are treated as one of the components of power technologies – as in the case of CCS integration to fossil fuelled power plants. In the used model these technologies are treated as a sole technology because the outputs of them can be used and/or stored for several purposes as happens in real implementations. Also, the electricity consumption for these technologies is estimated internally in the modelling phase. On the other hand, when we go through entire energy systems, the generated methane from Power-to-X (PtX) processes can be converted to LNG by liquefaction process and/or partially feed to the gas technologies as e-methane through the storage technologies. Same is valid for DAC as its output can be used for multi-purposes. Hence, if we perform PROI per technology, capturing annual energy investment flow for only electricity generation technologies, instead of taking them as sole technology, we have to considered them a part of the electricity generation process used in one process chain, not used as for multi-purposes. This requires redefinition of the system boundary for related processes and technologies. Furthermore, this still will not fit for the used model because we have intermediary storage options for their outputs (gas

	storage technologies) and there are multi-process systems integrated. For instance, we won't know how much capacity of DAC is required for LNG or e-methane separately, as the model provides total capacity requirement for this technology. This is an example of the system-level consideration. Especially, it will be very difficult to adapt the entire system as the technology types in these PtX processes increase (e.g., the inclusion of Fischer-Tropsch and other related processes). For these reasons, we are not in favour of using technology-level EROI and PROI methods as they do not fit the conceptual principles of a sector-coupled energy system, for which the present power sector design is a first step, as pointed out above. However, we can easily implement this in the systemwide EROI estimation because the capacities are given, and we treat the entire system as an integrated one. Overall, the presented annualised energy investment flow analysis is not PROI. There are significant differences in system boundaries, conceptualisation and operating principles that indirectly affect the results. We hope this clears up the ongoing confusion.
i. 100% of CED brought to year 1 (or divided by 5 per year 5-year time step)? or ii. CED values spread out between 2 times of annualized flows, per what you have in your supplemental: (time 1): CED of installation and decommission (?) divided by 5 years when creating data points every 5 years, and (time 2) CED of operation used each time step. iii. Something else?	The 100% of CED values for construction and decommissioning are divided by 5 per year for 5-year time step. For instance, the newly installed capacity between 2025 and 2030 in absolute values is the total newly installed capacity for 5 years, thus, this total capacity is divided by 5 to get the annual value. The operational CED values are estimated for each time step using the FLH of the actual year. This is done to make sure that only annual values (and not integrals of 5 years) are considered, on the annual CED values, but also for the operational values.
b. If your calculations in Figure S20 and S21 can be clarified for me, then I might agree to accept publication of your paper, but I'd require (of course up to the editor) that Fig. S21 be placed in the main body of the paper and the mathematics of that calculation described in an equation.	The Fig. S21 is included in the main revised R3 manuscript as Fig. 4 and the related equations are provided in Methods section of the revised R3 manuscript.
Major Comments on the Supplemental:	
17) Your response to me per your Supplemental Section 6 includes: "Furthermore, the dynamics of electricity generation and energy requirements are captured on a micro-level in PROI at technology level, which is different than system-level analysis	Kindly revisit the previous comments on our explanation why PROI is not suitable for our case, especially on issues related the enabling technologies, modelling structure and operational principles. We would like to reiterate that most power sector modelling

where the interactions between technologies are present and reflected in the systemwide EROI analysis.” a. I’m not sure of your point, but I think it is incorrect. The PROI-like calculation using “annual energy investment flow” you have performed to create Figure S20 and S21 BY DEFINITION takes into account all technologies within the system. Right? The “annual energy investment flow” is the sum of annual energy investment for every technology you are assessing. Right?	studies implementing PROI do not include the enabling technologies covered in the applied modelling environment. Yes, both the systemwide EROI and the presented annual energy investment flow analysis cover all technologies, and, in both cases, the entire system is considered as one entire system. We did not implement the PROI exactly, but rather provide proximate approach.
b. I think you are using a caricature, or strawman, definition of PROI that does not have to hold. That is to say, a PROI calculation can be at a single technology level OR a system wide level simply by summing up each technology as you are doing (if you had the different parts of life cycle energy inputs timed properly).	Power Return on Investment (PROI) applies to electricity generation technologies. The LUT-ESTM includes several technologies that do not generate electricity, but deliver gases or CO₂ as output or just electricity storage without generation, and they are not a subcomponent of another technology. The main reasons why PROI is unsuitable in this context are given in previous comments and discussed in the R1 and R2. We would like to reiterate that the systemwide EROI methodology and all data provided in this study are based on the impact assessment concept (LCA), which is explicitly stated in the manuscript and SI.
c. My main argument within my review is that PROI is much more informative and appropriate for system-wide assessment than EROI, mainly because your EROI calculation is amortizing up-front energy investments that by definition (of physical laws) cannot happen in real life. And you have been arguing the opposite is true. Regardless, I have agreed that you can of course do the mathematics of your EROI as you are doing, but I’m also stating it is not very meaningful because of the distortion of the timing of mass and energy flows (it violates physical conservation laws) that is not informative to policymakers, but misinformative	We would like to refer several of the previous responses on this case. The proposed methodology and findings are useful for understanding energy transition scenarios from an EROI perspective. The power sector is moving towards integrating enabling technologies with renewable energy. These technologies should also be included in the EROI estimates. Furthermore, the aim of this paper is to provide a systemwide EROI methodology based on impact assessment concept (LCA). For additional information we have added the annual energy investment flow in the R2.
Minor Comments on the Supplemental:	
18) Please change “The annual energy investment flow (Fig. S20), which limits the lifetime effect, ...” to what I think is a more accurate distinction, because “lifetime effect” is not a clear term and not used anywhere else in the manuscript or supplemental (as	We thank Reviewer #3 for pointing out this issue. We revised this statement as Reviewer #3 suggested. The revised sentence is listed below.

far as my search indicates). I think you can rewrite it as something like: a. “The annual energy investment flow (Fig. S20), which does not amortize the up-front energy investment over the power plant lifetime after installation, ...”	The annual energy investment flow (Fig. S20), which does not amortise the upfront energy investment over the plant lifetime after installation, shows the higher shares of energy requirement for both the operational and investment phases.
19) Your response to me stated (with regard to data plotted in Fig. S20): a. “The same calculation method as the systemwide EROI is used, but the CED values of the technologies spread over their lifetime are converted to CED values per year and instead annual net capacities are taken.” b. What is the meaning of “annual net capacities are taken”?	Thanks for asking. We apologise for this confusion. We missed to correct this statement. It should be “and all estimations are annualised”.

References:

1. Aghahosseini, A. *et al.* Energy system transition pathways to meet the global electricity demand for ambitious climate targets and cost competitiveness. *Applied Energy* **331**, 120401 (2023).
2. Bhandari, K. P., Collier, J. M., Ellingson, R. J. & Apul, D. S. Energy payback time (EPBT) and energy return on energy invested (EROI) of solar photovoltaic systems: A systematic review and meta-analysis. *Renewable and Sustainable Energy Reviews* **47**, 133–141 (2015).
3. IRENA. Renewable Capacity Statistics 2023. (2023). Available at: <https://www.irena.org/Publications/2023/Mar/Renewable-capacity-statistics-2023>. (Accessed: 6th September 2023)
4. King, C. W. & Hall, C. A. S. Relating financial and energy return on investment. *Sustainability* **3**, 1810–1832 (2011).
5. IRENA. Renewable Power Remains Cost-Competitive amid Fossil Fuel Crisis. (2022). Available at: <https://www.irena.org/news/pressreleases/2022/Jul/Renewable-Power-Remains-Cost-Competitive-amid-Fossil-Fuel-Crisis>. (Accessed: 16th November 2023)
6. IEA. *World Energy Investment 2023*. (2023). doi:10.1787/e0e92e98-en
7. Görig, M. & Breyer, C. Energy Learning Curves of PV Systems. *Environmental Progress & Sustainable Energy* **35**, 914–923 (2016).
8. Louwen, A., Van Sark, W. G. J. H. M., Faaij, A. P. C. & Schropp, R. E. I. Re-assessment of net energy production and greenhouse gas emissions avoidance after 40 years of photovoltaics development. *Nature Communications* **7**, 1–9 (2016).